# GLANCE: GLOBAL ACTIONS IN A NUTSHELL FOR COUNTERFACTUAL EXPLAINABILITY

## ABSTRACT

The widespread deployment of machine learning systems in critical real-world decision-making applications has highlighted the urgent need for counterfactual explainability methods that operate effectively. Global counterfactual explanations, expressed as actions to offer recourse, aim to provide succinct explanations and insights applicable to large population subgroups. Effectiveness is measured by the fraction of the population that is provided recourse, ensuring that the actions benefit as many individuals as possible. Keeping the cost of actions low ensures the proposed recourse actions remain practical and actionable. Limiting the number of actions that provide global counterfactuals is essential to maximize interpretability. The primary challenge, therefore, is balancing these trade-offs—maximizing effectiveness, minimizing cost, while maintaining a small number of actions. We introduce GLANCE, a versatile and adaptive framework, comprising two algorithms, that allows the careful balancing of the trade-offs among the three key objectives, with the size objective functioning as a tunable parameter to keep the actions few and easy to interpret. C-GLANCE employs a clustering approach that considers both the feature space and the space of counterfactual actions, thereby accounting for the distribution of points in a way that aligns with the structure of the model. T-GLANCE provides additional features to enhance flexibility. It employs a tree-based approach, that allows users to specify split features, to build a decision tree with a single counterfactual action at each node that can be used as a subgroup policy. Our extensive experimental evaluation demonstrates that our method consistently shows greater robustness and performance compared to existing methods across various datasets and models.

## 1 INTRODUCTION

Machine learning models are increasingly deployed in critical domains such as loan approvals, hiring, and healthcare. This widespread adoption has intensified the need for transparency and interpretability in model decisions, requiring users to understand how their input features influence the outcomes and how they might change them to achieve favorable outcomes, known as recourse (Miller, 2019). Counterfactual explanations have gathered extensive attention for their suitability for achieving algorithmic recourse (Karimi et al., 2020), their interpretability (Wachter et al., 2017), actionability (Ustun et al., 2019), utility in fairness audits (Sharma et al., 2019; Kavouras et al., 2023), etc. A counterfactual action, or simply an *action*, defines the specific feature changes needed to convert an unfavorable decision into a favorable one.

Traditionally, counterfactual explanations refer to *local explainability*, being tied to a particular negatively affected instance. However, in many real-world scenarios, *global counterfactual explainability* is more useful, offering generalized explanations that apply across the affected population. While a collection of all local counterfactuals could technically cover all affected individuals, this approach sacrifices interpretability, which is central to global explainability.

Similar to past work (Rawal & Lakkaraju, 2020; Kanamori et al., 2022), we define Global Counterfactual Explanations (GCE) as a small set of *global actions* that provide effective recourse for the affected population. Any global counterfactual solution must meet three objectives as identified from previous research: (1) be composed of a small number of actions to ensure interpretability (size),

(2) minimize the cost of implementing those actions (cost), and (3) offer recourse to as many affected individuals as possible (effectiveness).

As noted by Branke (2008), the relationships between multiple optimization objectives are often complex, and aggregating them into a single objective, which is common in practice (Rawal & Lakkaraju, 2020), can be problematic since they are typically non-commensurable. Framing GCEs as multi-objective optimization, allows us to explore the inherent *trade-offs* between effectiveness and cost, especially when the solution size is constrained.

To understand this optimization problem, assume a 2-d numerical feature space, and consider the two affected instances $x_1, x_2$ depicted in Figure 1a. Assume that recourse cost equals the distance to the decision boundary, drawn as a line in the figure. Observe that $a_1$ (resp. $a_2$) is the local action that provides recourse for $x_1$ (resp. $x_2$) at minimum cost.

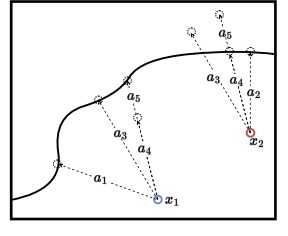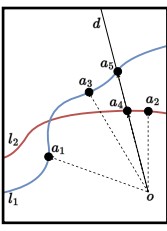

(a) Feature space  (b) Action space

Figure 1: A toy example depicting two negative instances $x_1, x_2$, and five actions: $a_1, a_2$ are the optimal local actions for $x_1, x_2$; $a_3$ is the optimal GCE maximizing effectiveness; $a_4$ and $a_5$ are actions along the optimal direction $d$ as defined in Ley et al. (2023) (a) The feature space, where the line is the decision boundary. (b) The action space, where $l_1, l_2$ depict the decision boundary from the perspective of $x_1, x_2$, respectively.

Further, consider the *action space* depicted in Figure 1b, where every action is represented as a point (or equivalently a vector relative to the center $o$ of the coordinate system). The blue $l_1$ and red $l_2$ lines represent the decision boundary seen from the perspectives of $x_1$ and $x_2$, respectively. The blue line $l_1$ separates the actions that provide recourse for $x_1$ (any action on the outside, away from $o$) from those that do not. Action $a_1$ lies on $l_1$, and is the closest point to $o$, and thus the min-cost local action for $x_1$. Similarly, the red line $l_2$ concerns $x_2$ and contains its min-cost action $a_2$.

Consider now the problem of finding a single action GCE. To provide recourse for both $x_1$ and $x_2$, we look for an action that lies outside both lines in Figure 1b. Among all such actions, $a_3$ has the minimum cost and thus is the optimal global action that maximizes effectiveness. If we trade off effectiveness for cost, $a_2$ is the optimal global action that minimizes cost, but has 50% effectiveness (brings recourse to $x_2$ but not to $x_1$).

GCE gives rise to a different optimization problem than its local counterpart. Even if optimal local actions are generated for each instance and a small subset is chosen as the global actions, this may still result in a suboptimal GCE. In our example, the optimal global action $a_3$ is not an optimal local action for either $x_1$ or $x_2$; in fact, $a_3$ can be viewed as a compromise between $a_1$ and $a_2$, the locally optimal actions. Finding good GCEs requires a nuanced exploration of the action space.

In this work, we distinguish between two variants of GCE that differ in how they assign actions to individuals. In *implicit* GCE, each individual chooses the best (i.e., min-cost) global action in GCE to achieve (if possible) recourse (as in e.g., Ley et al., 2023). Implicit GCE is helpful when one wishes to obtain a model-level understanding of recourse, or to select a global (horizontal) policy to apply to the entire affected population. In *explicit* GCE, individuals are partitioned according to their features and a single action applies to each partition (as in e.g., Rawal & Lakkaraju, 2020; Kanamori et al., 2022). Explicit GCE provides a more interpretable summary as it associates actions to partitions of the feature space, and can also be used to implement targeted policies to bring recourse to subpopulations.

## 1.1 KEY CONTRIBUTIONS

Recognizing that both the effectiveness and the cost of actions depend on the individuals' features and the model's decision boundary, we propose a novel clustering approach for GCEs that considers both feature and action spaces. Unlike previous claims that clustering is ineffective for global counterfactuals (see e.g., Kanamori et al. (2022)), we are the first to propose the joint clustering in feature and action spaces and demonstrate its effectiveness. We introduce GLANCE, a novel and highly adaptable framework, that consists of two algorithms based on hierarchical clustering techniques, C-GLANCE and T-GLANCE. GLANCE bridges local and global explanations by combining

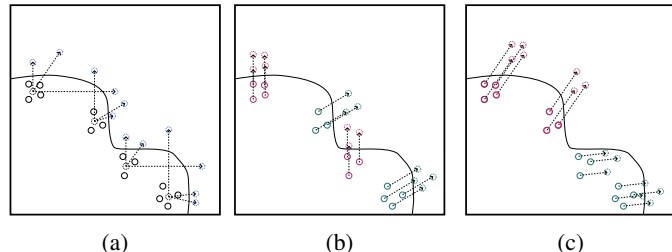

Figure 2: Intuition behind clustering approaches. (a) First, `C-GLANCE` generates diverse candidate action from the centroids of feature-based clusters. (b) Then, `C-GLANCE` merges clusters based on similarity in either the feature space or the possible recourse actions, grouping instances that may be further apart but can be explained by similar actions. (c) An alternative where clusters are formed solely based on the feature space may lead to suboptimal global actions—either high effectiveness at a very high cost or low cost but very low effectiveness.

clustering with local counterfactual generation, offering a scalable and effective methodology for GCEs. `GLANCE` balances actionable, minimal changes with interpretability through a tunable size objective, ensuring practical and interpretable solutions.

**C-GLANCE.** We present `C-GLANCE`, a clustering-based approach for the implicit GCE problem that returns $s$ global actions. At a high level, `C-GLANCE` consists of two phases: candidate action generation, and action selection.

To generate candidates, `C-GLANCE` first performs a fine-grained clustering of the individuals in the feature space. For each cluster, `C-GLANCE` then discovers a few diverse actions that bring recourse to the cluster centroid. The intuition here is that highly similar individuals are expected to have a similar perspective of the decision boundary, and can thus achieve recourse through similar cost-efficient actions. Figure 2a shows an example with four fine-grained clusters, where three diverse actions are discovered from each centroid. Each action represents an alternate way to bring recourse to cluster members, giving `C-GLANCE` the flexibility to later select among them.

At the end of the first phase, `C-GLANCE` has associated each cluster of individuals with a set of actions. In the second phase, `C-GLANCE` iteratively merges such clusters until exactly $s$ clusters remain. Two clusters are merged if they either concern similar individuals or contain similar actions. The intuition behind the former is similar to that of the candidate generation phase. The intuition for the latter is that if similar actions bring recourse to them it should be easier to identify a common low-cost, highly effective action, even though they are dissimilar feature-wise. Figure 2b shows an example where the red individuals are grouped together either through affinity in the feature or in the action space.

As discussed in (Kanamori et al., 2022), performing a coarse-grained clustering on feature space and then assigning a single action per cluster leads to inadequate global actions, that suffer either in cost or effectiveness. For example, in Figure 2c, individuals are clustered based on feature similarity and a single action is selected per cluster. Because the blue cluster is associated with rather dissimilar actions, choosing a single actions leads to a compromise in either effectiveness (as depicted) or cost.

**T-GLANCE.** Recognizing the need for explicit GCE, we present `T-GLANCE`, an algorithm employing a tree-based approach, similar to explainable clustering techniques (e.g., Moshkovitz et al., 2020). This allows assigning actions to individuals while satisfying the transparency and consistency properties required by Kanamori et al. (2022). `T-GLANCE` also allows users to select features that define specific groups, to produce targeted GCEs. For the user's facilitation, we also provide suggestions based on the importance feature permutation score for selecting features to be used in splits. `T-GLANCE` directly supports policymaking and audit scenarios—an application not addressed by existing methods. As shown in our experiments, `T-GLANCE` delivers competitive results, outperforming previous tree-based methods while maintaining efficiency.

**Experimental Evaluation.** We perform a comprehensive evaluation across various datasets and models, demonstrating that `GLANCE` consistently outperforms or matches state-of-the-art methods in both cost and effectiveness while remaining computational efficiency. To facilitate comparison in global counterfactual methods, we invoke Pareto dominance criteria from multiobjective opti-

mization, further showcasing GLANCE's stable, superior performance across diverse datasets and models.

## 1.2 RELATED WORK

**Counterfactual Explanations.** There has been a plethora of work with an emphasis on counterfactual explanations (Verma et al., 2020). An overview of methods on algorithmic recourse, which provides explanations and recommendations to individuals impacted by automated decision-making systems is presented by Karimi et al. (2020). These methods can be model-agnostic or model-specific (e.g., for trees Carreira-Perpiñán & Hada, 2021), focus on properties like diversity (Mothilal et al., 2020) or feasibility (Ustun et al., 2019)), global or local. While local counterfactuals are well defined, global counterfactuals present challenges, as they must provide recourse for all individuals within a specific group in the same manner while maintaining explainability and staying true to the notion of local counterfactual, which focuses on minimal changes. Achieving this balance globally presents significant difficulties.

Prior work does not make the distinction between explicit and implicit GCE. Regarding explicit GCE, Rawal & Lakkaraju (2020) introduced AReS, a framework for global counterfactual explanations, that jointly optimizes recourse correctness, coverage, and cost, providing an interpretable summary of recourses, expressed in a two-level rule set. However, the AReS framework may fail to cover the entire population. Ley et al. (2022) later improved the computational efficiency with Fast AReS. In another direction, Kanamori et al. (2022) introduced CET, which partitions the space and assigns an action to each part transparently and consistently. Although effective, its computational complexity limits scalability.

Regarding implicit GCE, Warren et al. (2023) developed Group-CF, that generates counterfactuals that seeks to maximize effectiveness, though it can result in higher costs. Ley et al. (2023) proposed GLOBE-CE, where global counterfactual explainability is defined differently, as a small set of action *directions* along which individuals can "move" to achieve recourse. Therefore, GLOBE-CE attacks a different optimization problem than the one we study here. For the toy example in Figure 1, the single direction that minimizes the total recourse cost for $x_1$ and $x_2$ is the direction $d$ depicted in Figure 1b. This direction contains actions $a_4$ and $a_5$ that bring recourse to $x_2$ and $x_1$, respectively, with minimum cost *along* $d$. It is important to note that neither $a_4$ nor $a_5$ is optimal as a GCE (or for local explainability). Even the set $\{a_4, a_5\}$ is a suboptimal GCE—the set $\{a_1, a_2\}$ dominates it with equal size and effectiveness but lower total cost. In general, (1) directly translating the output of GLOBE-CE into a GCE leads to numerous micro-actions (two in our example, but potentially as many as the number of individuals), which reduces interpretability, (2) choosing a few actions along the optimal directions may result in suboptimal GCEs (in our example, any single action along $d$ is dominated by $a$), and (3) since directions lack clear endpoints, they fail to specify the magnitude of change required, leading to uncertainty for individuals seeking recourse and limiting real-world applicability.

Other works include Carrizosa et al. (2024a;b), who use mixed-integer quadratic models for group-level explanations, and Koo et al. (2020), who employ Lagrangian methods. Research has also extended to generating global counterfactuals for graphs (Huang et al., 2023) and auditing subgroup fairness (Kavouras et al., 2023).

**Global Explainability.** Many works in global explainability are related to our approach. For instance, Chowdhury et al. (2022) summarize model logic by dividing the desired region of explanation features into subspaces based on similar logic, referred to as equi-explanation maps. Similarly, Lakkaraju et al. (2019) provides explanations for different subspaces characterized by specific features. Their method also allows users to customize the model explanations by selecting features of interest, akin to the flexibility offered by T-GLANCE. While these works share a conceptual focus on subspace-based explanations, they differ fundamentally from ours in their ultimate goals and outcomes. Both primarily aim to summarize model behavior or provide faithful subspace-level explanations, whereas GLANCE seeks to generate actionable global counterfactual explanations tailored to decision-making processes. Consequently, the direct comparison of these approaches with T-GLANCE is not entirely applicable.

**Other Related Work.** Many clustering algorithms lead to clusters that are hard to explain. There is a plethora of work on explainable or interpretable clustering, mainly using decision trees that under

specific assumptions come with provable guarantees, including the works of Dasgupta et al. (2020); Moshkovitz et al. (2020); Frost et al. (2020); Makarychev & Shan (2022); Bertsimas et al. (2021); Laber & Murtinho (2021), etc. Another area of related work is multiobjective optimization and Pareto optimality. We refer the reader to Branke (2008).

## 2 PROBLEM FORMULATION

We consider a black box binary classifier $h : \mathcal{X} \to \{-1, 1\}$, where the positive outcome is favorable and the negative is unfavorable. We will focus on the set $\mathcal{X}_{\text{aff}} \subseteq \mathcal{X}$ of **adversely affected individuals**, i.e., those who receive the unfavorable outcome. We denote as $\mathbb{A}$ the set of all possible actions (which is potentially infinite), where an **action** $a \in \mathbb{A}$ is a set of changes to feature values, e.g., $a = \{\text{country} \to \text{US}, \text{education-num} \to +2\}$, which, when applied to an instance $x \in \mathcal{X}_{\text{aff}}$, results in a **counterfactual** instance $x' = a(x)$. Every action $a$ has a cost, denoted as $\text{cost}(a, x)$, and is effective for an instance $x$ if $h(a(x)) = h(x') = 1$. Let $\mathbb{C} \subseteq \mathbb{A}$. The recourse cost $\text{rc}(\mathbb{C}, x)$ of an instance $x$ is the minimum cost incurred from an effective action in $\mathbb{C}$:

$$\text{rc}(\mathbb{C}, x) = \min\{\text{cost}(a, x) | a \in \mathbb{C} : h(a(x)) = 1\}$$

Let $X_{\mathbb{C}} = \{x \in \mathcal{X}_{\text{aff}} \,|\, h(a(x)) = 1, a \in \mathbb{C}\}$ be the set of instances that flip their prediction using one of the actions in $\mathbb{C}$. Then the effectiveness, also known as coverage Ley et al. (2022), of $\mathbb{C}$ for the affected instances $\mathcal{X}_{\text{aff}}$ is defined as the percentage of $\mathcal{X}_{\text{aff}}$ that managed to flip their prediction using one of the actions in $\mathbb{C}$:

$$\text{eff}(\mathbb{C}, \mathcal{X}_{\text{aff}}) = \frac{|X_{\mathbb{C}}|}{|\mathcal{X}_{\text{aff}}|},$$

The cost of $\mathbb{C}$ in $\mathcal{X}_{\text{aff}}$ is defined as the **average recourse cost** of the instances in $\mathcal{X}_{\text{aff}}$ :

$$\text{avc}(\mathbb{C}, \mathcal{X}_{\text{aff}}) = \frac{\sum\limits_{x \in X_{\mathbb{C}}} \text{rc}(\mathbb{C}, x)}{|X_{\mathbb{C}}|}.$$

Finally, let $\text{size}(\mathbb{C}) = |\mathbb{C}|$ denote the cardinality of a set $\mathbb{C}$.

As it is clear, multiple sets of actions can produce recourse for $\mathcal{X}_{\text{aff}}$ and the quality of such a set is a factor of the three notions we introduced: effectiveness, cost, and size. An ideal global counterfactual should maximize effectiveness while minimizing the cost and the size, based on the properties that state-of-the-art works (Rawal & Lakkaraju, 2020; Kanamori et al., 2022; Ley et al., 2023; Huang et al., 2023) have argued are essential. Formally, we define the global counterfactual explanation problem as follows:

**Problem 1** (Global Counterfactual Explanations (GCE)). *Given a black box model $h$ that classifies the $\mathcal{X}_{\text{aff}}$ instances to the negative class, our goal is to find the set $\mathbb{C} \subseteq \mathbb{A}, \mathbb{C} \neq \emptyset$ that represents a solution to the following multi-objective optimization:*

$$minimize \quad \left( \text{size}(\mathbb{C}), \; -\text{eff}(\mathbb{C}, \mathcal{X}_{\text{aff}}), \; \text{avc}(\mathbb{C}, \mathcal{X}_{\text{aff}}) \right)$$

The requirement for a small set of actions is to enhance the interpretability of the explanation. This can also be expressed as a constraint to the set size we can afford. In this case, the third objective of Problem 1 is replaced by the constraint $\text{size}(\mathbb{C}) \leq s$, where $s$ is a small positive integer. For the rest of the paper, we will use the following problem formulation:

**Problem 2** ($s$-GCE). *Given a black box model $h$ that classifies the $\mathcal{X}_{\text{aff}}$ instances to the negative class, our goal is to find the set $\mathbb{C} \subseteq \mathbb{A}$ that represents a solution to the following bi-objective optimization problem:*

$$minimize \quad \left( -\text{eff}(\mathbb{C}, \mathcal{X}_{\text{aff}}), \; \text{avc}(\mathbb{C}, \mathcal{X}_{\text{aff}}) \right)$$
$$s.t. \quad \text{size}(\mathbb{C}) \leq s$$

If the set of actions $\mathbb{A}$ is finite and explicitly given, $s$-CGE is NP-hard. For more details, we refer the reader to Appendix A. For insights into how increasing the size impacts effectiveness and average cost, and the importance of selecting actions to maintain a desirable balance, see Appendix B.

---

**Algorithm 1** C-GLANCE

---

**Input:** $\mathcal{X}_{\text{aff}}, m, k, s$ {$\mathcal{X}_{\text{aff}} :=$ affected individuals , $m :=$ number of candidate actions to generate, $k :=$ initial number of clusters, $s :=$ number of global counterfactual actions}

**Output:** $s$ global counterfactual actions

1: $C \leftarrow \texttt{cluster}(\mathcal{X}_{\text{aff}}, k)$ {Cluster $\mathcal{X}_{\text{aff}}$ into $k$ initial clusters}
2: **for** $c \in C$ **do**
3: $\quad \boldsymbol{ca}(c) \leftarrow \texttt{actions}(\texttt{centroid}(c), m)$ {For each cluster centroid generate $m$ actions}
4: **end for**
5: **while** $|C| > s$ **do**
6: $\quad \{c_1, c_2\} \leftarrow \arg\min_{\{c_1,c_2\} \subseteq C}(d_1(\texttt{centroid}(c_1), \texttt{centroid}(c_2)) + d_2(\texttt{average}(\boldsymbol{ca}(c_1)), \texttt{average}(\boldsymbol{ca}(c_2))))$
$\qquad\qquad\qquad\qquad\qquad$ {Find $c_1$,$c_2$ that minimize $d_1$+$d_2$}
7: $\quad C \leftarrow \texttt{merge}(\{c_1, c_2\} \in C)$ {Merge $c_1$,$c_2$ into one cluster}
8: $\quad \boldsymbol{ca}(\{c_1, c_2\}) \leftarrow \boldsymbol{ca}(c_1) \cup \boldsymbol{ca}(c_2)$ {Merge their action sets}
9: **end while**
10: **return** $\texttt{bca}(ca(c))$ for each $c \in C$ {Return optimal action from each of the $s$ clusters}

---

## 3 C-GLANCE

`C-GLANCE` (Algorithm 1) presents a clustering algorithm that follows an agglomerative approach. The first step is to establish multiple small clusters and determine representative diverse actions for each. The initial clusters will undergo merging, exploiting the notion of proximity in both the original feature space and the action space, until no more than $s$ clusters exist. Finally, a single action will be extracted from each cluster with universal applicability across all instances. For the time complexity analysis we refer the reader to Appendix C.

**Algorithm Description.** *(1) Initial clusters and actions generation.* The feature space is partitioned into $k$ clusters, by a clustering algorithm (e.g., $k$-means, line 1, alg. 1). We compute the *centroid* of each cluster and generate $m$ *diverse* counterfactual actions for each centroid (line 3, alg. 1), employing any candidate counterfactual generation method (see Appendix G for the methods used in the experimental evaluation). We aim to efficiently and effectively explore the action space by generating actions from widely dispersed points within the feature space and guiding them in diverse directions that cross the decision boundary.

*(2) Merging.* Similar to agglomerative clustering, our approach merges clusters with the goal of maximizing intra-cluster similarity and minimizing inter-cluster similarity with respect to both the feature and action spaces. Specifically, similarity is computed by a metric $D$, which is the sum of two distances: $d_1 : \mathcal{X}_{\text{aff}} \times \mathcal{X}_{\text{aff}} \rightarrow \mathbb{R}$, the distance between the centroids of each cluster, and $d_2 : \mathbb{A} \times \mathbb{A} \rightarrow \mathbb{R}$, the distance between the average counterfactual actions of each cluster; other formulations for $d_2$ are also possible, e.g., Wasserstein distance. Thus, clusters are combined based on both their feature proximity and the proximity of their respective actions. Until the desired number $s$ of clusters is reached, we merge (line 7, alg. 1) the two clusters that minimize the total distance $d_1 + d_2$ (line 6, alg. 1). Merging two clusters means that their action sets are combined, ensuring that the most effective actions for each cluster are retained (line 8, alg. 1).

*(3) GCE extraction and optimality.* For each of the $s$ clusters, we extract a single candidate action, and return the set of these $s$ actions (line 10, alg. 1). Assuming that the action generation and subsequent merging has succesfully grouped together individuals that can achieve recourse through similar cost-efficient actions, in the final step we target effectiveness, selecting at each cluster the optimal action, in terms of effectiveness, among those associated with the cluster.

**Approach Strengths.** `C-GLANCE` supports various clustering methods, cost metrics, and action generation techniques making it adaptable to different applications. In the App. Table 13, we present the results using various action generation methods, all of which lead to near-optimal solutions in terms of the effectiveness-cost tradeoff. Additionally, `C-GLANCE` demonstrates speed, robustness, and near-optimal solution quality across datasets, as shown in the experimental evaluation.

## 4 T-GLANCE

`T-GLANCE` (Algorithm 2) constructs a hierarchical partitionining of the feature space. Each partition level is derived by splitting one cell of the previous level partition along a feature, resulting in a decision tree like structure akin to classification and regression trees (CART) (Breiman et al., 1984; Bertsimas & Dunn, 2017). Each cell (whether an internal or leaf tree node) can be described as a conjunction of feature predicates, and is associated with a single counterfactual action, providing

---

**Algorithm 2** T-GLANCE

**Input:** $\mathcal{X}_{\text{aff}}$, $m$, $s$, $D'$. {$\mathcal{X}_{\text{aff}}$ := affected individuals, $m$ :=number of candidate actions to generate, $s$ := maximum number of leaf nodes, $D'$, a subset of possible features}
**Output:** $\mathcal{P}$ {The tree and the actions assigned in each node.}
1: $\mathcal{P}_0 \leftarrow \{\mathcal{X}_{\text{aff}}\}$ {The partition associated with the root of the tree.}
2: Assign to $\mathcal{P}_0$ action $a_0 = \text{bca}(\boldsymbol{ca}(\mathcal{P}_0), \mathcal{P}_0)$ {Generate actions $\boldsymbol{ca}(\mathcal{P}_0)$. $a_0$ maximizes eq.(1)}
3: $level \leftarrow 0, n_{nodes} \leftarrow 1, queue \leftarrow \mathcal{P}_0$
4: **while** $n_{nodes} + 1 \leq s$ **do**
5:     **if** $queue = \emptyset$ **then**
6:         $level \leftarrow level + 1$, $queue \leftarrow \mathcal{P}_{level}$
7:     **end if**
8:     Pick $A$ the first element in $queue$.
9:     $i \leftarrow \arg\max_{j \in D'} V(A, j)$ {Choose the split that maximizes the criterion of eq.(2).}
10:     **if** $V(A, i) \geq 0$ **then**
11:         Split the cell A into cells $A_k^i = \{x | x \in A \wedge x_i \in v_i(k)\}$, $k = 0, 1$
12:         Assign to $A_k^i$ action $a_k^i = \text{bca}(\boldsymbol{ca}(A_k^i), A_k^i)$ { Generate actions $\boldsymbol{ca}(A_k^i)$. $a_k^i$ maximizes eq.(1). }
13:         $queue \leftarrow queue \setminus \{A\}, \mathcal{P}_{level+1} \leftarrow \mathcal{P}_{level+1} \cup \{A_0^i, A_1^i\}$
14:     **else**
15:         $queue \leftarrow queue \setminus \{A\}$ {Do not split this node and continue to the next.}
16:     **end if**
17: **end while**
18: $\mathcal{P}_{level+1} \leftarrow \mathcal{P}_{level+1} \cup queue$
19: **return** $\mathcal{P}_{level+1}$ { Return the tree and the assigned actions.}

---

thus *interpretable* counterfactual explanations at multiple granularity levels. `T-GLANCE` decides to split a cell if more effective actions can be found for the child cells. Splits are performed along a user-provided subset of features. Once a feature is used to split a cell, it is not considered again for further splits. As a result, the height of the tree is limited by the cardinality of the user-provided feature subset. We next present the necessary definitions to introduce Algorithm 2. For the time complexity analysis, please refer to Appendix C.

**Definitions.** A partition $\mathcal{P}$ of $\mathbb{R}^d$ is a family of sets $\{A_1, \ldots, A_l\}$ such that $A_j \subseteq \mathbb{R}^d$, $A_j \cap A_k = \emptyset$ for all $j, k \in [l]$ and $\bigcup_{j=1}^s A_j = \mathbb{R}^d$. Let $\mathcal{P}$ a partition of $\mathcal{X}_{\text{aff}}$. Every element $A$ of $\mathcal{P}$ is called a cell of $\mathcal{P}$ or just a cell. Each cell can be further divided into two subcells $A_0^i, A_1^i$ with respect to any direction $i$, defined as $A_0^i = \{x \in A | x_i \in v_i(0)\}$ and $A_1^i = \{x \in A | x_i \in v_i(1)\}$, where $v_i(k)$ represent the values that correspond to the specific split (e.g., 0 or 1 if it is a binary feature, specific ranges for continuous features). Let $\boldsymbol{ca}(A) := \text{actions}(c, m)$ be the set of $m$ generated actions for the centroid $c$ of the cell $A$. For simplicity, we will use $\text{bca}(\boldsymbol{ca}(A), A)$ to refer to the action in $\boldsymbol{ca}(A)$ with the highest effectiveness. Formally,

$$\text{bca}(\boldsymbol{ca}(A), A) = \arg\max_{a \in \boldsymbol{ca}(A)} \{\text{eff}(\{a\}, A)\} \tag{1}$$

We define a local version of the *total effectiveness gain* achieved by a specific portion split $i$ to a local cell $A$.

$$V(A, i) \triangleq \text{eff}(\{\text{bca}(\boldsymbol{ca}(A), A)\}, A) - \overline{\text{eff}}(\{\text{bca}(\boldsymbol{ca}(A_k^i), A_k^i)\}, A_k^i)$$

$$= \text{eff}(\{\text{bca}(\boldsymbol{ca}(A), A)\}, A) - \frac{1}{|k|} \sum_{k=\{0,1\}} \text{eff}(\{\text{bca}(\boldsymbol{ca}(A_k^i), A_k^i)\}, A_k^i) \tag{2}$$

**Algorithm Description.** `T-GLANCE` selects a different direction for splitting at each iteration at every cell in the current partition. Specifically, the algorithm begins with a single cluster containing all affected instances and generates $m$ *diverse* actions for the centroid, retaining the best action according to the criterion in eq. (1) (lines 1–2, alg. 2). The root then is established as the first cell $A$ to be examined, and the possible splits are examined based on the actions that were generated for each split (lines 8–9, alg. 2). For simplicity, we present the algorithm with binary splits (branching factor of 2), though it can be generalized to accommodate any branching factor. We proceed with the split that maximizes eq. (2), provided it yields a nonnegative value, and add the new cells as nodes of the tree to be examined in the next repetitions (lines 10–13, alg. 2); otherwise the node is not split. Importantly, the set of directions allowed for splitting is a subset of the features, provided by the user, allowing for controlled and interpretable partitioning of the data.

**Termination Criteria.** This process continues until one of the following termination criteria occurs: (1) a node has no remaining features to consider for splitting, (2) the maximum score of the actions

from all remaining features is negative, or (3) the specified number of leaf nodes has been reached (i.e., size constraints).

**Approach Strengths.** A key strength of `T-GLANCE` is its user-driven flexibility, allowing policymakers or auditors to select the features that determine tree partitioning. By allowing users to guide the feature selection, `T-GLANCE` can adapt to specific policy needs or audit requirements, making it highly customizable. `T-GLANCE` provides multi-level solutions, assigning a single optimal counterfactual action to each tree node, ensuring transparency and consistency across decision-making processes. This enables both high-level and granular solutions for additional user flexibility, in contradiction with Kanamori et al. (2022), who optimize actions only at the leaf nodes. Additionally, the method remains efficient, robust, and effective across various datasets and scenarios, as shown in our evaluations, making it well-suited for real-world applications. Finally, unlike the stochastic local search used by Kanamori et al. (2022), which may not converge (in about 20% of our experiments), `T-GLANCE` guarantees a solution at every node.

## 5 EXPERIMENTAL EVALUATION

### 5.1 EXPERIMENTAL SETTING

**Baselines.** We compare `GLANCE` framework methods against state-of-the-art methods in Global Counterfactual Explanations, specifically: `AReS` (using the `Fast AReS` implementation), `CET`, `GroupCF`, and `GLOBE-CE`, all of which are constrained by a predefined action set size. Details on the implementation used for each method can be found in Appendix F.

**Datasets.** We use four established benchmark datasets from previous research: COMPAS (Angwin et al., 2016), German Credit (Dua & Graff, 2019), Default Credit (Yeh & Lien, 2009), and HELOC (Brown et al., 2018). Additionally, we introduce the Adult dataset (Becker & Kohavi, 1996) for further evaluation. Details on the datasets and their preprocessing can be found in Appendix D.

**Models.** We trained three different model types: XGBoost (XGB), Logistic Regression (LR), and Deep Neural Network (DNN). Hyperparameters and training accuracy statistics are provided in Appendix E. We used 5-fold cross-validation, to also evaluate the robustness of the results.

**Recourse Cost.** For computing recourse costs, we adhered to the guidelines established by Ley et al. (2023).

**Reproducibility.** All experiments were conducted on an in-house server with cloud infrastructure equipped with an Intel(R) Core(TM) i9-10900X CPU @ 3.70GHz, 128 GB of RAM. No GPU acceleration was utilized during these experiments. We provide the code for the reproducibility of our experiments as supplementary material.

**Running Time.** In our experiments, `GLOBE-CE` and `dGLOBE-CE` were the fastest methods, consistently delivering solutions within 20 seconds across all datasets and models. `C-GLANCE` and `T-GLANCE` followed, typically completing in under 300 seconds, although `T-GLANCE` reached up to 850 seconds in four cases. Other methods were slower, with `Fast AReS` ranging typically between 150–400 seconds, and peaking at 1,400 seconds in some runs. The least computationally efficient were `Group-CF` and `CET`, with maximum runtimes of 3,500 and 17,000 seconds, respectively, and `CET` failed to solve the underlying optimization problem after 20 hours of runtime for the Adult dataset across all models.

### 5.2 EXPERIMENTAL EVALUATION

Table 1 presents the summarized results of all competing methods. We compare `C-GLANCE` and `T-GLANCE` against the five other competitors across five datasets and three models, resulting in 75 head-to-head comparisons for each method, totaling 150. However, since `CET` failed to solve the underlying optimization problem for the Adult dataset across all models, the final count is 72 comparisons per `GLANCE` method. Furthermore, we perform additional evaluations by categorizing the competitors into implicit and explicit methods. Specifically, `C-GLANCE` is evaluated against implicit—`Group-CF`, `GLOBE-CE`, and `dGLOBE-CE`— resulting to 45 head-to-head comparisons. Similarly, `T-GLANCE` is compared with explicit competitors — `Fast AReS`, `CET`— yielding 27 head-to-head comparisons. Recall that all reported results concern the $s$-GCE problem for $s = 4$;

Table 1: Evaluating the effectiveness and average cost of `T-GLANCE` against explicit (`Fast AReS`, `CET`) and `C-GLANCE` against implicit (`Group-CF`, `GLOBE-CE`, and `dGLOBE-CE`) GCE methods for $s$-GCE problem with $s = 4$. $s$-GCE solutions with effectiveness below 80% (applicability threshold) are highlighted in red. Non-robust GCEs, identified by either a standard deviation (std) in effectiveness greater than 5% across folds or a std in cost greater than half the average cost, are highlighted in blue.

| MODELS | ALGORITHMS | DATASETS | | | | | | | | | |
|---|---|---|---|---|---|---|---|---|---|---|---|
| | | ADULT | | COMPAS | | DEFAULT CREDIT | | GERMAN CREDIT | | HELOC | |
| | | eff | avc | eff | avc | eff | avc | eff | avc | eff | avc |
| DNN | FAST ARES | 12.39 ± 1.06 | 1.0 ± 0.0 | 55.0 ± 0.86 | 1.21 ± 0.09 | 18.88 ± 2.16 | 1.0 ± 0.0 | 52.39 ± 1.63 | 1.0 ± 0.0 | 12.19 ± 0.58 | 1.03 ± 0.05 |
| | CET | NAN ± NAN | NAN ± NAN | 63.62 ± 10.35 | 0.96 ± 0.24 | 98.87 ± 0.62 | 6.32 ± 2.28 | 97.3 ± 2.46 | 1.58 ± 0.54 | 86.78 ± 10.62 | 8.67 ± 3.25 |
| | T-GLANCE | 100.0 ± 0.0 | 4.43 ± 0.43 | 99.53 ± 0.22 | 2.62 ± 0.24 | 100.0 ± 0.0 | 1.48 ± 0.43 | 96.97 ± 3.48 | 1.59 ± 0.8 | 99.8 ± 0.24 | 10.9 ± 1.22 |
| | Group-CF | 100.0 ± 0.0 | 10.08 ± 0.03 | 100.0 ± 0.0 | 4.48 ± 2.53 | 79.6 ± 20.79 | 1.53 ± 0.62 | 97.8 ± 4.4 | 1.85 ± 0.13 | 80.4 ± 10.17 | 3.09 ± 0.91 |
| | GLOBE-CE | 99.92 ± 0.0 | 4.24 ± 0.42 | 100.0 ± 0.0 | 4.54 ± 3.31 | 76.94 ± 37.55 | 5.14 ± 0.35 | 93.31 ± 3.48 | 2.0 ± 1.55 | 42.72 ± 46.97 | 11.77 ± 15.87 |
| | dGLOBE-CE | 99.92 ± 0.0 | 10.89 ± 1.37 | 100.0 ± 0.0 | 7.96 ± 3.91 | 87.38 ± 18.69 | 5.96 ± 4.14 | 97.36 ± 0.82 | 2.49 ± 0.27 | 99.96 ± 0.05 | 11.07 ± 8.6 |
| | C-GLANCE | 100.0 ± 0.0 | 4.6 ± 0.73 | 100.0 ± 0.0 | 2.34 ± 0.43 | 100.0 ± 0.0 | 1.2 ± 0.4 | 95.31 ± 3.15 | 1.25 ± 0.33 | 99.94 ± 0.05 | 11.24 ± 1.37 |
| LR | FAST ARES | 11.74 ± 2.4 | 1.0 ± 0.0 | 62.5 ± 1.82 | 1.24 ± 0.14 | 10.85 ± 5.45 | 1.07 ± 0.13 | 75.27 ± 2.96 | 1.0 ± 0.0 | 9.23 ± 1.24 | 1.12 ± 0.1 |
| | CET | NAN ± NAN | NAN ± NAN | 73.18 ± 4.34 | 1.24 ± 0.15 | 96.5 ± 2.85 | 3.79 ± 1.31 | 2.42 ± 0.24 | 2.42 ± 0.24 | 100.0 ± 0.0 | 3.57 ± 1.48 |
| | T-GLANCE | 100.0 ± 0.0 | 0.69 ± 0.01 | 99.79 ± 0.22 | 2.19 ± 0.1 | 100.0 ± 0.0 | 1.43 ± 0.36 | 99.58 ± 0.83 | 1.54 ± 0.32 | 100.0 ± 0.0 | 1.58 ± 0.37 |
| | GROUPCF | 100.0 ± 0.0 | 1.71 ± 0.39 | 100.0 ± 0.0 | 3.97 ± 2.38 | 95.4 ± 9.2 | 1.94 ± 1.2 | 97.6 ± 2.94 | 9.34 ± 3.85 | 90.6 ± 3.93 | 2.4 ± 1.38 |
| | GLOBE-CE | 99.92 ± 0.0 | 2.68 ± 0.17 | 95.74 ± 8.52 | 5.14 ± 3.77 | 99.94 ± 0.07 | 3.42 ± 1.99 | 57.09 ± 20.03 | 0.75 ± 1.04 | 99.9 ± 0.0 | 0.6 ± 0.54 |
| | dGLOBE-CE | 99.92 ± 0.0 | 5.91 ± 0.93 | 100.0 ± 0.0 | 6.71 ± 0.23 | 99.94 ± 0.07 | 10.38 ± 7.76 | 69.89 ± 15.35 | 2.47 ± 0.23 | 99.9 ± 0.0 | 1.63 ± 0.35 |
| | C-GLANCE | 100.0 ± 0.0 | 1.04 ± 0.07 | 100.0 ± 0.0 | 2.33 ± 0.38 | 100.0 ± 0.0 | 1.05 ± 0.11 | 100.0 ± 0.0 | 1.21 ± 0.06 | 100.0 ± 0.0 | 1.55 ± 0.54 |
| XGB | FAST ARES | 6.13 ± 0.42 | 1.0 ± 0.0 | 59.83 ± 3.12 | 1.1 ± 0.05 | 31.86 ± 5.12 | 1.05 ± 0.04 | 51.27 ± 1.57 | 1.0 ± 0.0 | 8.49 ± 1.32 | 1.16 ± 0.13 |
| | CET | NAN ± NAN | NAN ± NAN | 58.4 ± 9.3 | 1.06 ± 0.24 | 86.29 ± 9.94 | 4.5 ± 2.64 | 99.46 ± 1.08 | 2.73 ± 0.49 | 86.78 ± 6.7 | 12.51 ± 2.75 |
| | T-GLANCE | 99.86 ± 0.14 | 1.8 ± 0.51 | 99.02 ± 0.93 | 2.6 ± 0.46 | 94.6 ± 3.13 | 2.75 ± 1.42 | 99.46 ± 1.08 | 1.33 ± 0.42 | 96.42 ± 2.45 | 24.85 ± 8.4 |
| | GROUPCF | 96.8 ± 1.72 | 1.41 ± 0.54 | 100.0 ± 0.0 | 4.06 ± 2.1 | 95.2 ± 1.6 | 1.41 ± 0.64 | 100.0 ± 0.0 | 5.78 ± 4.11 | 78.4 ± 5.82 | 5.63 ± 1.93 |
| | GLOBE-CE | 82.87 ± 12.14 | 30.1 ± 10.39 | 87.13 ± 11.14 | 9.75 ± 7.2 | 82.7 ± 7.26 | 20.82 ± 1.73 | 77.05 ± 11.26 | 1.14 ± 1.24 | 27.66 ± 5.06 | 12.52 ± 32.48 |
| | dGLOBE-CE | 93.76 ± 1.98 | 64.76 ± 1.29 | 99.84 ± 0.31 | 12.46 ± 3.42 | 97.47 ± 0.82 | 42.58 ± 3.57 | 86.96 ± 9.79 | 2.66 ± 0.77 | 77.64 ± 11.51 | 128.0 ± 0.0 |
| | C-GLANCE | 99.85 ± 0.12 | 5.98 ± 4.22 | 99.51 ± 0.46 | 2.96 ± 0.82 | 98.13 ± 1.05 | 3.68 ± 1.64 | 100.0 ± 0.0 | 1.06 ± 0.03 | 98.94 ± 0.66 | 19.99 ± 1.91 |

Appendix H presents an extensive experimental evaluation, including: additional and more detailed results for $s = 4$; results for $s = 8$ for all methods and results under no size constraints for `GLOBE-CE`; results utilizing different counterfactual generation methods.

**Pareto Dominance.** We summarize method performance by determining whether one solution dominates another based on effectiveness and cost. Specifically, *a solution $\mathbb{C}$ of $s$-GCE Pareto dominates another solution $\mathbb{C}'$ if it offers equal or better effectiveness and cost, and is strictly better in at least one of these objectives*. As shown in Table 2, `C-GLANCE` dominates other methods in 39 out of 72 cases (54%), while is dominated only once by `dGLOBE-CE` (in HELOC-DNN—cf. Table 1). `T-GLANCE` dominates other methods in 30 out of 72 cases (41%), while it is dominated once by `Group-CF` (in Default Credit-XGB) and once by `CET` (in German Credit-DNN). In the three cases in which our method is dominated, the performance in terms of effectiveness is comparable. Overall, the `GLANCE` methods dominate other solutions in almost half of the cases (48%) and are dominated in only 2%.

**Solution Practicality.** In the prior Pareto-dominance evaluation (Table 2) we compare solutions with optimal or near-optimal effectiveness to many solutions that exhibit unacceptable or unsatisfactory effectiveness. These lower-performing solutions are impractical for GCE, as the goal is to offer recourse to a large population segment. Solutions with low effectiveness fail to meet this goal, limiting their applicability in real-world scenarios. A solution that leaves a significant percentage of individuals without recourse undermines the very purpose of GCE, as noted by Ley et al. (2023). It is also important to note that achieving low recourse costs is easier for smaller subpopulations, especially those near the decision boundary, which explains the lower domination number when

Table 2: Pareto domination evaluation of solutions, for $s$-GCE problem with $s = 4$. The table reports the rate (number of times over available comparisons) at which `GLANCE` methods dominate competitors (listed in the DOMINATES column) and the rate at which `GLANCE` methods are dominated by competitors (listed in the IS DOMINATED column).

| | C-GLANCE (IMPLICIT) | | T-GLANCE (EXPLICIT) | |
|---|---|---|---|---|
| | DOMINATES | IS DOMINATED | DOMINATES | IS DOMINATED |
| FAST ARES (EXPLICIT) | 1/15 | 0/15 | 1/15 | 0/15 |
| CET (EXPLICIT) | 6/12 | 0/12 | 5/12 | 1/12 |
| GROUPCF (IMPLICIT) | 9/15 | 0/15 | 6/15 | 1/15 |
| GLOBE-CE (IMPLICIT) | 11/15 | 0/15 | 9/15 | 0/15 |
| DGLOBE-CE (IMPLICIT) | 12/15 | 1/15 | 9/15 | 0/15 |
| COMPETITORS (EXPLICIT) | | | 6/27 | 1/27 |
| COMPETITORS (IMPLICIT) | 32/45 | 1/45 | | |
| COMPETITORS (ALL) | 39/72 | 1/72 | 30/72 | 2/72 |

Table 3: Pareto domination evaluation of solutions, for $s$-GCE problem with $s = 4$, after applying the eff $> 80\%$ threshold. The table reports the rate (number of times over available comparisons) at which GLANCE methods dominate competitors (listed in the DOMINATES column) and the rate at which GLANCE methods are dominated by competitors (listed in the IS DOMINATED column).

| | C-GLANCE (IMPLICIT) | | T-GLANCE (EXPLICIT) | |
| --- | --- | --- | --- | --- |
| | DOMINATES | IS DOMINATED | DOMINATES | IS DOMINATED |
| FAST AReS (EXPLICIT) | 0/0 | 0/0 | 0/0 | 0/0 |
| CET (EXPLICIT) | 6/9 | 0/9 | 5/9 | 1/9 |
| GROUPCF (IMPLICIT) | 8/13 | 0/13 | 5/13 | 1/13 |
| GLOBE-CE (IMPLICIT) | 8/10 | 0/10 | 7/10 | 0/10 |
| DGLOBE-CE (IMPLICIT) | 10/13 | 1/13 | 7/13 | 0/13 |
| COMPETITORS (EXPLICIT) | | | 5/9 | 1/9 |
| COMPETITORS (IMPLICIT) | 26/36 | 1/36 | | |
| COMPETITORS (ALL) | 32/45 | 1/45 | 24/45 | 2/45 |

comparing with explicit methods. We consider a solution to be practical if it achieves effectiveness of at least 80%. In Table 1, we highlight the impractical solutions in red. GLANCE methods never return impractical solutions. In contrast, all the solutions returned by Fast AReS are deemed impractical, while the other methods return 2–5 impractical solutions each. Table 3 summarizes the head-to-head comparisons only when considering practical solutions. GLANCE framework dominates other methods in a much larger percentage—over 62% of the cases. It remains dominated in the same three cases, maintaining the percentage of dominated cases at a low 3%.

**Explicit-Implicit Evaluation**. When evaluating our methods within their respective explicit or implicit categories, C-GLANCE dominates other implicit methods in 71% of cases (32 out of 45), compared to a 54% domination rate across all methods. When focusing solely on practical solutions, this rate increases to 72% (26 out of 36). For T-GLANCE, the domination rate against explicit methods is 22% (6 out of 27), lower than the 48% achieved when considering all methods. This drop is attributed to the high number of impractical solutions provided by competing explicit methods. However, after removing impractical solutions, T-GLANCE dominates 56% of cases, slightly higher than the 53% in head-to-head comparisons across all methods. Overall, the GLANCE framework demonstrates strong performance, dominating 52% of cases within categories (69% when considering only practical solutions) while keeping the percentage of dominated cases at a low 4% (3% when considering only practical solutions).

**Robustness.** We expect methods to be robust, *consistently* generating highly effective and low-cost GCEs across different data splits, which is crucial for real-world deployment. Without robustness, recourse actions can vary significantly, undermining trust and leading to unfair outcomes, especially in critical areas like healthcare or finance. Evaluating the stability of effectiveness and cost metrics across different folds is key to determining the practical applicability of a counterfactual explanation method. Standard deviation measures this stability. An effectiveness deviation above 5% indicates an inconsistency in providing recourse, while a cost deviation exceeding half the average suggests unpredictable, impractical actions. These fluctuations make a solution unreliable and we highlight them in red in Table 1. Figure 3 visualizes Table's 1 results, i.e., the effectiveness of each solution vs. the cost (scaled to 0–1 values), along with their standard deviation. This figure provides a clear comparison of the trade-offs between cost and effectiveness, underscoring the robustness and consistency of the methods across different scenarios.

## 6 CONCLUSION

This paper introduces GLANCE, a robust and flexible framework for generating global counterfactual explanations that effectively balances the trade-offs between effectiveness, cost, and interpretability. An extensive experimental evaluation demonstrates that GLANCE constructs counterfactual explanations that are more effective and cheaper than those produced by the state of the art under size constraints.

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

## A  HARDNESS

To analyze the complexity of the $s$-GCE problem, we suppose that the set of all actions is finite by discretizing continuous features.

**Theorem 3** (NP-hardness). *The $s$-GCE problem is NP-hard.*

*Proof.* We establish the NP-hardness of the $s$-GCE problem by reducing the well-known *Max $k$-Coverage* problem to it.

The *Max $k$-Coverage* problem is defined as follows: Given a universe $\mathcal{U}$ of elements and a collection of subsets $\mathcal{S} = \{S_1, S_2, \ldots, S_n\}$ where each $S_i \subseteq \mathcal{U}$, along with an integer $k$, the goal is to select at most $k$ subsets from $\mathcal{S}$ such that the total number of covered elements is maximized.

To perform the reduction, we proceed by mapping each subset $S_i$ to an action $a_i$ in the $s$-GCE problem:

- **Actions and Subsets Correspondence:** Each action $a_i$ corresponds to a subset $S_i$ of elements in $\mathcal{U}$.

- **Coverage Mapping:** The effectiveness of an action $a_i$ in the $s$-GCE problem is defined as the elements in $S_i$.

- **Ignoring Costs:** We ignore the cost associated with actions in the $s$-GCE problem, focusing solely on coverage.

Under this mapping, the objective of selecting at most $k$ actions in the $s$-GCE problem to maximize the coverage of elements aligns exactly with the objective of the Max $k$-Coverage problem.

$\square$

Given that even the relaxed version of the $s$-GCE problem that ignores costs is NP-hard, identifying the optimal solution for the $s$-GCE is computationally infeasible. Moreover, the action space, although finite, may be too large to process efficiently. Therefore, our approach involves generating a manageable subset $\mathbb{A}' \subseteq \mathbb{A}$ of actions before extracting the final $s$ actions. In the next section, we will introduce C-GLANCE, an efficient heuristic designed to solve the $s$-GCE problem.

## B  SIZE-COST AND SIZE-EFFECTIVENESS TRADE-OFFS

Let $\mathbb{C}$ be a set of actions. Let a new action $a$ and the $\mathbb{C}' = \mathbb{C} \cup \{a\}$. We analyze how increasing the size (size) of the solution, impacts the effectiveness (eff) and the average cost (avc).

Case 1 **The new action does not flip any new instances.**
   In this scenario, the effectiveness remains the same.
- If the new action is cheaper than the current actions:
 Some instances already receiving recourse from the current actions may now be addressed by the cheaper new action, reducing the average cost.
- If the new action is costlier than the current actions:
 The current actions will continue to provide recourse for the instances they already cover, leaving the average cost unchanged.

Case 2 **The new action flips new instances.**
   In this scenario, the effectiveness increases, as more instances now receive recourse.
- If the cost of the new action is lower than the current average cost:
 The new action provides recourse for additional instances at a lower cost, thereby reducing the average cost.
- If the cost of the new action is higher than the current average cost:
 The inclusion of the new action raises the average cost, as the costlier new instances contribute positively to the total, increasing the average cost.

This analysis highlights the trade-offs between size, effectiveness, and cost, emphasizing the importance of carefully selecting new actions to achieve a desirable balance. Notably, achieving near-optimal effectiveness (close to 100%) often requires adding actions that address outliers, which may have higher costs. However, one of the strengths of our method is its ability to achieve near-optimal effectiveness with relatively low average costs across most datasets.

## C    TIME COMPLEXITY ANALYSIS

Let $n$ be the number of instances, $k$ the initial number of clusters, $s$ the number of global counterfactual actions, $m$ the number of candidate actions to generate, and $d$ the number of features. Let $\mathcal{T}_{CF}(d, \text{model})$ be the time of generating a single candidate counterfactual action and $\mathcal{T}_{\text{model}}$ be the prediction time of the black-box model.

### C.1    C-GLANCE

Let's assume that $k$-means is used for the clustering and Iterations are the number of iterations required for convergence. Then `C-GLANCE` time complexity is:

- **Clustering (K-means):** $\mathcal{O}(n \cdot k \cdot d \cdot \text{Iterations})$
- **Action Generation:** $\mathcal{O}(k \cdot m \cdot \mathcal{T}_{CF}(d, \text{model}))$
- **Merging Clusters:** $\mathcal{O}\Big((k - s) \cdot k^2 \cdot d + (k - s) \cdot \big(n \cdot d + m \cdot (k - s) \cdot d\big)\Big)$
    - **Finding pair of closest clusters:**
        * For one iteration: $\mathcal{O}(k^2 \cdot d)$
        * For all iterations: $\mathcal{O}((k - s) \cdot k^2 \cdot d)$
    - **Merging pair of closest clusters:**
        * For one iteration: $\mathcal{O}\big(n \cdot d + m \cdot (k - s) \cdot d\big)$
        * For all iterations: $\mathcal{O}\big((k - s) \cdot n \cdot d + m \cdot (k - s) \cdot d\big)$
- **Evaluation and selection of final actions:** $\mathcal{O}(m \cdot k \cdot n \cdot d + n \cdot \mathcal{T}_{\text{model}})$

After simplifying common terms, the total complexity is:

$$\mathcal{O}\big(n \cdot k \cdot d \cdot \text{Iterations} + k \cdot m \cdot \mathcal{T}_{CF}(d, \text{model}) + d \cdot (k - s) \cdot k^2 + n \cdot k \cdot m \cdot (d + \mathcal{T}_{\text{model}})\big)$$

### C.2    T-GLANCE

Let $d' \leq d$ be the number of the selected features.

- **Initial action assignment (root):** $\mathcal{O}\Big(m\big(\mathcal{T}_{CF}(d, \text{model}) + n \cdot d + n \cdot \mathcal{T}_{\text{model}}\big)\Big)$
- **Tree construction:** $\mathcal{O}\Big(2^{d'} \cdot d' \cdot m \cdot \big(n \cdot (d + \mathcal{T}_{\text{model}}) + \mathcal{T}_{CF}(d, model)\big)\Big)$
    - Search-and-split a single node: $\mathcal{O}\Big(d' \cdot m \cdot \big(n \cdot (d + \mathcal{T}_{\text{model}}) + \mathcal{T}_{CF}(d, model)\big)\Big)$
        * Brief Explanation: For every possible split, generate m actions on both children and evaluate them. Given these, the rest (finding maximum effectiveness action, computing V(A, i)) are much faster.
    - Maximum number of nodes: $\mathcal{O}\Big(2^{d'}\Big)$

The total complexity is dominated of course by the tree construction term and is equal to:

$$\mathcal{O}\Big(2^{d'} \cdot d' \cdot m \cdot \big(n \cdot (d + \mathcal{T}_{\text{model}}) + \mathcal{T}_{CF}(d, model)\big)\Big)$$

# D DATASETS & PREPROCESSING

We use five publicly available datasets as benchmarks. Our choice is based on their established use in previous works. Table 4 summarizes the datasets' information, including the number of instances, the number of categorical and continuous features, the input dimensions (i.e., the number of continuous features plus the number of categorical after a preprocessing step using one-hot-encoding), and the number of instances used for training and testing, evaluated using a 5-fold cross-validation strategy. This approach illustrates the resilience of our method across various splits.

The dataset preprocessing follows the approach in Ley et al. (2023), using the publicly available repository at `https://github.com/danwley/GLOBE-CE/`. For completeness, we also provide a short description of each dataset below.

**Adult** The Adult dataset (also known as "Census Income" dataset) is designed for the task of predicting whether an individual's income exceeds $ 50K/yr, based on census data. The data as well as more detailed information can be obtained at `https://archive.ics.uci.edu/dataset/2/adult`.

For the preprocessing of this dataset, we first drop the "education-num" feature due to redundancy with the "education" feature (which takes on string values). Next, we remove missing values and map the class labels '<=50K' and '>50K' to 0 and 1, respectively. Additional minor transformations, mostly involving data types, are applied and can be reviewed in our source code.

**COMPAS** The COMPAS dataset (Correctional Offender Management Profiling for Alternative Sanctions) Angwin et al. (2016) is available at `https://github.com/propublica/compas-analysis/blob/master/compas-scores-two-years.csv`. Detailed description and information on the dataset can be found at `https://www.propublica.org/article/how-we-analyzed-the-compas-recidivism-algorithm`. It categorizes recidivism risk based on several factors, including race.

For the preprocessing of this dataset, we drop the "days_b_screening_arrest" feature, as it contains missing values. We also turn jail-in and jail-out dates to durations and turn negative durations to 0. Some additional filters are taken from the COMPAS analysis by ProPublica. Finally, the target variable's values are transformed into the canonical 0 for the negative class and 1 for the positive class.

**German Credit** The German Credit dataset Dua & Graff (2019) classifies people described by a set of attributes as good or bad credit risks. A detailed description and the dataset can be found in `https://archive.ics.uci.edu/ml/datasets/statlog+(german+credit+data)`.

The only preprocessing step we performed for this dataset was the transformation of the target variable's values into 0 - 1.

**Default Credit** The Default Credit dataset Yeh & Lien (2009) is designed to classify the risk of default on customer payments, aiming to support the development and assessment of models for predicting creditworthiness and the likelihood of loan default. It can be obtained at `https://archive.ics.uci.edu/ml/datasets/default+of+credit+card+clients`.

To properly work with this dataset, we needed to drop the "ID" feature, since it holds no useful information, and transform the target labels into the canonical 0 - 1 values.

**HELOC** The HELOC (Home Equity Line of Credit) dataset Brown et al. (2018) contains anonymized information about home equity line of credit applications made by real homeowners, classifying credit risk. It is available at `https://community.fico.com/s/explainable-machine-learning-challenge`. All the features on this dataset are numeric.

A substantial percentage of these features' values are missing, so the main preprocessing step we performed here was to remove rows where all values are missing, and then replace all remaining

missing values with the median of the respective feature. Other than that, we only needed to transform target labels to 0-1.

Table 4: Summary of the datasets used in our experiments. Specifically, we list the number of instances, input dimensions (i.e., the number of continuous features plus the number of categorical after the one-hot-encoding preprocessing step performed internally by the models), the number of categorical and continuous features, and the number of instances used for training and testing, evaluated using a 5-fold cross-validation strategy.

| DATASET | NO. INSTANCES | INPUT DIM. | CATEGORICAL | CONTINUOUS | TRAIN | TEST |
|---|---|---|---|---|---|---|
| ADULT | 30161 | 102 | 8 | 5 | 24128 | 6033 |
| COMPAS | 6172 | 15 | 4 | 2 | 4937 | 1235 |
| DEFAULT CREDIT | 30000 | 91 | 9 | 14 | 24000 | 6000 |
| GERMAN CREDIT | 1000 | 71 | 17 | 3 | 800 | 200 |
| HELOC | 9871 | 23 | 0 | 23 | 7896 | 1975 |

## E  MODELS AND HYPERPARAMETERS

In our experimental evaluation, we utilize three distinct models: XGBoost (XGB), Logistic Regression (LR), and Deep Neural Networks (DNNs). Following Ley et al. (2023), we maintain an 80:20 train-test split, but, instead of splitting once, we perform 5-fold cross-validation, using each fold as a test set while training on the remaining four. The distinctive hyperparameters for each model are described in detail in this section. Additionally, we showcase the performance metric (accuracy), with all reported accuracies presented as the mean and standard deviation across the folds, providing a standardized foundation for comparative analysis of our methodologies.

**XGBoost (XGB)**  Implementation from the common `xgboost`[1] library. Hyperparameter values for each dataset and the model's accuracy on the test set are shown in table 5.

Table 5: XGBoost Hyperparameter Configurations.

| DATASET | DEPTH | ESTIMATORS | $\gamma, \alpha, \lambda$ | TEST ACCURACY |
|---|---|---|---|---|
| ADULT | 6 | 100 | 0,0,1 | 86.62% ± 0.18% |
| COMPAS | 4 | 100 | 1,0,1 | 67.95% ± 1.73% |
| DEFAULT CREDIT | 10 | 200 | 2,4,1 | 81.35% ± 0.18% |
| GERMAN CREDIT | 6 | 500 | 0,0,1 | 76.9% ± 0.86% |
| HELOC | 6 | 100 | 4,4,1 | 72.97% ± 1.14% |

**Logistic Regression (LR)**  Implementation from the common `sklearn`[2] library. Hyperparameter values for each dataset and the model's accuracy on the test set are shown in table 6.

Table 6: Logistic Regression Hyperparameter Configurations.

| DATASET | MAX ITER. | CLASS WEIGHTS(0:1) | TEST ACCURACY |
|---|---|---|---|
| ADULT | 100 | 1:1 | 79.09% ± 0.12% |
| COMPAS | 1000 | 1:1 | 66.69% ± 0.97% |
| DEFAULT CREDIT | 1000 | 1:1 | 82.08% ± 0.19% |
| GERMAN CREDIT | 1000 | 1:1 | 74.80% ± 1.91% |
| HELOC | 3000 | 1:1 | 73.06% ± 0.96% |

**Deep Neural Network (DNN)**  Implementation using `pytorch`[3] library. Hyperparameter values for each dataset and the model's accuracy on the test set are shown in table 7.

---

[1]`https://xgboost.readthedocs.io/en/stable/`
[2]`https://scikit-learn.org/stable/modules/generated/sklearn.linear_model.LogisticRegression.html`
[3]`https://pytorch.org/docs/stable/index.html`

Table 7: Deep Neural Network Hyperparameter Configurations.

| DATASET | HIDDEN LAYERS WIDTH | TEST ACCURACY |
|---|---|---|
| ADULT | [64, 32, 16] | $84.13\% \pm 0.30\%$ |
| COMPAS | [64, 32, 16] | $66.98\% \pm 1.27\%$ |
| DEFAULT CREDIT | [64, 32, 16, 8] | $81.37\% \pm 0.13\%$ |
| GERMAN CREDIT | [8, 4] | $71.3\% \pm 2.27\%$ |
| HELOC | [64, 32, 16, 8] | $70.97\% \pm 0.54\%$ |

## F  COMPETING METHODS

**GLOBE-CE.** The `GLOBE-CE` method, introduced by Ley et al. (2023) was executed based on the implementation available on their GitHub (`https://github.com/danwley/GLOBE-CE`).

Table 8: `GLOBE-CE` results without constraining the action set size

| MODELS | DATASETS | | | | | | | | | | | | | | |
|---|---|---|---|---|---|---|---|---|---|---|---|---|---|---|---|
| | ADULT | | | COMPAS | | | DEFAULT CREDIT | | | GERMAN CREDIT | | | HELOC | | |
| | eff | avc | size | eff | avc | size | eff | avc | size | eff | avc | size | eff | avc | size |
| DNN | $100.0 \pm 0.0$ | $1.24 \pm 0.06$ | 322.4 | $100.0 \pm 0.0$ | $1.11 \pm 0.6$ | 186.2 | $98.9 \pm 0.73$ | $1.1 \pm 0.04$ | 18.4 | $95.69 \pm 2.42$ | $1.12 \pm 0.09$ | 4.2 | $84.32 \pm 19.32$ | $2.63 \pm 1.03$ | 224.8 |
| LR | $100.0 \pm 0.0$ | $1.2 \pm 0.03$ | 545.2 | $98.92 \pm 8.52$ | $1.41 \pm 0.19$ | 97.8 | $99.94 \pm 0.07$ | $1.08 \pm 0.01$ | 4.6 | $71.29 \pm 14.27$ | $1.05 \pm 0.09$ | 1.5 | $100.00 \pm 0.0$ | $0.44 \pm 0.06$ | 512.8 |
| XGB | $93.76 \pm 0.91$ | $4.28 \pm 4.46$ | 46.8 | $92.34 \pm 8.87$ | $1.22 \pm 0.27$ | 42.6 | $94.62 \pm 2.85$ | $1.05 \pm 0.28$ | 40.6 | $84.97 \pm 10.46$ | $1.17 \pm 0.15$ | 2.2 | $30.32 \pm 4.07$ | $2.11 \pm 0.47$ | 17 |

Additional results of `GLOBE-CE` can be found in Table 8. Here we do not constrain the set of actions and we observe better performance, especially in terms of cost. However, these solutions do not represent solutions for $s$-GCE problem, since the solution size can be up to an average of 545.2 actions using 5-fold cross-validation (in Adult-LR case).

**dGLOBE-CE.** The `dGLOBE-CE` method, introduced by Ley et al. (2023), was executed based on the implementation available on their GitHub repository (`https://github.com/danwley/GLOBE-CE`).

**AReS.** The `AReS` method, first introduced by Rawal & Lakkaraju (2020), was executed using the enhanced `Fast AReS` version implemented by Ley et al. (2023). The code for this version was also obtained from the same GitHub repository (`https://github.com/danwley/GLOBE-CE`). We used the default hyperparameters provided in this implementation.

**GroupCF.** The `GroupCF` method, introduced by Warren et al. (2023), was implemented using custom code due to some ambiguity in the instructions provided in the original GitHub repository (`https://github.com/e-delaney/group_cfe`), which made the original implementation challenging to follow.

**CET.** The `CET` method, introduced by Kanamori et al. (2022), was executed using the original implementation available on GitHub (`https://github.com/kelicht/cet`). We had to replace the solver they used for the MIP solved at each node of their tree. IBM CPLEX solver is proprietary and we replaced it with the GUROBI solver, for which we had an available license. We used the default hyperparameters provided in this implementation.

## G  CANDIDATE COUNTERFACTUAL ACTION GENERATORS

In this section, we outline the methods used to generate candidate counterfactual explanations and present comparative results employing different generation methods. To demonstrate the modularity of our framework, we developed and utilized various methods, showing that our approach is not tied to a specific counterfactual generation technique. The comparative results of these experiments are shown in Tables 13 and 14. For the results in Table 1 of the main paper and Table 10, DiCE (Mothilal et al., 2020) was used to generate candidate counterfactuals.

### G.1 DICE

Detailed information on its methodology can be found in the corresponding paper by Mothilal et al. (2020), and we encourage interested readers to refer to it for a deeper understanding.

For the scope of this paper, it suffices to say that DiCE offers a number of alternative algorithms for finding local counterfactuals. In our experiments, we use the 'random' algorithm, which selects features to change at random and tests whether the change results in a successful counterfactual.

The other algorithms were either unsuitable for a black-box setting (e.g., requiring gradients), computationally inefficient (e.g., genetic algorithm), or failed to generate cost-efficient counterfactual (e.g., KDTree) during our initial experimentation.

### G.2 RANDOM SAMPLING

We developed a method called 'Random Sampling'. To find counterfactuals for an affected instance, this method iteratively modifies its features one at a time. The process begins by randomly altering one feature at a time, generating multiple new candidate instances. Each modified instance is then evaluated by querying the black-box model to determine whether it qualifies as a valid counterfactual. The method proceeds to modify additional features, building a set of potential counterfactuals.

Key differences between our method and DiCE's random approach include:

1. We focus on modifying only the top $k_f$ most important features (using permutation feature importance), set to 3 in all experiments.

2. For categorical features, only the top $k_c$ most frequent categories among unaffected individuals are considered replacement candidates (set to 10 for all experiments).

3. We also introduce vectorization in certain operations, improving computational efficiency over DiCE's implementation.

### G.3 NEAREST NEIGHBORS

This method is implemented by storing all unaffected individuals in memory. When queried to provide $k$ counterfactuals for an affected individual, it retrieves the $k$ nearest neighbors from the set of unaffected instances based on their proximity to the affected individual. This approach ensures that the generated counterfactuals are valid and closely aligned with the original instance, improving the relevance of the recommendations.

### G.4 NEAREST NEIGHBORS SCALED

This algorithm closely resembles the 'Nearest Neighbors' one but introduces a key enhancement. Rather than returning the $k$ nearest neighbors, it performs a localized search along the multidimensional line connecting the affected individual and any neighbor. We sample points along this line segment, and if any are classified as positive by the model, they are returned instead, as they are closer to the affected instance, potentially offering a more cost-efficient recourse.

## H EXPERIMENTAL RESULTS

### H.1 EXPERIMENTAL PROCEDURE

We evaluate our algorithms by comparing them with the following methods: `Fast AReS` by Ley et al. (2022), `CET` by Kanamori et al. (2022), `GroupCF` by Warren et al. (2023), `GLOBE-CE` and `dGLOBE-CE` by Ley et al. (2023).

All of the methods are constrained to solve $s$-GCE problem, with $s = 4$ and $s = 8$. These constraints are applied differently depending on the method. In `C-GLANCE` and `GroupCF`, we configure the resulting number clusters to match the target sizes. `GLOBE-CE` is constrained by a maximum of 4 or 8 scalars, while `dGLOBE-CE` is limited to 4 or 8 directions, with up to two scalars per direction. Lastly, `T-GLANCE` and `CET` are constrained with a maximum of 4 or 8 leaves.

In the main paper, we present summarized results for $s = 4$ in Table 1, with complementary results for $s = 8$ in Table 10 of the following section. The summarized results include the effectiveness and cost of each solution for all the dataset and model combinations. In Tables 15 to 19, we provide detailed results for each model, adding runtime and the number of actions per dataset and method alongside effectiveness and cost.

### H.2 EXPERIMENTS FOR EFFECTIVENESS-COST TRADE-OFFS

As shown in Table 1, our methods significantly outperform CET and Fast AReS in terms of effectiveness. Fast AReS fails to exceed 80% effectiveness in any case, and CET achieves near-optimal effectiveness in less than half the cases, sometimes failing to find a solution altogether. These low effectiveness levels make fair comparisons challenging, as solving for a small subset of the population is inherently easier.

To address this challenge, we demonstrate in Table 9 that by intentionally lowering our solutions' effectiveness and selecting cost-efficient actions, our approach can still dominate other methods, achieving both greater effectiveness and lower cost. This adaptability underscores the strength of our methodology in maintaining superior performance even when optimizing for different trade-offs.

Our algorithm's default strategy selects optimal solutions based on maximum effectiveness. However, it also supports alternative strategies, such as selecting the lowest-cost solution, the lowest-cost solution above a specified effectiveness threshold, or the most effective solution below a specified cost. For the experiments presented Table 9, we used the "lowest cost above a certain effectiveness threshold" strategy in C-GLANCE, setting thresholds based on the effectiveness levels achieved by other methods. This flexibility highlights the algorithm's ability to align with various evaluation scenarios and user-defined trade-offs.

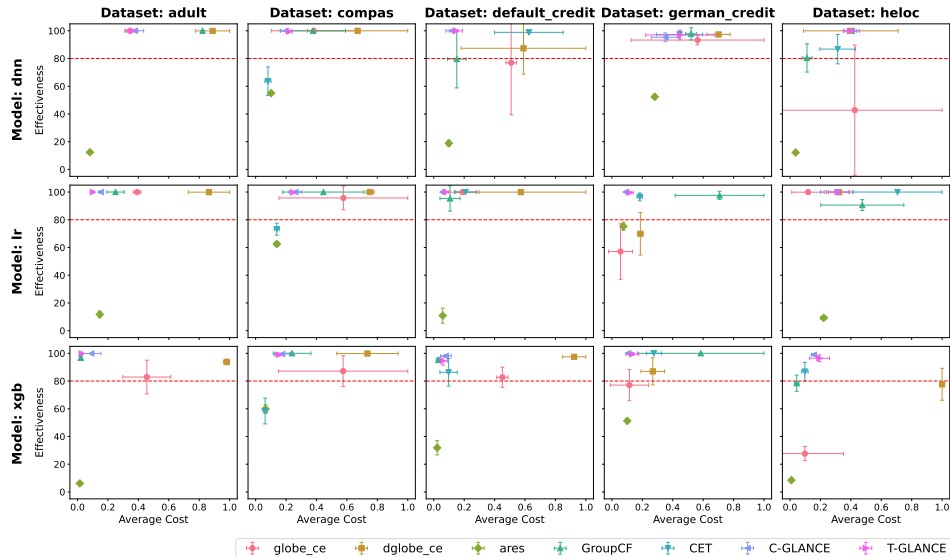

Figure 3: Comparison of effectiveness and average cost, normalized with the maximum cost achieved in each dataset/model combination) for the solution of $s$-GCE with $s = 4$. Standard deviations are represented by error bars. The red horizontal lines represent the eff $> 80\%$ threshold for evaluating the practicality of the solutions.

Table 9: Experiments when C-GLANCE effectiveness is intentionally reduced (provided as threshold) to find dominating solutions toward the lower-effectiveness solutions of other methods.

| MODELS | CSOMPETITORS | DATASETS | | | | | | | | | | | |
| --- | --- | --- | --- | --- | --- | --- | --- | --- | --- | --- | --- | --- | --- |
| | | ADULT | | | | DEFAULT CREDIT | | | | HELOC | | | |
| | | COMPETITOR eff | COMPETITOR avc | C-GLANCE eff | C-GLANCE avc | COMPETITOR eff | COMPETITOR avc | C-GLANCE eff | C-GLANCE avc | COMPETITOR eff | COMPETITOR avc | C-GLANCE eff | C-GLANCE avc |
| DNN | FAST ARES | 12.39 ± 1.06 | 1.0 ± 0.0 | 26.59 ± 10.5 | 0.66 ± 0.23 | 18.88 ± 2.16 | 1.0 ± 0.0 | 21.54 ± 4.11 | 0.3 ± 0.1 | 12.19 ± 0.58 | 1.03 ± 0.05 | 21.32 ± 5.13 | 0.11 ± 0.11 |
| | CET | NaN | NaN | NaN | NaN | NaN | 6.32 ± 2.28 | 99.83 ± 0.1 | 0.82 ± 0.04 | 86.78 ± 10.62 | 8.67 ± 3.25 | 92.67 ± 4.69 | 3.52 ± 1.09 |
| | GROUPCF | 100.0 ± 0.0 | 10.08 ± 0.03 | 100.0 ± 0.0 | 1.73 ± 0.06 | 79.6 ± 20.79 | 1.53 ± 0.62 | 81.44 ± 2.18 | 0.8 ± 0.08 | 80.4 ± 10.17 | 3.09 ± 0.91 | 82.31 ± 1.81 | 3.09 ± 0.9 |
| | GLOBE-CE | 99.92 ± 0.0 | 4.24 ± 0.42 | 99.96 ± 0.01 | 1.71 ± 0.05 | 76.94 ± 37.55 | 5.14 ± 0.35 | 81.44 ± 2.18 | 0.8 ± 0.08 | 42.72 ± 46.97 | 11.77 ± 15.87 | 45.49 ± 2.1 | 1.19 ± 0.71 |
| | dGLOBE-CE | 99.92 ± 0.0 | 10.89 ± 1.37 | 99.96 ± 0.01 | 1.71 ± 0.05 | 87.38 ± 18.69 | 5.96 ± 4.14 | 91.22 ± 3.87 | 0.81 ± 0.07 | 99.96 ± 0.05 | 11.07 ± 8.6 | 100.0 ± 0.0 | 4.22 ± 1.10 |
| LR | FAST ARES | 11.74 ± 2.4 | 1.0 ± 0.0 | 57.91 ± 24.77 | 0.63 ± 0.26 | 10.85 ± 5.45 | 1.07 ± 0.13 | 65.14 ± 15.62 | 0.55 ± 0.11 | 9.23 ± 1.24 | 1.12 ± 0.1 | 71.55 ± 3.72 | 0.43 ± 0.04 |
| | CET | NaN | NaN | NaN | NaN | 100.0 ± 0.0 | 3.79 ± 1.31 | 100.0 ± 0.0 | 0.73 ± 0.07 | 100.0 ± 0.0 | 3.57 ± 1.48 | 100.0 ± 0.0 | 0.72 ± 0.11 |
| | GROUPCF | 100.0 ± 0.0 | 1.71 ± 0.39 | 100.0 ± 0.0 | 0.78 ± 0.06 | 95.4 ± 9.2 | 1.94 ± 1.2 | 99.05 ± 0.38 | 0.71 ± 0.07 | 90.6 ± 3.93 | 2.4 ± 1.38 | 95.18 ± 1.75 | 0.6 ± 0.08 |
| | GLOBE-CE | 99.92 ± 0.0 | 2.68 ± 0.17 | 99.96 ± 0.02 | 0.78 ± 0.06 | 99.94 ± 0.07 | 3.42 ± 1.99 | 100.0 ± 0.0 | 0.73 ± 0.07 | 99.9 ± 0.0 | 0.6 ± 0.54 | 99.92 ± 0.04 | 0.7 ± 0.11 |
| | dGLOBE-CE | 99.92 ± 0.0 | 5.91 ± 0.93 | 99.96 ± 0.02 | 0.78 ± 0.06 | 99.94 ± 0.07 | 10.38 ± 7.76 | 100.0 ± 0.0 | 0.73 ± 0.07 | 99.9 ± 0.0 | 1.63 ± 0.35 | 99.92 ± 0.04 | 0.7 ± 0.11 |
| XGB | FAST ARES | 6.13 ± 0.42 | 1.0 ± 0.0 | 31.16 ± 8.02 | 0.66 ± 0.13 | 31.86 ± 5.12 | 1.05 ± 0.04 | 40.03 ± 7.82 | 0.13 ± 0.04 | 8.49 ± 1.32 | 1.16 ± 0.13 | 9.66 ± 1.12 | 0.4 ± 0.05 |
| | CET | NaN | NaN | NaN | NaN | 86.29 ± 9.94 | 4.5 ± 2.64 | 89.31 ± 2.77 | 0.73 ± 0.06 | 86.78 ± 6.7 | 12.51 ± 2.75 | 90.72 ± 3.35 | 11.53 ± 2.94 |
| | GROUPCF | 96.8 ± 1.72 | 1.41 ± 0.54 | 97.16 ± 0.41 | 0.81 ± 0.02 | 95.2 ± 1.6 | 1.41 ± 0.64 | 95.73 ± 0.18 | 1.01 ± 0.05 | 78.4 ± 5.82 | 5.63 ± 1.93 | 82.46 ± 6.55 | 8.45 ± 3.1 |
| | GLOBE-CE | 82.87 ± 12.14 | 30.1 ± 10.39 | 94.04 ± 0.46 | 0.77 ± 0.01 | 82.7 ± 7.26 | 20.82 ± 1.73 | 83.77 ± 0.84 | 0.63 ± 0.08 | 27.66 ± 5.06 | 12.52 ± 32.48 | 28.64 ± 0.99 | 0.85 ± 0.19 |
| | dGLOBE-CE | 93.76 ± 1.98 | 64.76 ± 1.29 | 94.04 ± 0.46 | 0.77 ± 0.01 | 97.47 ± 0.82 | 42.58 ± 3.57 | 97.64 ± 0.12 | 1.37 ± 0.25 | 77.64 ± 11.51 | 128.0 ± 0.0 | 82.46 ± 6.55 | 8.45 ± 3.1 |

## H.3 RESULTS FOR S = 8

Table 10 presents the comparison of our methods, C-GLANCE and T-GLANCE, with other state-of-the-art methods for a maximum of 8 actions. We distinguish the results of the explicit (T-GLANCE, Fast AReS, CET) and implicit (C-GLANCE, Group-CF, GLOBE-CE, and dGLOBE-CE), however each of our methods can be compared to all competitor GCE methods, regardless of being implicit or explicit. The results indicate that C-GLANCE and T-GLANCE demonstrate superior performance in most cases, achieving nearly 100% effectiveness with minimal and stable costs across all datasets and models. Exceptions are noted in the HELOC DNN and XGB experiments, where all methods report higher costs due to the dataset's numeric-only features. In contrast, Fast AReS struggles significantly, particularly on the Adult dataset, showing effectiveness as low as 13.8% for DNN, 12.51% for LR, and 7.19% for XGB, explaining its lower costs. Other methods, such as GroupCF, CET, GLOBE-CE, and dGLOBE-CE, generally perform well in terms of effectiveness but in almost all cases incur higher costs compared to C-GLANCE and T-GLANCE.

Table 10: Evaluating the effectiveness and average cost of T-GLANCE against explicit (Fast AReS, CET) and C-GLANCE against implicit (Group-CF, GLOBE-CE, and dGLOBE-CE) GCE methods for $s$-GCE problem with $s = 8$. GCE solutions with effectiveness below 80% are highlighted in red. Non-robust GCEs, identified by either a standard deviation (std) in effectiveness greater than 5% across folds or a std in cost greater than half the average cost, are highlighted in blue.

| MODELS | ALGORITHMS | DATASETS | | | | | | | | | |
| --- | --- | --- | --- | --- | --- | --- | --- | --- | --- | --- | --- |
| | | ADULT | | COMPAS | | DEFAULT CREDIT | | GERMAN CREDIT | | HELOC | |
| | | eff | avc | eff | avc | eff | avc | eff | avc | eff | avc |
| DNN | FAST ARES | 13.8 ± 0.92 | 1.0 ± 0.0 | 63.02 ± 1.58 | 1.11 ± 0.04 | 23.75 ± 1.81 | 1.02 ± 0.03 | 65.47 ± 2.2 | 1.0 ± 0.0 | 15.44 ± 1.91 | 1.05 ± 0.06 |
| | CET | NaN ± NaN | NaN ± NaN | 74.66 ± 5.33 | 1.02 ± 0.07 | 98.53 ± 2.11 | 3.68 ± 1.46 | 100.0 ± 0.0 | 1.27 ± 0.37 | 99.36 ± 0.92 | 8.86 ± 2.99 |
| | T-GLANCE | 100.0 ± 0.0 | 3.77 ± 0.27 | 99.65 ± 0.15 | 2.18 ± 0.15 | 100.0 ± 0.0 | 1.25 ± 0.38 | 96.65 ± 4.08 | 0.94 ± 0.3 | 99.78 ± 0.21 | 9.92 ± 1.83 |
| | GROUPCF | 100.0 ± 0.0 | 11.54 ± 2.98 | 100.0 ± 0.0 | 5.65 ± 3.14 | 84.8 ± 13.23 | 2.15 ± 0.98 | 99.6 ± 0.8 | 2.49 ± 0.82 | 84.8 ± 12.32 | 1.98 ± 0.5 |
| | GLOBE-CE | 100.0 ± 0.0 | 2.54 ± 0.14 | 100.0 ± 0.0 | 4.31 ± 2.84 | 93.25 ± 11.26 | 2.73 ± 0.85 | 95.37 ± 2.9 | 1.89 ± 0.38 | 45.49 ± 44.72 | 14.19 ± 14.14 |
| | DGLOBE-CE | 100.0 ± 0.0 | 11.85 ± 1.53 | 100.0 ± 0.0 | 7.28 ± 1.25 | 99.95 ± 0.04 | 6.77 ± 3.67 | 99.09 ± 1.72 | 2.34 ± 0.17 | 100.0 ± 0.0 | 6.65 ± 0.15 |
| | C-GLANCE | 100.0 ± 0.0 | 3.96 ± 0.56 | 100.0 ± 0.0 | 1.77 ± 0.16 | 100.0 ± 0.0 | 1.2 ± 0.4 | 96.43 ± 3.6 | 0.86 ± 0.18 | 99.98 ± 0.04 | 9.42 ± 2.22 |
| LR | FAST ARES | 12.51 ± 2.44 | 1.0 ± 0.0 | 68.19 ± 0.6 | 1.03 ± 0.01 | 11.67 ± 6.18 | 1.04 ± 0.08 | 71.9 ± 2.65 | 1.0 ± 0.0 | 10.95 ± 1.73 | 1.11 ± 0.07 |
| | CET | NaN ± NaN | NaN ± NaN | 82.51 ± 2.97 | 1.51 ± 0.23 | 100.0 ± 0.0 | 2.0 ± 1.36 | 98.75 ± 1.67 | 2.02 ± 0.33 | 100.0 ± 0.0 | 2.99 ± 1.3 |
| | T-GLANCE | 100.0 ± 0.0 | 0.79 ± 0.16 | 99.71 ± 0.16 | 1.94 ± 0.11 | 100.0 ± 0.0 | 1.13 ± 0.18 | 99.55 ± 0.91 | 1.34 ± 0.22 | 100.0 ± 0.0 | 1.52 ± 0.32 |
| | GROUPCF | 100.0 ± 0.0 | 2.33 ± 1.91 | 100.0 ± 0.0 | 4.67 ± 2.29 | 100.0 ± 0.0 | 3.98 ± 2.86 | 98.9 ± 1.65 | 10.56 ± 2.43 | 96.6 ± 1.96 | 1.98 ± 0.5 |
| | GLOBE-CE | 100.0 ± 0.0 | 1.67 ± 0.11 | 97.74 ± 8.52 | 5.22 ± 3.3 | 99.94 ± 0.07 | 2.24 ± 0.98 | 58.42 ± 18.45 | 2.25 ± 0.33 | 99.9 ± 0.0 | 0.58 ± 0.5 |
| | DGLOBE-CE | 100.0 ± 0.0 | 6.12 ± 0.64 | 100.0 ± 0.0 | 6.71 ± 0.23 | 99.94 ± 0.07 | 10.38 ± 7.76 | 69.89 ± 15.35 | 2.47 ± 0.23 | 99.9 ± 0.0 | 1.63 ± 0.35 |
| | C-GLANCE | 100.0 ± 0.0 | 1.03 ± 0.07 | 100.0 ± 0.0 | 1.69 ± 0.06 | 100.0 ± 0.0 | 1.05 ± 0.11 | 100.0 ± 0.0 | 1.18 ± 0.05 | 100.0 ± 0.0 | 1.2 ± 0.12 |
| XGB | FAST ARES | 7.19 ± 0.3 | 1.0 ± 0.0 | 66.07 ± 2.5 | 1.14 ± 0.1 | 36.88 ± 5.46 | 1.07 ± 0.24 | 68.88 ± 1.27 | 1.0 ± 0.0 | 11.22 ± 2.05 | 1.09 ± 0.09 |
| | CET | NaN ± NaN | NaN ± NaN | 68.62 ± 15.67 | 1.3 ± 0.52 | 90.01 ± 10.44 | 2.61 ± 1.06 | 100.0 ± 0.0 | 2.18 ± 0.37 | 90.25 ± 2.51 | 12.99 ± 3.49 |
| | T-GLANCE | 99.94 ± 0.04 | 1.65 ± 0.39 | 99.39 ± 0.26 | 2.41 ± 0.4 | 94.1 ± 3.57 | 1.85 ± 0.16 | 99.46 ± 1.08 | 1.07 ± 0.16 | 98.25 ± 1.85 | 18.75 ± 4.92 |
| | GROUPCF | 99.4 ± 1.2 | 1.88 ± 1.36 | 89.17 ± 11.09 | 9.78 ± 6.35 | 97.8 ± 0.75 | 1.92 ± 1.33 | 80.25 ± 9.62 | 4.05 ± 2.55 | 81.4 ± 4.22 | 5.14 ± 1.46 |
| | GLOBE-CE | 85.76 ± 9.11 | 13.48 ± 4.71 | 89.17 ± 11.09 | 9.78 ± 6.35 | 84.94 ± 6.59 | 11.21 ± 1.29 | 80.25 ± 9.62 | 2.51 ± 0.33 | 28.88 ± 4.71 | 12.51 ± 30.88 |
| | DGLOBE-CE | 94.38 ± 2.01 | 62.37 ± 3.07 | 99.96 ± 0.08 | 9.17 ± 2.01 | 99.08 ± 0.66 | 28.73 ± 9.8 | 88.17 ± 8.2 | 2.64 ± 0.78 | 81.95 ± 10.38 | 128.0 ± 0.0 |
| | C-GLANCE | 99.87 ± 0.09 | 3.85 ± 2.33 | 99.83 ± 0.24 | 2.06 ± 0.46 | 98.87 ± 0.59 | 2.14 ± 0.18 | 100.0 ± 0.0 | 1.02 ± 0.04 | 99.0 ± 0.55 | 16.65 ± 2.44 |

We repeat the experimental evaluation utilizing Pareto-dominance, now for $s = 8$. As shown in Table 11 C-GLANCE dominates other methods in 39 out of 72 cases (54%) while is dominated once by GLOBE-CE (in Adult-DNN—cf. Table 10) and once by dGLOBE-CE (in Heloc-DNN). T-GLANCE dominates other methods in 34 out of 72 comparisons, and is dominated one by Group-CF (in Default Credit-XGB) and once by CET (in German Credit-DNN). Overall, the GLANCE methods dominate other solutions in almost half of the cases (48%) and are dominated in only 3%.

Table 12 summarizes the head-to-head comparisons only when considering practical solutions. GLANCE framework dominates other methods in a much larger percentage—over 62% of the cases.

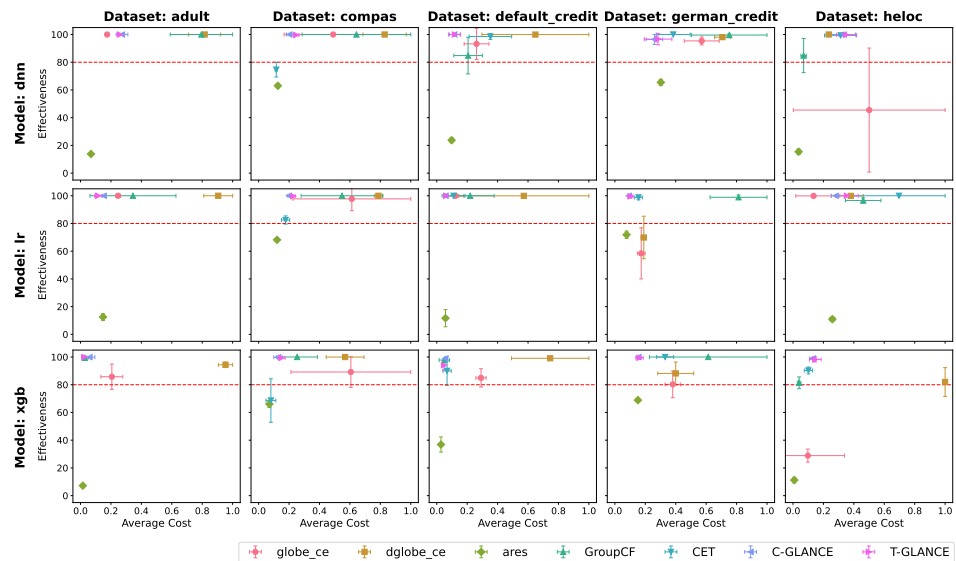

Figure 4: Comparison of effectiveness and average cost, normalized with the maximum cost achieved in each dataset/model combination) for the solution of $s$-GCE with $s = 8$. Standard deviations are represented by error bars. The red horizontal lines represent the eff $> 80\%$ threshold for evaluating the practicality of the solution.

Table 11: Pareto domination evaluation of solutions, for $s$-GCE problem with $s = 8$. This table reports the rate (number of times over available comparisons) at which GLANCE methods dominate competitors (listed in the DOMINATES column) and the rate at which GLANCE methods are dominated by competitors (listed in the IS DOMINATED column).

|  | C-CLANCE (IMPLICIT) | | T-CLANCE (EXPLICIT) | |
|---|---|---|---|---|
|  | DOMINATES | IS DOMINATED | DOMINATES | IS DOMINATED |
| FAST ARES (EXPLICIT) | 1/15 | 0/15 | 2/15 | 0/15 |
| CET (EXPLICIT) | 6/12 | 0/12 | 5/12 | 0/12 |
| GROUPCF (IMPLICIT) | 9/15 | 0/15 | 7/15 | 0/15 |
| GLOBE-CE (IMPLICIT) | 12/15 | 1/15 | 11/15 | 1/15 |
| DGLOBE-CE (IMPLICIT) | 11/15 | 1/15 | 9/15 | 1/15 |
| COMPETITORS (EXPLICIT) |  |  | 7/27 | 0/27 |
| COMPETITORS (IMPLICIT) | 32/45 | 2/45 |  |  |
| COMPETITORS (ALL) | 39/72 | 2/72 | 34/72 | 2/72 |

Table 12: Pareto domination evaluation of solutions, for $s$-GCE problem with $s = 8$, after applying the eff $> 80\%$ threshold. This table reports the rate (number of times over available comparisons) at which GLANCE methods dominate competitors (listed in the DOMINATES column) and the rate at which GLANCE methods are dominated by competitors (listed in the IS DOMINATED column).

|  | C-CLANCE (IMPLICIT) | | T-CLANCE (EXPLICIT) | |
|---|---|---|---|---|
|  | DOMINATES | IS DOMINATED | DOMINATES | IS DOMINATED |
| FAST ARES (EXPLICIT) | 0/0 | 0/0 | 0/0 | 0/0 |
| CET (EXPLICIT) | 6/10 | 0/10 | 5/10 | 0/10 |
| GROUPCF (IMPLICIT) | 9/15 | 0/15 | 7/15 | 0/15 |
| GLOBE-CE (IMPLICIT) | 10/12 | 1/12 | 9/12 | 1/12 |
| DGLOBE-CE (IMPLICIT) | 10/14 | 1/14 | 8/14 | 1/14 |
| COMPETITORS (EXPLICIT) |  |  | 5/10 | 0/10 |
| COMPETITORS (IMPLICIT) | 29/41 | 2/41 |  |  |
| COMPETITORS (ALL) | 35/51 | 2/51 | 29/51 | 2/51 |

It remains dominated in the same four cases, maintaining the percentage of dominated cases at a low 4%.

Finally, when considering only implicit methods, `C-GLANCE` dominates competitor methods in 71% of cases (32 out of 45), compared to a 54% domination rate across all methods. When focusing solely on practical solutions, this rate stays the same at 71 % (29 out of 41). For `T-GLANCE`, the domination rate against explicit methods is 26% (7 out of 27), lower than the 47% achieved when considering all methods. This drop is attributed to the high number of impractical solutions provided by competing explicit methods. However, after removing impractical solutions, `T-GLANCE` dominates 50% of cases, lower than the 57% in head-to-head comparisons across all methods. Overall, the `GLANCE` framework demonstrates strong performance, dominating 54% of cases within categories (67% of cases when considering only practical solutions) while keeping the percentage of dominated cases at a low 2% (4% when considering only practical solutions).

## H.4 COMPERATIVE EVALUATION WITH DIFFERENT COUNTERFACTUAL GENERATION METHODS

Table 13: Evaluation of the effectiveness and cost of `C-GLANCE` and `T-GLANCE`, utilizing different candidate action generation methods for $s$-GCE when $s = 4$. The best metrics are shown in **bold**.

| MODELS | ALGORITHMS | DATASETS | | | | | | | | | |
|---|---|---|---|---|---|---|---|---|---|---|---|
| | | ADULT | | COMPAS | | DEFAULT CREDIT | | GERMAN CREDIT | | HELOC | |
| | | eff | avc | eff | avc | eff | avc | eff | avc | eff | avc |
| DNN | C-GLANCE - DICE | 100.0 ± 0.0 | 4.6 ± 0.73 | 100.0 ± 0.0 | 2.34 ± 0.43 | 100.0 ± 0.0 | 1.2 ± 0.4 | 99.46 ± 1.08 | 1.22 ± 0.41 | **99.94 ± 0.05** | 11.24 ± 1.37 |
| | C-GLANCE - NEARESTNEIGHBORS | 100.0 ± 0.0 | 6.04 ± 0.35 | 100.0 ± 0.0 | 2.27 ± 0.31 | 100.0 ± 0.0 | 4.07 ± 0.08 | 100.0 ± 0.0 | 4.9 ± 0.69 | 99.82 ± 0.15 | 20.18 ± 3.28 |
| | C-GLANCE - NEARESTNEIGHBORSSCALED | 99.99 ± 0.03 | 5.04 ± 0.51 | 100.0 ± 0.0 | **1.74 ± 0.38** | 100.0 ± 0.0 | 3.02 ± 0.58 | 100.0 ± 0.0 | 2.1 ± 0.4 | 99.79 ± 0.3 | 18.88 ± 2.11 |
| | C-GLANCE - RANDOMSAMPLING | 98.01 ± 0.66 | **2.03 ± 0.01** | 100.0 ± 0.0 | 2.08 ± 0.21 | 100.0 ± 0.0 | **1.0 ± 0.0** | 99.43 ± 1.14 | **1.0 ± 0.2** | 94.02 ± 4.28 | **2.53 ± 1.12** |
| | T-GLANCE - DICE | 100.0 ± 0.0 | 4.43 ± 0.43 | 99.53 ± 0.22 | 2.62 ± 0.24 | 100.0 ± 0.0 | 1.48 ± 0.43 | 96.97 ± 3.48 | 1.59 ± 0.8 | **99.8 ± 0.24** | 10.9 ± 1.22 |
| | T-GLANCE - NEARESTNEIGHBORS | 99.96 ± 0.08 | 9.4 ± 1.65 | 99.52 ± 0.29 | 2.71 ± 0.19 | 99.95 ± 0.06 | 4.13 ± 0.38 | **100.0 ± 0.0** | 8.54 ± 1.9 | 93.49 ± 2.03 | 17.71 ± 2.06 |
| | T-GLANCE - NEARESTNEIGHBORSSCALED | 99.97 ± 0.04 | 5.92 ± 1.06 | **99.88 ± 0.16** | 2.47 ± 0.35 | 99.73 ± 0.37 | 3.22 ± 0.32 | 100.0 ± 0.0 | 2.35 ± 0.44 | 95.32 ± 1.19 | 18.97 ± 2.08 |
| | T-GLANCE - RANDOMSAMPLING | 97.57 ± 0.36 | **2.91 ± 0.05** | 99.65 ± 0.08 | **2.28 ± 0.4** | 99.97 ± 0.05 | **1.0 ± 0.0** | 99.53 ± 0.93 | **1.06 ± 0.46** | 95.9 ± 1.85 | **6.45 ± 1.31** |
| LR | C-GLANCE - DICE | **100.0 ± 0.0** | 1.04 ± 0.07 | 100.0 ± 0.0 | 2.33 ± 0.38 | 100.0 ± 0.0 | 1.05 ± 0.11 | 100.0 ± 0.0 | **1.21 ± 0.06** | **100.0 ± 0.0** | 1.55 ± 0.54 |
| | C-GLANCE - NEARESTNEIGHBORS | **100.0 ± 0.0** | 3.76 ± 0.96 | 100.0 ± 0.0 | 2.53 ± 0.32 | 100.0 ± 0.0 | 1.78 ± 0.39 | 100.0 ± 0.0 | 4.21 ± 0.56 | 99.94 ± 0.05 | 21.52 ± 1.33 |
| | C-GLANCE - NEARESTNEIGHBORSSCALED | **100.0 ± 0.0** | 3.19 ± 0.51 | 100.0 ± 0.0 | **2.14 ± 0.15** | 100.0 ± 0.0 | 1.11 ± 0.14 | 100.0 ± 0.0 | 2.02 ± 0.27 | 99.94 ± 0.12 | 20.95 ± 3.47 |
| | C-GLANCE - RANDOMSAMPLING | **100.0 ± 0.0** | **0.72 ± 0.12** | 100.0 ± 0.0 | 2.21 ± 0.2 | 99.94 ± 0.12 | **1.0 ± 0.0** | 100.0 ± 0.0 | 1.31 ± 0.15 | **100.0 ± 0.0** | **1.4 ± 0.27** |
| | T-GLANCE - DICE | **100.0 ± 0.0** | **0.69 ± 0.01** | 99.79 ± 0.22 | **2.19 ± 0.1** | 100.0 ± 0.0 | 1.43 ± 0.36 | 99.58 ± 0.83 | 1.54 ± 0.32 | **100.0 ± 0.0** | 1.58 ± 0.37 |
| | T-GLANCE - NEARESTNEIGHBORS | 99.92 ± 0.04 | 3.52 ± 1.37 | 99.79 ± 0.22 | 3.51 ± 0.19 | 100.0 ± 0.0 | 2.98 ± 0.28 | 100.0 ± 0.0 | 9.66 ± 1.55 | 99.31 ± 0.92 | 19.48 ± 2.53 |
| | T-GLANCE - NEARESTNEIGHBORSSCALED | 98.64 ± 1.3 | 2.98 ± 0.88 | **99.83 ± 0.15** | 3.45 ± 0.25 | 100.0 ± 0.0 | 2.1 ± 0.18 | 100.0 ± 0.0 | 2.14 ± 0.26 | 99.54 ± 0.68 | 18.15 ± 2.02 |
| | T-GLANCE - RANDOMSAMPLING | **100.0 ± 0.0** | 0.71 ± 0.12 | 99.71 ± 0.16 | 2.3 ± 0.28 | 99.97 ± 0.06 | **1.04 ± 0.07** | 98.71 ± 1.05 | **1.46 ± 0.25** | **100.0 ± 0.0** | **1.57 ± 0.66** |
| XGB | C-GLANCE - DICE | 99.85 ± 0.12 | 5.98 ± 4.22 | 99.51 ± 0.46 | 2.96 ± 0.82 | 98.13 ± 1.05 | 3.68 ± 1.64 | **100.0 ± 0.0** | **1.05 ± 0.02** | 98.94 ± 0.66 | 19.99 ± 1.91 |
| | C-GLANCE - NEARESTNEIGHBORS | **100.0 ± 0.0** | 4.53 ± 0.56 | 99.6 ± 0.34 | 2.95 ± 0.58 | **100.0 ± 0.0** | 3.8 ± 0.4 | 100.0 ± 0.0 | 3.67 ± 0.58 | 99.51 ± 0.42 | 19.3 ± 2.66 |
| | C-GLANCE - NEARESTNEIGHBORSSCALED | 99.98 ± 0.03 | 4.12 ± 0.31 | 99.44 ± 0.54 | 3.0 ± 0.33 | 100.0 ± 0.0 | 3.09 ± 0.42 | 100.0 ± 0.0 | 1.75 ± 0.2 | **99.68 ± 0.13** | 20.92 ± 3.96 |
| | C-GLANCE - RANDOMSAMPLING | 91.52 ± 4.44 | **2.1 ± 0.19** | **99.88 ± 0.16** | **2.83 ± 0.76** | 91.52 ± 4.44 | **2.25 ± 1.06** | 100.0 ± 0.0 | 1.14 ± 0.07 | 82.75 ± 5.2 | **9.6 ± 1.84** |
| | T-GLANCE - DICE | 99.86 ± 0.14 | **1.8 ± 0.51** | 99.02 ± 0.93 | 2.6 ± 0.46 | 94.6 ± 3.13 | 2.75 ± 1.42 | 99.46 ± 1.08 | 1.33 ± 0.42 | **96.42 ± 2.45** | 24.85 ± 8.4 |
| | T-GLANCE - NEARESTNEIGHBORS | **99.87 ± 0.22** | 9.44 ± 1.71 | 98.99 ± 0.54 | 3.25 ± 0.33 | **100.0 ± 0.0** | 4.35 ± 0.47 | **100.0 ± 0.0** | 7.8 ± 1.26 | 94.94 ± 1.91 | 18.4 ± 1.41 |
| | T-GLANCE - NEARESTNEIGHBORSSCALED | 99.68 ± 0.25 | 9.04 ± 1.75 | **99.24 ± 0.67** | 3.25 ± 0.41 | 100.0 ± 0.0 | 3.93 ± 0.53 | 100.0 ± 0.0 | 2.35 ± 0.36 | 96.07 ± 1.85 | 20.39 ± 1.69 |
| | T-GLANCE - RANDOMSAMPLING | 99.61 ± 0.66 | 2.03 ± 0.3 | 98.67 ± 1.0 | **2.53 ± 0.34** | 87.04 ± 7.36 | **2.18 ± 1.65** | 99.33 ± 1.33 | **1.23 ± 0.21** | 79.63 ± 7.83 | **8.02 ± 1.43** |

Table 14: Evaluation of the effectiveness and cost of `C-GLANCE` and `T-GLANCE`, utilizing different candidate action generation methods for $s$-GCE when $s = 8$. The best metrics are shown in bold.

| MODELS | ALGORITHMS | DATASETS | | | | | | | | | |
|---|---|---|---|---|---|---|---|---|---|---|---|
| | | ADULT | | COMPAS | | DEFAULT CREDIT | | GERMAN CREDIT | | HELOC | |
| | | eff | avc | eff | avc | eff | avc | eff | avc | eff | avc |
| DNN | C-GLANCE - DICE | 100.0 ± 0.0 | 3.96 ± 0.56 | 100.0 ± 0.0 | 1.77 ± 0.16 | 100.0 ± 0.0 | 1.2 ± 0.4 | 99.46 ± 1.08 | 0.78 ± 0.27 | **99.98 ± 0.04** | 9.42 ± 2.22 |
| | C-GLANCE - NEARESTNEIGHBORS | 100.0 ± 0.01 | 4.99 ± 0.62 | 100.0 ± 0.0 | 1.71 ± 0.18 | 100.0 ± 0.0 | 3.84 ± 0.26 | 100.0 ± 0.0 | 3.94 ± 0.39 | 99.75 ± 0.26 | 16.07 ± 1.32 |
| | C-GLANCE - NEARESTNEIGHBORSSCALED | 100.0 ± 0.0 | 4.44 ± 0.15 | 100.0 ± 0.0 | 1.08 ± 0.08 | 100.0 ± 0.0 | 2.5 ± 0.24 | 100.0 ± 0.0 | 1.65 ± 0.14 | 99.71 ± 0.15 | 16.17 ± 1.72 |
| | C-GLANCE - RANDOMSAMPLING | 95.36 ± 3.8 | **2.84 ± 0.8** | 100.0 ± 0.0 | 1.74 ± 0.22 | 99.82 ± 0.29 | **1.07 ± 0.14** | 100.0 ± 0.0 | **0.7 ± 0.18** | 98.55 ± 0.45 | 10.01 ± 0.53 |
| | T-GLANCE - DICE | 100.0 ± 0.0 | 3.77 ± 0.27 | 99.65 ± 0.15 | 2.19 ± 0.15 | 100.0 ± 0.0 | 1.25 ± 0.38 | 96.65 ± 4.08 | 0.94 ± 0.3 | **99.78 ± 0.21** | 9.42 ± 1.83 |
| | T-GLANCE - NEARESTNEIGHBORS | 99.95 ± 0.08 | 7.71 ± 1.11 | 99.79 ± 0.14 | 2.53 ± 0.17 | 100.0 ± 0.0 | 4.18 ± 0.29 | **100.0 ± 0.0** | 7.54 ± 0.97 | 96.48 ± 0.9 | 19.11 ± 1.76 |
| | T-GLANCE - NEARESTNEIGHBORSSCALED | 99.81 ± 0.16 | 7.56 ± 1.74 | **99.88 ± 0.1** | 2.24 ± 0.39 | 99.97 ± 0.05 | 3.47 ± 0.72 | 100.0 ± 0.0 | 2.11 ± 0.57 | 97.48 ± 0.89 | 20.98 ± 1.93 |
| | T-GLANCE - RANDOMSAMPLING | 96.76 ± 0.39 | **2.87 ± 0.06** | 99.58 ± 0.22 | 2.18 ± 0.45 | 100.0 ± 0.0 | **1.16 ± 0.31** | 100.0 ± 0.0 | **0.86 ± 0.38** | 96.76 ± 4.58 | **7.7 ± 2.89** |
| LR | C-GLANCE - DICE | 100.0 ± 0.0 | 1.03 ± 0.07 | 100.0 ± 0.0 | **1.69 ± 0.06** | 100.0 ± 0.0 | 1.05 ± 0.11 | 100.0 ± 0.0 | **1.15 ± 0.03** | 100.0 ± 0.0 | 1.2 ± 0.12 |
| | C-GLANCE - NEARESTNEIGHBORS | 100.0 ± 0.0 | 3.22 ± 0.49 | 100.0 ± 0.0 | 1.9 ± 0.12 | 100.0 ± 0.0 | 1.39 ± 0.11 | 100.0 ± 0.0 | 3.98 ± 0.55 | 99.88 ± 0.14 | 17.3 ± 1.9 |
| | C-GLANCE - NEARESTNEIGHBORSSCALED | 100.0 ± 0.0 | 2.96 ± 0.62 | 100.0 ± 0.0 | 1.81 ± 0.18 | 100.0 ± 0.0 | 1.01 ± 0.0 | 100.0 ± 0.0 | 1.7 ± 0.2 | 99.88 ± 0.15 | 17.39 ± 0.69 |
| | C-GLANCE - RANDOMSAMPLING | 100.0 ± 0.0 | **0.67 ± 0.07** | 100.0 ± 0.0 | 1.78 ± 0.25 | 99.94 ± 0.12 | **1.0 ± 0.0** | 100.0 ± 0.0 | 1.19 ± 0.05 | **100.0 ± 0.0** | **1.19 ± 0.15** |
| | T-GLANCE - DICE | 100.0 ± 0.0 | 0.69 ± 0.01 | 99.71 ± 0.16 | **1.94 ± 0.11** | 100.0 ± 0.0 | 1.13 ± 0.18 | 99.55 ± 0.91 | 1.34 ± 0.22 | **100.0 ± 0.0** | **1.52 ± 0.32** |
| | T-GLANCE - NEARESTNEIGHBORS | 99.92 ± 0.04 | 3.52 ± 1.37 | 99.79 ± 0.22 | 3.51 ± 0.19 | 100.0 ± 0.0 | 2.98 ± 0.28 | 100.0 ± 0.0 | 7.47 ± 2.24 | 99.31 ± 0.92 | 19.48 ± 2.53 |
| | T-GLANCE - NEARESTNEIGHBORSSCALED | 98.64 ± 1.3 | 2.98 ± 0.88 | **99.83 ± 0.15** | 3.45 ± 0.25 | 100.0 ± 0.0 | 2.1 ± 0.18 | 100.0 ± 0.0 | 1.82 ± 0.29 | 99.54 ± 0.68 | 18.15 ± 2.02 |
| | T-GLANCE - RANDOMSAMPLING | 100.0 ± 0.0 | **0.63 ± 0.09** | 99.71 ± 0.16 | 2.46 ± 0.32 | 100.0 ± 0.0 | **1.02 ± 0.02** | 100.0 ± 0.0 | **1.27 ± 0.17** | 100.0 ± 0.0 | 1.62 ± 0.6 |
| XGB | C-GLANCE - DICE | 99.87 ± 0.09 | 3.85 ± 2.33 | 99.83 ± 0.24 | 2.06 ± 0.46 | 98.87 ± 0.59 | 2.14 ± 0.18 | 100.0 ± 0.0 | **0.97 ± 0.08** | 99.0 ± 0.55 | 16.65 ± 2.44 |
| | C-GLANCE - NEARESTNEIGHBORS | 100.0 ± 0.0 | 4.43 ± 0.55 | 99.6 ± 0.34 | **1.8 ± 0.16** | 100.0 ± 0.0 | 3.2 ± 0.45 | 100.0 ± 0.0 | 3.63 ± 0.58 | 99.49 ± 0.47 | 15.76 ± 1.71 |
| | C-GLANCE - NEARESTNEIGHBORSSCALED | 99.98 ± 0.03 | 3.9 ± 0.41 | 99.48 ± 0.57 | 1.88 ± 0.09 | 100.0 ± 0.0 | 2.4 ± 0.31 | 100.0 ± 0.0 | 1.62 ± 0.13 | **99.72 ± 0.17** | 18.58 ± 1.88 |
| | C-GLANCE - RANDOMSAMPLING | 99.76 ± 0.37 | **2.13 ± 0.17** | **99.84 ± 0.24** | 2.06 ± 0.21 | 93.86 ± 2.69 | **1.51 ± 0.52** | 100.0 ± 0.0 | 1.06 ± 0.1 | 84.76 ± 4.77 | **7.11 ± 1.13** |
| | T-GLANCE - DICE | **99.94 ± 0.04** | **1.65 ± 0.39** | **99.39 ± 0.26** | **2.41 ± 0.4** | 94.1 ± 3.57 | **1.85 ± 0.16** | 99.46 ± 1.08 | **1.07 ± 0.16** | **98.25 ± 1.85** | 18.75 ± 4.92 |
| | T-GLANCE - NEARESTNEIGHBORS | 99.87 ± 0.22 | 9.44 ± 1.71 | 98.99 ± 0.54 | 3.25 ± 0.33 | 100.0 ± 0.0 | 4.35 ± 0.47 | 100.0 ± 0.0 | 5.39 ± 1.33 | 94.94 ± 1.91 | 18.4 ± 1.41 |
| | T-GLANCE - NEARESTNEIGHBORSSCALED | 99.68 ± 0.25 | 9.04 ± 1.75 | 99.24 ± 0.67 | 3.25 ± 0.41 | 100.0 ± 0.0 | 3.93 ± 0.53 | 100.0 ± 0.0 | 2.09 ± 0.28 | 96.07 ± 1.85 | 20.39 ± 1.69 |
| | T-GLANCE - RANDOMSAMPLING | 99.67 ± 0.52 | 1.7 ± 0.57 | 98.3 ± 0.83 | 2.61 ± 0.48 | 87.39 ± 5.65 | 2.55 ± 1.29 | 99.33 ± 1.33 | 1.33 ± 0.57 | 79.59 ± 7.41 | **9.37 ± 1.06** |

Tables 13 and 14 present a comprehensive comparison of `C-GLANCE` and `T-GLANCE` utilizing the counterfactual generation methods described in Appendix G, i.e., DiCE, NearestNeighbors, NearestNeighborsScaled, and RandomSampling, for the $s$-GCE problem with $s = 4$ and $s = 8$, respectively. Across all datasets and models, both `C-GLANCE` and `T-GLANCE` consistently exhibit superb

performance in terms of effectiveness, frequently achieving near 100%. Since the performance in terms of effectiveness is comparable, we will focus on the cost our algorithms achieved under the different generation methods. The best performance is observed with the RandomSampling method, achieving the best results in 41 out of 60 cases. DiCE, which is regarded as state-of-the-art, follows, achieves the lowest cost in 15 out of 60 cases. NearestNeighbors and NearestNeighborsScaled achieve the best cost in only 1 out of 60 and 3 out of 60 cases respectively. In general, `C-GLANCE` and `T-GLANCE`, can demonstrate great performance under all action generation methods, maintaining high effectiveness and low costs in almost all cases.

## H.5 DETAILED RESULTS

Table 15: Detailed results for the solution of $s$-GCE ($s = 4$ and $s = 8$) for COMPAS dataset. The table reports effectiveness, cost, size, and runtime, including their standard deviations for each method.

| METHOD / MODEL | DNN | | | | LR | | | | XGB | | | |
|---|---|---|---|---|---|---|---|---|---|---|---|---|
| | eff | avc | size | RUNTIME | eff | avc | size | RUNTIME | eff | avc | size | RUNTIME |
| FAST AReS-4 | 55.0 ± 0.86 | 1.21 ± 0.09 | 4 | 59.26 ± 26.71 | 62.5 ± 1.82 | 1.24 ± 0.14 | 4 | 11.52 ± 5.39 | 59.83 ± 3.12 | 1.1 ± 0.05 | 4 | 12.88 ± 5.5 |
| CET - 4 | 63.62 ± 10.35 | 0.96 ± 0.24 | 7.0 | 1339.42 ± 247.04 | 73.18 ± 4.34 | 1.24 ± 0.15 | 4.0 | 663.22 ± 45.34 | 58.4 ± 9.3 | 1.06 ± 0.24 | 3.0 | 3280.17 ± 834.28 |
| GROUPCF - 4 | 100.0 ± 0.0 | 4.48 ± 2.53 | 8 | 173.24 ± 2.53 | 100.0 ± 0.0 | 3.97 ± 2.38 | 4 | 45.04 ± 2.38 | 100.0 ± 0.0 | 4.06 ± 2.1 | 4 | 36.07 ± 2.1 |
| GLOBE-CE - 4 | 100.0 ± 0.0 | 4.54 ± 3.31 | 3.0 | 1.09 ± 0.06 | 95.74 ± 8.52 | 5.14 ± 3.77 | 2.6 | 0.42 ± 0.01 | 87.13 ± 11.14 | 9.75 ± 7.2 | 1.6 | 1.06 ± 0.01 |
| DGLOBE-CE - 4 | 100.0 ± 0.0 | 7.96 ± 3.91 | 3.4 | 1.55 ± 0.07 | 100.0 ± 0.0 | 6.71 ± 0.23 | 4.0 | 0.61 ± 0.01 | 99.84 ± 0.31 | 12.46 ± 3.42 | 3.4 | 1.53 ± 0.03 |
| C-GLANCE-4 | 100.0 ± 0.0 | 2.34 ± 0.43 | 4.0 | 177.41 ± 0.0 | 100.0 ± 0.0 | 2.33 ± 0.38 | 4.0 | 82.09 ± 0.0 | 99.51 ± 0.46 | 2.96 ± 0.82 | 4.0 | 286.26 ± 0.0 |
| T-GLANCE-4 | 99.53 ± 0.22 | 2.62 ± 0.24 | 3.0 | 72.01 ± 0.0 | 99.79 ± 0.22 | 2.19 ± 0.1 | 3.0 | 50.14 ± 0.0 | 99.02 ± 0.93 | 2.6 ± 0.46 | 3.0 | 156.0 ± 0.0 |
| FAST AReS-8 | 63.02 ± 1.58 | 1.11 ± 0.04 | 8 | 65.84 ± 29.93 | 68.19 ± 0.6 | 1.03 ± 0.01 | 8 | 15.2 ± 6.9 | 66.07 ± 2.5 | 1.14 ± 0.1 | 8 | 17.69 ± 8.05 |
| CET - 8 | 74.66 ± 5.33 | 1.02 ± 0.07 | 8.0 | 656.64 ± 119.51 | 82.51 ± 2.97 | 1.51 ± 0.23 | 6.0 | 442.76 ± 64.47 | 68.62 ± 15.67 | 1.3 ± 0.52 | 6.0 | 1787.18 ± 360.63 |
| GROUPCF - 8 | 100.0 ± 0.0 | 5.65 ± 3.14 | 8 | 210.34 ± 3.68 | 100.0 ± 0.0 | 4.67 ± 2.29 | 8 | 48.53 ± 2.29 | 100.0 ± 0.0 | 4.08 ± 2.15 | 8 | 37.53 ± 2.15 |
| GLOBE-CE - 8 | 100.0 ± 0.0 | 4.31 ± 2.84 | 6.8 | 1.06 ± 0.04 | 97.74 ± 8.52 | 5.22 ± 3.3 | 5.8 | 0.45 ± 0.02 | 89.17 ± 11.09 | 9.78 ± 6.35 | 3.8 | 1.08 ± 0.03 |
| DGLOBE-CE - 8 | 100.0 ± 0.0 | 7.28 ± 1.25 | 6.8 | 2.3 ± 0.09 | 100.0 ± 0.0 | 6.71 ± 0.23 | 8.0 | 0.88 ± 0.01 | 99.96 ± 0.08 | 9.17 ± 2.01 | 5.6 | 2.23 ± 0.03 |
| C-GLANCE-8 | 100.0 ± 0.0 | 1.77 ± 0.16 | 8.0 | 177.35 ± 0.0 | 100.0 ± 0.0 | 1.69 ± 0.06 | 8.0 | 84.11 ± 0.0 | 99.83 ± 0.24 | 2.06 ± 0.46 | 8.0 | 173.1 ± 0.0 |
| T-GLANCE-8 | 99.65 ± 0.15 | 2.18 ± 0.15 | 4.0 | 21.48 ± 0.64 | 99.71 ± 0.16 | 1.94 ± 0.11 | 4.0 | 13.06 ± 0.46 | 99.39 ± 0.26 | 2.41 ± 0.4 | 4.0 | 40.05 ± 4.5 |

Table 15 presents a comparative analysis of all algorithms for the COMPAS dataset, revealing distinct performance patterns. `CET` and Fast `AReS`, both with 4 and 8 actions, show very low costs across all models; however, this is largely attributable to their relatively low effectiveness scores.

Methods like `GLOBE-CE`, `dGLOBE-CE` display significantly higher effectiveness, mostly above 90% across all models, `GroupCF` yields perfect effectiveness of 100%. However, these methods come with higher costs, indicating that increased effectiveness is often tied to a rise in cost.

On the other hand, `C-GLANCE` and `T-GLANCE` methods consistently achieve near-perfect or perfect effectiveness ($\sim 100\%$) across all models while maintaining much lower costs. This balance of high effectiveness and low cost makes `C-GLANCE` and `T-GLANCE` the optimal methods for the COMPAS dataset.

Table 16: Detailed results for the solution of $s$-GCE ($s = 4$ and $s = 8$) for German Credit dataset. The table reports effectiveness, cost, size, and runtime, including their standard deviations for each method.

| METHOD / MODEL | DNN | | | | LR | | | | XGB | | | |
|---|---|---|---|---|---|---|---|---|---|---|---|---|
| | eff | avc | size | RUNTIME | eff | avc | size | RUNTIME | eff | avc | size | RUNTIME |
| FAST AReS-4 | 52.39 ± 1.63 | 1.0 ± 0.0 | 4 | 119.34 ± 54.6 | 75.27 ± 2.96 | 1.0 ± 0.0 | 4 | 47.88 ± 23.06 | 51.27 ± 1.57 | 1.0 ± 0.0 | 4 | 55.01 ± 27.27 |
| CET - 4 | 97.3 ± 2.46 | 1.58 ± 0.54 | 3.0 | 230.36 ± 47.15 | 96.5 ± 2.85 | 2.42 ± 0.24 | 3.0 | 237.17 ± 11.2 | 100.0 ± 0.0 | 2.73 ± 0.49 | 3.0 | 347.41 ± 30.21 |
| GROUPCF - 4 | 97.8 ± 4.4 | 1.85 ± 0.13 | 4 | 25.54 ± 0.13 | 97.6 ± 2.94 | 9.34 ± 3.85 | 4 | 7.91 ± 3.85 | 100.0 ± 0.0 | 5.78 ± 4.11 | 4 | 5.87 ± 4.11 |
| GLOBE-CE - 4 | 93.31 ± 3.48 | 2.0 ± 1.55 | 2.0 | 1.02 ± 0.01 | 57.09 ± 20.03 | 4.67 ± 2.29 | 4.0 | 0.44 ± 0.0 | 77.05 ± 11.26 | 1.14 ± 1.24 | 1.0 | 1.34 ± 0.03 |
| DGLOBE-CE - 4 | 97.36 ± 0.82 | 2.49 ± 0.27 | 4.0 | 1.57 ± 0.01 | 69.89 ± 15.35 | 2.47 ± 0.23 | 3.2 | 0.7 ± 0.01 | 86.96 ± 9.79 | 2.66 ± 0.77 | 3.6 | 2.09 ± 0.04 |
| C-GLANCE-4 | 95.31 ± 3.15 | 1.25 ± 0.33 | 4.0 | 78.45 ± 0.0 | 100.0 ± 0.0 | 1.21 ± 0.06 | 4.0 | 61.58 ± 0.0 | 100.0 ± 0.0 | 1.06 ± 0.03 | 4.0 | 67.36 ± 0.0 |
| T-GLANCE-4 | 96.97 ± 3.48 | 1.59 ± 0.8 | 3.0 | 85.12 ± 0.0 | 99.58 ± 0.83 | 1.54 ± 0.32 | 3.0 | 75.64 ± 0.0 | 99.46 ± 1.08 | 1.33 ± 0.42 | 3.0 | 185.11 ± 0.0 |
| FAST AReS-8 | 65.47 ± 2.2 | 1.0 ± 0.0 | 8 | 415.74 ± 175.82 | 71.9 ± 2.65 | 1.0 ± 0.0 | 8 | 59.8 ± 32.65 | 68.88 ± 1.27 | 1.0 ± 0.0 | 8 | 64.77 ± 29.67 |
| CET - 8 | 100.0 ± 0.0 | 1.27 ± 0.37 | 1.0 | 142.29 ± 27.72 | 98.75 ± 1.67 | 2.02 ± 0.33 | 6.0 | 156.14 ± 15.7 | 100.0 ± 0.0 | 2.18 ± 0.37 | 5.0 | 239.2 ± 25.84 |
| GROUPCF - 8 | 99.6 ± 0.8 | 2.49 ± 0.82 | 8 | 28.93 ± 0.82 | 98.9 ± 1.65 | 10.56 ± 2.43 | 8 | 15.6 ± 2.33 | 100.0 ± 0.0 | 4.05 ± 2.55 | 8 | 7.84 ± 2.55 |
| GLOBE-CE - 8 | 95.37 ± 2.9 | 1.89 ± 0.38 | 2.0 | 1.03 ± 0.01 | 58.42 ± 18.45 | 2.25 ± 0.33 | 1.0 | 0.45 ± 0.0 | 80.25 ± 9.62 | 2.51 ± 0.33 | 1.0 | 1.36 ± 0.05 |
| DGLOBE-CE - 8 | 98.09 ± 1.72 | 2.34 ± 0.17 | 7.8 | 2.34 ± 0.03 | 69.89 ± 15.35 | 2.47 ± 0.23 | 5.6 | 1.02 ± 0.04 | 88.17 ± 8.2 | 2.64 ± 0.78 | 6.2 | 3.06 ± 0.04 |
| C-GLANCE-8 | 96.43 ± 3.6 | 0.86 ± 0.18 | 8.0 | 78.38 ± 0.0 | 100.0 ± 0.0 | 1.18 ± 0.05 | 8.0 | 63.1 ± 0.0 | 100.0 ± 0.0 | 1.02 ± 0.04 | 8.0 | 68.96 ± 0.0 |
| T-GLANCE-8 | 96.65 ± 4.08 | 0.94 ± 0.3 | 4.0 | 32.19 ± 1.34 | 99.55 ± 0.91 | 1.34 ± 0.22 | 3.8 | 29.11 ± 1.18 | 99.46 ± 1.08 | 1.07 ± 0.16 | 3.8 | 68.67 ± 1.51 |

Table 16 presents a comparative analysis of all algorithms for the German Credit dataset. `Fast AReS` remains the weakest method overall, continuing to exhibit poor effectiveness. Notably, while `CET` incurs a slightly higher cost than on the COMPAS dataset, it achieves significantly greater effectiveness, and the increase in cost is quite reasonable. The `GLOBE-CE` and `dGLOBE-CE` methods demonstrate mixed results. They perform relatively well under the DNN model, achieving effectiveness scores above 90%, but their effectiveness declines when applied to the LR and XGB models. On

the other hand, `T-GLANCE` and `C-GLANCE` continue to excel due to their robustness and consistent high performance. These methods consistently reach near-perfect effectiveness ($\sim$100%, and only for DNN 95%-97%) across all models while maintaining very low costs. Notably, `C-GLANCE`-8 achieves the lowest cost of 0.86 under the DNN model, while still maintaining high effectiveness.

Table 17: Detailed results for the solution of $s$-GCE ($s = 4$ and $s = 8$) for Default Credit dataset. The table reports effectiveness, cost, size, and runtime, including their standard deviations for each method.

| METHOD / MODEL | DNN | | | | LR | | | | XGB | | | |
|---|---|---|---|---|---|---|---|---|---|---|---|---|
| | eff | avc | size | RunTime | eff | avc | size | RunTime | eff | avc | size | RunTime |
| Fast AReS-4 | 18.88 ± 2.16 | 1.0 ± 0.0 | 4 | 789.48 ± 306.69 | 10.85 ± 5.45 | 1.07 ± 0.13 | 4 | 244.78 ± 128.93 | 31.86 ± 5.12 | 1.05 ± 0.04 | 4 | 226.08 ± 107.9 |
| CET - 4 | 98.87 ± 0.62 | 6.32 ± 2.28 | 3.0 | 4404.58 ± 453.7 | 100.0 ± 0.0 | 3.79 ± 1.31 | 1.0 | 3711.21 ± 164.94 | 86.29 ± 9.94 | 4.5 ± 2.64 | 3.0 | 7982.05 ± 668.76 |
| GroupCF - 4 | 79.6 ± 20.79 | 1.53 ± 0.62 | 4 | 2415.7 ± 0.62 | 95.4 ± 9.2 | 1.94 ± 1.2 | 4 | 306.76 ± 1.2 | 95.2 ± 1.6 | 1.41 ± 0.64 | 4 | 991.53 ± 0.64 |
| GLOBE-CE - 4 | 76.94 ± 37.55 | 5.14 ± 0.35 | 1.4 | 3.33 ± 0.2 | 99.94 ± 0.07 | 3.42 ± 1.99 | 1.6 | 1.76 ± 0.04 | 82.7 ± 7.26 | 20.82 ± 1.73 | 2.6 | 2.44 ± 0.05 |
| dGLOBE-CE - 4 | 87.38 ± 18.69 | 5.96 ± 4.14 | 3.4 | 4.98 ± 0.21 | 99.94 ± 0.07 | 10.38 ± 7.76 | 1.2 | 2.54 ± 0.07 | 97.47 ± 0.82 | 42.58 ± 3.57 | 4.0 | 3.48 ± 0.09 |
| C-GLANCE-4 | 100.0 ± 0.0 | 1.2 ± 0.4 | 4.0 | 290.89 ± 0.0 | 100.0 ± 0.0 | 1.05 ± 0.11 | 4.0 | 243.45 ± 0.0 | 98.13 ± 1.05 | 3.68 ± 1.64 | 4.0 | 319.45 ± 0.0 |
| T-GLANCE-4 | 100.0 ± 0.0 | 1.48 ± 0.43 | 3.0 | 775.09 ± 0.0 | 100.0 ± 0.0 | 1.43 ± 0.36 | 3.0 | 876.08 ± 0.0 | 94.6 ± 3.13 | 2.75 ± 1.42 | 3.0 | 688.19 ± 0.0 |
| Fast AReS-8 | 23.75 ± 1.81 | 1.02 ± 0.03 | 8 | 428.04 ± 199.29 | 11.67 ± 6.18 | 1.04 ± 0.08 | 8 | 272.64 ± 120.89 | 36.88 ± 5.46 | 1.07 ± 0.06 | 8 | 280.88 ± 135.37 |
| CET - 8 | 98.53 ± 2.11 | 3.68 ± 1.46 | 4.0 | 2500.02 ± 328.94 | 100.0 ± 0.0 | 2.0 ± 1.36 | 2.0 | 1795.08 ± 182.55 | 90.01 ± 10.44 | 2.61 ± 1.06 | 5.0 | 5105.11 ± 923.27 |
| GroupCF - 8 | 84.8 ± 13.23 | 2.15 ± 0.98 | 8 | 2476.21 ± 0.82 | 100.0 ± 0.0 | 3.98 ± 2.86 | 8 | 279.86 ± 2.86 | 97.8 ± 0.75 | 1.92 ± 1.33 | 8 | 631.31 ± 1.33 |
| GLOBE-CE - 8 | 93.25 ± 11.26 | 2.73 ± 0.85 | 3.6 | 3.49 ± 0.27 | 99.94 ± 0.07 | 2.24 ± 0.98 | 1.8 | 1.75 ± 0.05 | 84.94 ± 6.59 | 11.21 ± 1.29 | 5.2 | 2.47 ± 0.06 |
| dGLOBE-CE - 8 | 99.95 ± 0.04 | 6.77 ± 3.67 | 4.8 | 6.79 ± 0.16 | 99.94 ± 0.07 | 10.38 ± 7.76 | 1.2 | 3.51 ± 0.07 | 99.08 ± 0.66 | 28.73 ± 9.8 | 8.0 | 5.05 ± 0.12 |
| C-GLANCE-8 | 100.0 ± 0.0 | 1.2 ± 0.4 | 8.0 | 289.46 ± 0.0 | 100.0 ± 0.0 | 1.05 ± 0.11 | 8.0 | 226.96 ± 0.0 | 98.87 ± 0.59 | 2.14 ± 0.18 | 8.0 | 257.39 ± 0.0 |
| T-GLANCE-8 | 100.0 ± 0.0 | 1.25 ± 0.38 | 3.8 | 177.22 ± 0.46 | 100.0 ± 0.0 | 1.13 ± 0.18 | 3.8 | 167.72 ± 7.88 | 94.1 ± 3.57 | 1.85 ± 0.16 | 4.0 | 191.86 ± 4.82 |

Table 17 provides a comparative analysis of all algorithms for the Default Credit dataset. `C-GLANCE` and `T-GLANCE` methods exhibit perfect effectiveness scores across both the DNN and LR models while maintaining the lowest costs, second only to `Fast AReS`, highlighting their superior cost-effectiveness.

In the XGB model, `GroupCF`-4 dominates `T-GLANCE`-4 and achieves smaller cost compared to `C-GLANCE`-4, which, however, has better performance in terms of effectiveness. Concerning the case $s = 8$, `GroupCF`-8 has slightly better cost but worse effectiveness than `C-GLANCE`-8, and slightly better effectiveness but worse cost than `T-GLANCE`-8. Overall, our methods ensure competitive, robust and highly efficient performance.

Table 18: Detailed results for the solution of $s$-GCE ($s = 4$ and $s = 8$) for HELOC dataset. The table reports effectiveness, cost, size, and runtime, including their standard deviations for each method.

| METHOD / MODEL | DNN | | | | LR | | | | XGB | | | |
|---|---|---|---|---|---|---|---|---|---|---|---|---|
| | eff | avc | size | RunTime | eff | avc | size | RunTime | eff | avc | size | RunTime |
| Fast AReS-4 | 12.19 ± 0.58 | 1.03 ± 0.05 | 4 | 1562.21 ± 656.59 | 9.23 ± 1.24 | 1.12 ± 0.1 | 4 | 961.23 ± 541.41 | 8.49 ± 1.32 | 1.16 ± 0.13 | 4 | 963.0 ± 492.98 |
| CET - 4 | 86.78 ± 10.62 | 8.67 ± 3.25 | 1.0 | 11090.16 ± 4597.13 | 100.0 ± 0.0 | 3.57 ± 1.48 | 2.0 | 2662.77 ± 739.88 | 86.78 ± 6.7 | 12.51 ± 2.75 | 2.0 | 17339.7 ± 1363.44 |
| GroupCF - 4 | 80.4 ± 10.17 | 3.09 ± 0.91 | 4 | 468.52 ± 0.91 | 90.6 ± 3.93 | 2.4 ± 1.38 | 4 | 209.26 ± 1.38 | 78.4 ± 5.82 | 5.63 ± 1.93 | 4 | 201.24 ± 1.93 |
| GLOBE-CE - 4 | 42.72 ± 46.97 | 11.77 ± 15.87 | 1.4 | 1.43 ± 0.13 | 99.9 ± 0.0 | 0.6 ± 0.54 | 3.0 | 0.65 ± 0.06 | 27.66 ± 5.06 | 12.52 ± 32.48 | 2.4 | 1.18 ± 0.02 |
| dGLOBE-CE - 4 | 99.96 ± 0.05 | 11.07 ± 8.6 | 2.0 | 2.2 ± 0.14 | 99.9 ± 0.0 | 1.63 ± 0.35 | 4.0 | 0.94 ± 0.05 | 77.64 ± 11.51 | 128.0 ± 0.0 | 4.0 | 1.77 ± 0.03 |
| C-GLANCE-4 | 99.94 ± 0.05 | 11.24 ± 1.37 | 4.0 | 307.06 ± 0.0 | 100.0 ± 0.0 | 1.55 ± 0.54 | 4.0 | 243.6 ± 0.0 | 98.94 ± 0.66 | 19.99 ± 1.91 | 4.0 | 183.45 ± 0.0 |
| T-GLANCE-4 | 99.8 ± 0.24 | 10.9 ± 1.22 | 3.0 | 135.37 ± 0.0 | 100.0 ± 0.0 | 1.58 ± 0.37 | 3.0 | 108.79 ± 0.0 | 96.42 ± 2.45 | 24.85 ± 8.4 | 3.0 | 72.95 ± 0.0 |
| Fast AReS-8 | 15.44 ± 1.91 | 1.05 ± 0.06 | 8 | 1070.19 ± 558.28 | 10.95 ± 1.73 | 1.11 ± 0.07 | 8 | 901.3 ± 493.42 | 11.22 ± 2.05 | 1.09 ± 0.09 | 8 | 829.62 ± 420.35 |
| CET - 8 | 99.36 ± 0.92 | 8.86 ± 2.99 | 4.0 | 6758.87 ± 3384.51 | 100.0 ± 0.0 | 2.99 ± 1.3 | 2.0 | 1628.08 ± 344.57 | 90.25 ± 2.51 | 12.99 ± 3.49 | 6.0 | 9108.94 ± 1624.21 |
| GroupCF - 8 | 84.8 ± 12.32 | 1.98 ± 0.5 | 8 | 501.45 ± 2.5 | 96.6 ± 1.96 | 1.98 ± 0.5 | 8 | 249.14 ± 0.5 | 81.4 ± 4.22 | 5.14 ± 1.46 | 8 | 197.68 ± 1.46 |
| GLOBE-CE - 8 | 45.49 ± 44.72 | 14.19 ± 14.14 | 3.4 | 1.44 ± 0.02 | 99.9 ± 0.0 | 0.58 ± 0.5 | 7.0 | 0.64 ± 0.02 | 28.88 ± 4.71 | 12.51 ± 30.88 | 3.0 | 1.18 ± 0.02 |
| dGLOBE-CE - 8 | 100.0 ± 0.0 | 6.65 ± 0.15 | 4.4 | 3.2 ± 0.02 | 99.9 ± 0.0 | 1.63 ± 0.35 | 8.0 | 1.35 ± 0.04 | 81.95 ± 10.38 | 128.0 ± 0.0 | 8.0 | 2.55 ± 0.04 |
| C-GLANCE-8 | 99.98 ± 0.04 | 9.42 ± 2.22 | 8.0 | 301.47 ± 0.0 | 100.0 ± 0.0 | 1.2 ± 0.12 | 8.0 | 217.9 ± 0.0 | 99.0 ± 0.55 | 16.65 ± 2.44 | 8.0 | 182.83 ± 0.0 |
| T-GLANCE-8 | 99.78 ± 0.21 | 9.92 ± 1.83 | 4.0 | 40.64 ± 0.43 | 100.0 ± 0.0 | 1.52 ± 0.32 | 3.0 | 39.0 ± 0.89 | 98.25 ± 1.85 | 18.75 ± 4.92 | 4.0 | 75.9 ± 0.8 |

Table 18 presents a comparative analysis of all algorithms for the Heloc dataset, which, due to its exclusively numeric features, poses challenges in achieving low costs. Despite this complexity, the methods `T-GLANCE`, `C-GLANCE`, and `dGLOBE-CE` stand out by delivering near-perfect or perfect effectiveness in the DNN models, while `T-GLANCE`, `C-GLANCE`, `dGLOBE-CE`, and `GLOBE-CE` excel in the LR models. `dGLOBE-CE`-8 provides the best combination of effectiveness and cost for DNN, and `GLOBE-CE`-8 achieves the same for LR. For the XGB models, `T-GLANCE` and `C-GLANCE` lead in effectiveness across all methods. Overall, our methods demonstrate strong and consistent performance, unlike other methods, where results fluctuate considerably.

Table 19 presents a comparative analysis of all algorithms for the Adult dataset. The `Fast AReS` method consistently performs poorly, achieving notably low effectiveness scores across all models.

It is also noteworthy that in the Adult dataset which is the largest of all in terms of instances and dimensions, the `CET` method did not run properly due to exceedingly high runtimes or infeasibility determined by the Gurobi optimizer. This is indicated by the ">20h or Infeasible (on Gurobi)" note present in the runtime column for both `CET`-4 and `CET`-8 across all models.

Table 19: Detailed results for the solution of $s$-GCE ($s = 4$ and $s = 8$) for Adult dataset. The table reports effectiveness, cost, size, and runtime, including their standard deviations for each method.

| METHOD / MODEL | DNN | | | | LR | | | | XGB | | | |
|---|---|---|---|---|---|---|---|---|---|---|---|---|
| | eff | avc | size | RUNTIME | eff | avc | size | RUNTIME | eff | avc | size | RUNTIME |
| FAST ARES - 4 | 12.39 ± 1.06 | 1.0 ± 0.0 | 4.0 | 266.51 ± 127.13 | 11.74 ± 2.4 | 1.0 ± 0.0 | 4.0 | 124.2 ± 57.64 | 6.13 ± 0.42 | 1.0 ± 0.0 | 4.0 | 108.66 ± 57.21 |
| CET - 4 | - | - | - | >20H OR INFEASIBLE (ON GUROBI) | - | - | - | >20H OR INFEASIBLE (ON GUROBI) | - | - | - | >20H OR INFEASIBLE (ON GUROBI) |
| GROUPCF - 4 | 100.0 ± 0.0 | 10.08 ± 0.03 | 4.0 | 3224.75 ± 0.03 | 100.0 ± 0.0 | 1.71 ± 0.39 | 4.0 | 925.25 ± 0.39 | 96.8 ± 1.72 | 1.41 ± 0.54 | 4.0 | 596.57 ± 0.54 |
| GLOBE-CE - 4 | 99.92 ± 0.0 | 4.24 ± 0.42 | 3.0 | 6.6 ± 0.22 | 99.92 ± 0.0 | 2.68 ± 0.17 | 3.0 | 6.48 ± 0.17 | 82.87 ± 12.14 | 30.1 ± 10.39 | 2.4 | 5.44 ± 0.18 |
| DGLOBE-CE - 4 | 99.92 ± 0.0 | 10.89 ± 1.37 | 3.6 | 9.74 ± 0.21 | 99.92 ± 0.0 | 5.91 ± 0.93 | 4.0 | 9.36 ± 0.06 | 93.76 ± 1.98 | 64.76 ± 1.29 | 4.0 | 8.26 ± 0.14 |
| C-GLANCE - 4 | 100.0 ± 0.0 | 4.6 ± 0.73 | 4.0 | 246.82 ± 0.0 | 100.0 ± 0.0 | 1.04 ± 0.07 | 4.0 | 194.83 ± 0.0 | 99.85 ± 0.12 | 5.98 ± 4.22 | 4.0 | 134.27 ± 0.0 |
| T-GLANCE-4 | 100.0 ± 0.0 | 4.43 ± 0.43 | 3.0 | 410.47 ± 0.0 | 100.0 ± 0.0 | 0.69 ± 0.01 | 3.0 | 422.45 ± 0.0 | 99.86 ± 0.14 | 1.8 ± 0.51 | 3.0 | 334.14 ± 0.0 |
| FAST ARES - 8 | 13.8 ± 0.92 | 1.0 ± 0.0 | 8.0 | 271.04 ± 126.28 | 12.51 ± 2.44 | 1.0 ± 0.0 | 8.0 | 157.94 ± 73.44 | 7.19 ± 0.3 | 1.0 ± 0.0 | 8.0 | 142.82 ± 67.84 |
| CET - 8 | - | - | - | >20H OR INFEASIBLE (ON GUROBI) | - | - | - | >20H OR INFEASIBLE (ON GUROBI) | - | - | - | >20H OR INFEASIBLE (ON GUROBI) |
| GROUPCF - 8 | 100.0 ± 0.0 | 11.54 ± 2.98 | 8.0 | 3539.35 ± 2.98 | 100.0 ± 0.0 | 2.33 ± 1.91 | 8.0 | 872.1 ± 1.91 | 99.4 ± 1.2 | 1.88 ± 1.36 | 8.0 | 1705.8 ± 1.36 |
| GLOBE-CE - 8 | 100.0 ± 0.0 | 2.54 ± 0.14 | 6.6 | 6.56 ± 0.18 | 100.0 ± 0.0 | 1.67 ± 0.11 | 7.0 | 6.41 ± 0.08 | 85.76 ± 9.11 | 13.48 ± 4.71 | 3.8 | 5.58 ± 0.11 |
| DGLOBE-CE - 8 | 100.0 ± 0.0 | 11.85 ± 1.53 | 6.4 | 14.09 ± 0.4 | 100.0 ± 0.0 | 6.12 ± 0.64 | 8.0 | 13.57 ± 0.22 | 94.38 ± 2.01 | 62.37 ± 3.07 | 7.4 | 12.01 ± 0.33 |
| C-GLANCE - 8 | 100.0 ± 0.0 | 3.96 ± 0.56 | 8.0 | 240.16 ± 0.0 | 100.0 ± 0.0 | 1.03 ± 0.07 | 8.0 | 179.43 ± 0.0 | 99.87 ± 0.09 | 3.85 ± 2.33 | 8.0 | 126.65 ± 0.0 |
| T-GLANCE-8 | 100.0 ± 0.0 | 3.77 ± 0.27 | 4.0 | 119.66 ± 0.69 | 100.0 ± 0.0 | 0.79 ± 0.16 | 4.0 | 122.45 ± 0.63 | 99.94 ± 0.04 | 1.65 ± 0.39 | 4.0 | 128.22 ± 1.21 |

On the other hand, `GroupCF`, `C-GLANCE`, `T-GLANCE`, and `GLOBE-CE` methods all perform notably well in terms of effectiveness. Under the DNN model, `T-GLANCE`-4 and `T-GLANCE`-8 achieve perfect effectiveness scores of 100%, with costs of 4.43 and 3.77, respectively. `C-GLANCE`-4 and `C-GLANCE`-8 mirror this perfect effectiveness, also achieving 100% with similarly good cost performance. Notably, `GLOBE-CE`-8 achieves the best cost performance under the DNN model with an effectiveness score of 100% and a cost of 2.54.

In the LR model, `T-GLANCE`-4 and `T-GLANCE`-8 perform exceptionally well, achieving 100% effectiveness with costs of 0.69 and 0.79, respectively. `C-GLANCE` also performs admirably, with 100% effectiveness and costs 1.04 and 1.03. Similarly, in the XGB model, `T-GLANCE`-4, `T-GLANCE`-8, `C-GLANCE`-4, and `C-GLANCE`-8 maintain high effectiveness scores of 99.86%, 99.94%, 99.85%, and 99.87%, with really good respective costs.

Upon evaluating the performance across the five datasets — COMPAS, German Credit, Default Credit, HELOC, and Adult Income — it is evident that our methods, `T-GLANCE` and `C-GLANCE`, consistently demonstrate concrete performance. They achieve near-perfect or perfect effectiveness scores while maintaining low costs across different models, such as DNN, LR, and XGB. This robustness and consistency underline their efficiency and practicality in handling diverse datasets and complexities.

# I  A USER STUDY TO ASSESS HOW GLOBAL COUNTERFACTUAL EXPLANATIONS ARE PERCEIVED

## I.1  DESIGN

We have designed a user study, with the following goals.

- Assess (a) how different concepts for global counterfactual explainability are understood and (b) which concepts are preferred by *individuals who seek recourse*.
- Assess (a) how different concepts for global counterfactual explainability are understood and (b) which concepts are preferred by *the system/model owner who seeks to provide recourse* to the affected population.
- Assess how people evaluate the *trade-off between effectiveness and average recourse cost*.

The user study has three parts, with each having a different goal.

### I.1.1  THE VIEWPOINT OF INDIVIDUALS SEEKING RECOURSE

The study puts participants in the following hypothetical situation. "Consider a company that at the end of the year gives a bonus to some of its employees. There are some employees that did not receive the bonus. They would like to know what they can do differently so that they will receive a bonus next year, i.e., they want to achieve recourse. They have the option to be involved in additional *projects* to increase their chances of receiving the bonus. Working in an additional project, means that their work hours will increase."

The study then asks three questions.

**Q1** " Suppose you're an employee and have the option to be involved in one of these projects.

A. All employees working on Project A received a bonus. Among them, the average number of additional working time per week was 5 hours.

B. All employees working on Project B received a bonus. No employee worked more than 8 additional hours per week.

Which project would you choose based on the information provided?"

The goal of Q1 is to understand which type of information people prefer. A is similar to the information counterfactual directions (as in Ley et al. (2023)) provides, while B is similar to the information counterfactual actions provide.

**Q2** "Suppose you're an employee and have the option to be involved in one of these projects.

C. 75% of the employees working on Project C received a bonus. Among them, the average number of additional working time per week was 4 hours.

D. 75% of the employees working on Project D received a bonus. No employee worked more than 6 additional hours per week.

Which project would you choose based on the information provided?"

Q2 has the same goal with Q1, but it emphasizes that information comes with no guarantees.

**Q3** " Suppose you're an employee and you consider working on Project E to increase your chances of getting the bonus. There are three pieces of information about Project E.

Info-1. For employees working on Project E who received the bonus, the average number of additional working time per week was 5 hours.

Info-2. All employees working on Project E who logged at least 10 additional working hours received the bonus.

Info-3. 60% of employees working on Project E who logged at least 5 additional working hours received the bonus.

Rank the pieces of information from the most helpful to the least helpful."

Q3 aims to understand which type of information people find most helpful. Info-1 is similar to the information counterfactual directions provide, while Info-2 and Info-3 are similar to the information counterfactual actions provide.

### I.1.2 THE VIEWPOINT OF SYSTEM OWNERS SEEKING TO PROVIDE RECOURSE

This part puts the participants in the role of the system owner. "Assume you are the company CEO, summarizing the benefits of project opportunities.

This year, two types of employees, x1 and x2, did not earn a bonus. These employee types differ in how they allocate their work hours:

- x1 employees spend more time in the office and less in the field (i.e., outside the office).
- x2 employees spend more time in the field and less in the office.

The number of employees who did not earn a bonus is evenly split between these two groups."

Then, with reference to Figure 5, the study provides this additional information. "Employee types are represented as points on the office hours-field hours plane, shown in the left image. Projects are depicted as directions on this plane, indicating the need for additional office and field hours. In some cases, it is more intuitive to view projects from the perspective of the employee types, represented as the origin o, as shown in the right image. "

Then the user study asks two questions.

**Q1** "There are three possible projects that employees can work on:

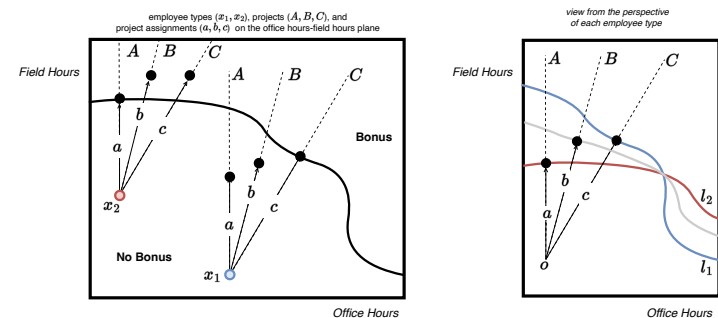

Figure 5: Image used in the User Study to illustrate the concepts of employee types, projects, and project placements.

A: The project requiring the least effort from any individual employee to receive the bonus.

B: The project requiring the least average effort across all employees to receive the bonus.

C: The project requiring the smallest maximum effort from any employee to receive the bonus.

The CEO must decide which project to promote to employees. Which one should it be?"

The goal of Q1 is to understand what kind of information system owners might find most helpful. B is similar to the information counterfactual directions provide, while A and C are similar to the information counterfactual actions provide.

**Q2** "Consider the following three project placements. A placement refers to an assignment to a project with fixed office and field hours. These placements are represented as vectors along the project directions on the office hours-field hours plane.

a: Placement on Project A requiring the least effort from any employee to receive the bonus, i.e., 200 hours (+200 field hours).

b: Placement on Project B requiring the least average effort across employees to receive the bonus, i.e., 300 hours (+50 office hours, +250 field hours).

c: Placement on Project C requiring the least maximum effort from any employee to receive the bonus, i.e., 400 hours (+150 office hours, +250 field hours).

The CEO must decide which project placement to offer to employees. Which one should it be?"

The goal of Q2 is to understand which action (project placement) system owners might prefer. Action b is similar to the action one can extract from counterfactual directions, while b and c are optimal GCE actions, with c in particular being the action with the minimum cost among those that achieve perfect effectiveness (optimal solution to the 1-GCE problem).

I.1.3  EVALUATING THE TRADE-OFF BETWEEN EFFECTIVENESS AND AVERAGE RECOURSE COST

In the last part, we ask participants to investigate the effectiveness and average recourse costs that are achieved by different methods for global counterfactual explainability. We show them figures like those in Figure 3, but anonymized and randomly permuted, and ask them to select the superior method.

Specifically, we provide them with this information. " In the following images, you will see the evaluation of various algorithms that produce global recourse summaries for different ML models applied on different datasets. A global recourse summary outlines a few actions that individuals can take to achieve recourse, i.e., obtain a favorable outcome

Each algorithm is evaluated on two aspects:

- Effectiveness: The percentage of the population for whom the algorithm provides recourse. The higher (up to 100%) the effectiveness is, the better the algorithm.

- Average Cost: The average cost incurred by individuals who achieve recourse. The lower (down to 0) the cost is, the better the algorithm.

The images display the effectiveness and average cost for an algorithm as a point with error bars; the error bars denote the standard deviation across multiple evaluations. The closer a point is to the top left corner, the better the algorithm."

We then ask two types of questions for each figure, depicting either the four methods for implicit GCE or the three methods for explicit GCE.

**Q1** "Based on the evaluation results presented in the figure, rank the three/four algorithms from best (1st position) to worst (3rd/4th position)."

**Q2** "How many times (out of 15) is Algo 0/1/2/3 the better algorithm?"

These questions aim to understand how people combine the two aspects of effectiveness and average cost to assess the quality of global counterfactual explanations.

### I.2 RESULTS

Following the best practices outlined in Ley et al. (2023); Chowdhury et al. (2022), we recruited 40 participants consisting of (a) master's students in a data science program who attended lectures on machine learning explainability and (b) PhD students from various machine learning domains with extensive knowledge of explainability. After excluding six participants due to inconsistent responses to repeated questions and/or incomplete rankings, we analyzed the responses of the remaining 34 participants.

In the first part of the study, where participants assumed the role of individuals seeking recourse, we find that *participants prefer information conveyed through concrete counterfactual actions* (similar to GCEs produced by methods like ARES Rawal & Lakkaraju (2020) and this work) over counterfactual directions represented by average-cost actions (akin to GCEs generated by GLOBE-CE Ley et al. (2023)). Specifically, for Q1, 56% of participants preferred information tied to concrete actions (project B) over average-cost actions (project A). This preference increased to 71% in Q2 when uncertainty rose (effectiveness decreased from 100% to 75%). When asked to rank their preference among three types of information in Q3, 62% of participants rated Info-1, which represents an optimal solution to the 1-GCE problem (similar to action $c$ in Figure 5), as the most helpful. Conversely, 41% ranked Info-2, which represents the average-cost action along the optimal direction as defined in Ley et al. (2023) (similar to action $b$ in Figure 5), as the least helpful. Additionally, Info-3, another optimal solution to the 1-GCE problem (similar to action $a$ in Figure 5), was considered more helpful than Info-2.

In the second part of the study, where participants assumed the role of system owners aiming to provide recourse to adversely affected individuals, we again found that *participants preferred information conveyed through concrete counterfactual actions* rather than counterfactual directions. Specifically, when asked (Q1) to choose among three projects corresponding to directions $A$, $B$, and $C$ in Figure 5, 65% favored projects/directions $A$ or $C$ (both optimal 1-GCE solutions) over project/direction $B$ (aligned with the optimal direction as defined in Ley et al. (2023)), with $C$ emerging as the clear favorite, selected 50% of the time. A similar pattern was observed in project placement decisions (Q2), involving actions $a$, $b$, and $c$ in Figure 5, where 60% of participants preferred the optimal actions $a$ and $c$ over the average-cost action $b$.

In the third part of the study, participants were asked to evaluate which method for global counterfactual explanations is superior based on performance in terms of average cost and effectiveness. We found that *participants unanimously agree that the GLANCE methods outperform others*. Specifically, in the first two questions, participants assessed explicit GCE methods (T-GLANCE, ARES, CET). All participants (100%) identified T-GLANCE as the superior method, finding it outperformed the others approximately 13.41 out of 15 times (as illustrated in Figure 4). In the next two questions, participants evaluated implicit GCE methods (C-GLANCE, GLOBE-CE, dGLOBE-CE, GroupCF).

Table 20: Local vs global counterfactuals

| MODELS | ALGORITHMS | DATASETS | | | |
|--------|-----------|----------|---|---|---|
| | | COMPAS | | GERMAN CREDIT | |
| | | EFF | AVC | EFF | AVC |
| DNN | T-GLANCE | 99.53 ± 0.22 | 2.62 ± 0.24 | 96.97 ± 3.48 | 1.59 ± 0.8 |
| | C-GLANCE | 100.0 ± 0.0 | 2.34 ± 0.43 | 95.31 ± 3.15 | 1.25 ± 0.33 |
| | LOCAL | 100.0 ± 0.0 | 2.75 ± 0.53 | 100.0 ± 0.0 | 4.74 ± 0.93 |
| LR | T-GLANCE | 99.79 ± 0.22 | 2.19 ± 0.1 | 99.58 ± 0.83 | 1.54 ± 0.32 |
| | C-GLANCE | 100.0 ± 0.0 | 2.33 ± 0.38 | 100.0 ± 0.0 | 1.21 ± 0.06 |
| | LOCAL | 100.0 ± 0.0 | 2.73 ± 0.23 | 100.0 ± 0.0 | 4.9 ± 0.4 |
| XGB | T-GLANCE | 99.02 ± 0.93 | 2.6 ± 0.46 | 99.46 ± 1.08 | 1.33 ± 0.42 |
| | C-GLANCE | 99.51 ± 0.46 | 2.96 ± 0.82 | 100.0 ± 0.0 | 1.06 ± 0.03 |
| | LOCAL | 100.0 ± 0.0 | 2.87 ± 0.27 | 100.0 ± 0.0 | 5.09 ± 0.33 |

Again, all participants (100%) selected `C-GLANCE` as the superior method, noting it outperformed the others approximately 10.81 out of 15 times (as depicted in Figure 3).

## J  ADDRESSING POTENTIAL EDGE CASES IN RECOURSE ALGORITHMS

While the proposed algorithms are designed to provide robust and actionable recourse, certain edge cases may pose challenges:

Highly nonlinear decision boundaries might cause the model's decision boundary to vary significantly across small regions, leading to oversimplified recourse suggestions. We address this issue by using proximity-aware clustering and local counterfactual action generation to better approximate the decision boundary, ensuring more accurate and actionable recommendations.

Sparse data regions or outlier points might result in unrealistic, infeasible, or overly tailored recourse actions. To mitigate these challenges, we generate many clusters and diverse counterfactuals, ensuring that feasible solutions are identified without being disproportionately influenced by outliers. This approach enhances robustness and broad applicability.

Ambiguity in selecting the "optimal" action can arise when multiple actions have similar costs and effectiveness, potentially leading to overly complex recourse suggestions. We resolve this by employing predefined criteria that prioritize actionable and interpretable solutions while maintaining a balance between cost and effectiveness.

## K  LOCAL VS. GLOBAL COUNTERFACTUAL COSTS

To illustrate the challenges in defining optimal solutions for global counterfactual explanations, we conducted an experiment comparing the cost of local counterfactuals to the global counterfactual actions generated by our method. Using the Compas and German Credit datasets, we generated all possible local counterfactuals for each instance and calculated the average cost across the datasets. The results are presented in Table 20 under the row "LOCAL".

Surprisingly, despite generating only 4 global counterfactual actions, our method achieved significantly lower average costs than the local counterfactuals, even though both were generated using the same local counterfactual method. This counterintuitive result highlights the inherent variability and complexity in local counterfactual generation, emphasizing the difficulty in defining an optimal solution at both local and global levels.

This experiment demonstrates the effectiveness of our approach in achieving cost-efficient global counterfactuals while reinforcing the challenges of theoretical analysis in this domain.

Table 21: Default Credit Effectiveness Mean

| MODEL | INITIAL CLUSTERS DIVERSE ACTIONS | 10.0 | 20.0 | 30.0 | 40.0 | 50.0 | 60.0 | 70.0 | 80.0 | 90.0 | 100.0 |
|---|---|---|---|---|---|---|---|---|---|---|---|
| DNN | 5 | 99.91 | 100.00 | 99.86 | 99.97 | 100.00 | 99.94 | 100.00 | 99.97 | 99.97 | 100.00 |
| | 10 | 99.91 | 100.00 | 99.88 | 99.97 | 99.97 | 100.00 | 99.97 | 99.97 | 99.97 | 99.97 |
| | 15 | 99.89 | 100.00 | 99.94 | 100.00 | 99.97 | 99.94 | 99.97 | 99.97 | 100.00 | 100.00 |
| | 20 | 99.91 | 100.00 | 99.97 | 100.00 | 99.97 | 99.97 | 100.00 | 99.97 | 99.94 | 100.00 |
| | 25 | 99.89 | 100.00 | 100.00 | 99.97 | 99.97 | 99.97 | 99.94 | 100.00 | 99.97 | 99.97 |
| | 30 | 99.94 | 100.00 | 99.97 | 100.00 | 99.97 | 100.00 | 100.00 | 100.00 | 100.00 | 100.00 |
| | 35 | 99.94 | 99.97 | 99.97 | 99.97 | 99.97 | 100.00 | 99.97 | 99.97 | 99.97 | 100.00 |
| | 40 | 99.94 | 100.00 | 100.00 | 99.97 | 99.97 | 99.97 | 99.97 | 99.97 | 100.00 | 100.00 |
| | 45 | 99.94 | 100.00 | 99.94 | 100.00 | 100.00 | 99.97 | 99.97 | 99.97 | 100.00 | 100.00 |
| | 50 | 99.91 | 100.00 | 99.97 | 99.97 | 99.97 | 99.97 | 100.00 | 100.00 | 100.00 | 100.00 |
| LR | 5 | 100.00 | 100.00 | 100.00 | 100.00 | 100.00 | 100.00 | 100.00 | 100.00 | 100.00 | 100.00 |
| | 10 | 100.00 | 100.00 | 100.00 | 100.00 | 100.00 | 100.00 | 100.00 | 100.00 | 100.00 | 100.00 |
| | 15 | 100.00 | 100.00 | 100.00 | 100.00 | 100.00 | 100.00 | 100.00 | 100.00 | 100.00 | 100.00 |
| | 20 | 100.00 | 100.00 | 100.00 | 100.00 | 100.00 | 100.00 | 100.00 | 100.00 | 100.00 | 100.00 |
| | 25 | 100.00 | 100.00 | 100.00 | 100.00 | 100.00 | 100.00 | 100.00 | 100.00 | 100.00 | 100.00 |
| | 30 | 100.00 | 100.00 | 100.00 | 100.00 | 100.00 | 100.00 | 100.00 | 100.00 | 100.00 | 100.00 |
| | 35 | 100.00 | 100.00 | 100.00 | 100.00 | 100.00 | 100.00 | 100.00 | 100.00 | 100.00 | 100.00 |
| | 40 | 100.00 | 100.00 | 100.00 | 100.00 | 100.00 | 100.00 | 100.00 | 100.00 | 100.00 | 100.00 |
| | 45 | 100.00 | 100.00 | 100.00 | 100.00 | 100.00 | 100.00 | 100.00 | 100.00 | 100.00 | 100.00 |
| | 50 | 100.00 | 100.00 | 100.00 | 100.00 | 100.00 | 100.00 | 100.00 | 100.00 | 100.00 | 100.00 |
| XGB | 5 | 94.42 | 94.39 | 95.48 | 96.50 | 95.53 | 94.95 | 96.29 | 96.14 | 96.42 | 94.59 |
| | 10 | 94.03 | 96.08 | 97.38 | 95.99 | 95.85 | 98.24 | 96.79 | 96.94 | 95.54 | 96.68 |
| | 15 | 95.95 | 95.97 | 97.73 | 96.30 | 97.39 | 98.01 | 97.97 | 97.94 | 97.92 | 96.49 |
| | 20 | 96.02 | 96.62 | 97.79 | 96.19 | 98.54 | 98.54 | 98.62 | 98.52 | 98.11 | 98.34 |
| | 25 | 96.66 | 95.72 | 98.57 | 96.84 | 98.62 | 98.65 | 98.57 | 97.76 | 98.21 | 98.36 |
| | 30 | 96.68 | 96.69 | 98.62 | 96.31 | 98.49 | 98.67 | 98.46 | 98.49 | 98.43 | 98.97 |
| | 35 | 96.73 | 97.17 | 99.00 | 97.56 | 98.67 | 98.85 | 98.46 | 98.84 | 98.58 | 98.70 |
| | 40 | 96.94 | 97.69 | 98.82 | 97.44 | 98.90 | 98.63 | 98.92 | 99.07 | 98.80 | 98.99 |
| | 45 | 97.76 | 98.51 | 98.82 | 97.69 | 98.82 | 99.00 | 98.44 | 98.89 | 98.82 | 99.07 |
| | 50 | 97.20 | 98.49 | 98.82 | 98.63 | 98.74 | 99.11 | 98.74 | 98.94 | 98.87 | 99.01 |

## L    IMPACT OF INITIAL CLUSTERS AND DIVERSE ACTIONS

To evaluate the influence of key parameters in C-GLANCE on solution quality, interpretability, and runtime, we analyzed the role of initial clusters and diverse candidate actions. The product of these parameters defines the total number of generated actions, which directly impacts the algorithm's performance.

Increasing the number of generated actions often improves effectiveness but typically comes at the expense of increased cost. However, selecting a subset of actions with optimal effectiveness tends to reduce overall cost. This pattern is clearly observed in Tables 21 to 24.

Regarding individual parameter effects, increasing the number of initial clusters proves particularly beneficial for larger datasets with widely distributed points. This approach allows for better grouping in the feature space, thereby enhancing the relevance and feasibility of the generated counterfactual actions. Conversely, increasing the diversity of candidate actions is more advantageous for complex models with intricate decision boundaries. A higher diversity ensures that the generated counterfactual actions are well-aligned with the model's structure, leading to better adaptation to the complexities of the decision boundary.

Table 22: Default Credit Average Cost Mean

| MODEL | INITIAL CLUSTERS DIVERSE ACTIONS | 10.0 | 20.0 | 30.0 | 40.0 | 50.0 | 60.0 | 70.0 | 80.0 | 90.0 | 100.0 |
|---|---|---|---|---|---|---|---|---|---|---|---|
| DNN | 5 | 4.85 | 3.15 | 1.45 | 1.58 | 1.40 | 2.27 | 1.13 | 2.65 | 1.41 | 1.32 |
| | 10 | 2.91 | 2.53 | 1.98 | 1.41 | 1.59 | 1.39 | 1.33 | 1.51 | 1.03 | 1.28 |
| | 15 | 4.66 | 1.90 | 1.19 | 1.25 | 1.21 | 1.01 | 1.03 | 1.11 | 1.02 | 1.03 |
| | 20 | 2.76 | 1.71 | 1.27 | 1.39 | 1.02 | 1.00 | 1.01 | 1.08 | 1.02 | 1.01 |
| | 25 | 3.91 | 1.96 | 1.27 | 1.40 | 1.11 | 1.01 | 1.29 | 1.25 | 1.02 | 1.00 |
| | 30 | 2.16 | 1.90 | 1.27 | 1.40 | 1.01 | 1.01 | 1.21 | 1.28 | 1.01 | 1.01 |
| | 35 | 2.16 | 1.62 | 1.27 | 1.39 | 1.00 | 1.01 | 1.29 | 1.07 | 1.00 | 1.00 |
| | 40 | 1.82 | 1.47 | 1.21 | 1.23 | 1.00 | 1.01 | 1.02 | 1.19 | 1.01 | 1.01 |
| | 45 | 1.78 | 1.54 | 1.21 | 1.07 | 1.01 | 1.00 | 1.02 | 1.07 | 1.00 | 1.01 |
| | 50 | 1.82 | 1.34 | 1.02 | 1.03 | 1.01 | 1.00 | 1.22 | 1.08 | 1.01 | 1.01 |
| LR | 5 | 1.93 | 1.43 | 1.33 | 1.47 | 1.45 | 1.48 | 1.30 | 1.27 | 1.32 | 1.51 |
| | 10 | 1.91 | 1.13 | 1.18 | 1.34 | 1.43 | 1.29 | 1.26 | 1.09 | 1.02 | 1.38 |
| | 15 | 1.59 | 1.13 | 1.17 | 1.27 | 1.23 | 1.34 | 1.22 | 1.09 | 1.23 | 1.17 |
| | 20 | 1.22 | 1.01 | 1.27 | 1.35 | 1.22 | 1.24 | 1.20 | 1.11 | 0.99 | 1.08 |
| | 25 | 1.20 | 1.13 | 1.19 | 1.10 | 1.11 | 1.32 | 1.19 | 1.05 | 1.17 | 1.10 |
| | 30 | 1.20 | 1.00 | 1.27 | 1.13 | 1.22 | 1.20 | 1.09 | 1.19 | 1.21 | 1.18 |
| | 35 | 1.11 | 1.00 | 1.16 | 1.12 | 1.15 | 1.17 | 1.05 | 0.99 | 1.07 | 1.23 |
| | 40 | 1.11 | 1.00 | 1.20 | 1.16 | 1.10 | 1.11 | 1.11 | 1.00 | 0.99 | 0.99 |
| | 45 | 1.11 | 1.00 | 1.11 | 1.07 | 1.08 | 1.06 | 1.18 | 0.97 | 1.21 | 0.98 |
| | 50 | 1.11 | 1.00 | 1.05 | 1.07 | 1.07 | 1.15 | 1.06 | 1.09 | 1.07 | 0.96 |
| XGB | 5 | 3.44 | 3.43 | 2.83 | 2.76 | 3.30 | 2.99 | 2.67 | 2.67 | 2.65 | 3.39 |
| | 10 | 2.97 | 2.81 | 2.91 | 3.88 | 5.08 | 4.05 | 2.97 | 2.89 | 2.99 | 3.47 |
| | 15 | 3.13 | 2.47 | 2.83 | 2.81 | 3.92 | 3.89 | 3.56 | 3.30 | 3.26 | 4.06 |
| | 20 | 2.98 | 3.35 | 3.81 | 3.10 | 4.08 | 4.38 | 3.69 | 3.37 | 3.17 | 4.04 |
| | 25 | 2.95 | 3.15 | 3.81 | 3.03 | 2.63 | 3.29 | 4.26 | 3.75 | 3.75 | 3.84 |
| | 30 | 3.42 | 3.82 | 3.97 | 3.16 | 4.59 | 2.97 | 4.48 | 3.23 | 3.57 | 4.45 |
| | 35 | 2.65 | 3.01 | 4.18 | 3.06 | 5.12 | 4.24 | 3.27 | 3.43 | 4.10 | 3.75 |
| | 40 | 2.43 | 3.09 | 3.39 | 3.98 | 3.54 | 4.57 | 4.12 | 3.52 | 3.74 | 4.59 |
| | 45 | 2.57 | 3.83 | 3.39 | 4.03 | 2.84 | 4.84 | 5.66 | 4.11 | 3.47 | 5.14 |
| | 50 | 2.50 | 3.08 | 2.89 | 3.79 | 3.58 | 3.94 | 4.99 | 4.39 | 4.45 | 3.77 |

Table 23: HELOC Effectiveness Mean

| MODEL | INITIAL CLUSTERS DIVERSE ACTIONS | 10.0 | 20.0 | 30.0 | 40.0 | 50.0 | 60.0 | 70.0 | 80.0 | 90.0 | 100.0 |
|---|---|---|---|---|---|---|---|---|---|---|---|
| DNN | 5 | 99.21 | 99.67 | 99.61 | 99.34 | 99.53 | 99.46 | 99.88 | 99.69 | 99.88 | 99.61 |
| | 10 | 99.41 | 99.66 | 99.22 | 99.28 | 99.43 | 99.92 | 99.92 | 99.86 | 99.92 | 99.92 |
| | 15 | 99.36 | 99.08 | 99.21 | 99.27 | 99.39 | 99.92 | 99.92 | 99.73 | 99.92 | 99.94 |
| | 20 | 99.72 | 99.08 | 99.14 | 99.67 | 99.92 | 99.89 | 99.98 | 99.92 | 99.97 | 99.98 |
| | 25 | 99.74 | 99.08 | 99.14 | 99.39 | 99.98 | 99.92 | 99.98 | 99.96 | 99.98 | 99.98 |
| | 30 | 99.78 | 99.08 | 99.23 | 99.98 | 99.98 | 100.00 | 99.95 | 99.98 | 99.96 | 99.98 |
| | 35 | 99.78 | 99.10 | 99.27 | 99.98 | 100.00 | 100.00 | 99.98 | 99.98 | 99.96 | 99.98 |
| | 40 | 99.76 | 98.89 | 99.60 | 100.00 | 100.00 | 99.98 | 99.94 | 99.98 | 99.98 | 100.00 |
| | 45 | 99.76 | 99.12 | 99.65 | 100.00 | 99.98 | 99.98 | 100.00 | 99.98 | 99.98 | 99.96 |
| | 50 | 99.76 | 99.12 | 99.65 | 100.00 | 99.98 | 100.00 | 99.94 | 99.98 | 99.98 | 99.96 |
| LR | 5 | 100.00 | 100.00 | 100.00 | 100.00 | 100.00 | 100.00 | 100.00 | 100.00 | 100.00 | 100.00 |
| | 10 | 100.00 | 100.00 | 100.00 | 100.00 | 100.00 | 100.00 | 100.00 | 100.00 | 100.00 | 100.00 |
| | 15 | 100.00 | 100.00 | 100.00 | 100.00 | 100.00 | 100.00 | 100.00 | 100.00 | 100.00 | 100.00 |
| | 20 | 100.00 | 100.00 | 100.00 | 100.00 | 100.00 | 100.00 | 100.00 | 100.00 | 100.00 | 100.00 |
| | 25 | 100.00 | 100.00 | 100.00 | 100.00 | 100.00 | 100.00 | 100.00 | 100.00 | 100.00 | 100.00 |
| | 30 | 100.00 | 100.00 | 100.00 | 100.00 | 100.00 | 100.00 | 100.00 | 100.00 | 100.00 | 100.00 |
| | 35 | 100.00 | 100.00 | 100.00 | 100.00 | 100.00 | 100.00 | 100.00 | 100.00 | 100.00 | 100.00 |
| | 40 | 100.00 | 100.00 | 100.00 | 100.00 | 100.00 | 100.00 | 100.00 | 100.00 | 100.00 | 100.00 |
| | 45 | 100.00 | 100.00 | 100.00 | 100.00 | 100.00 | 100.00 | 100.00 | 100.00 | 100.00 | 100.00 |
| | 50 | 100.00 | 100.00 | 100.00 | 100.00 | 100.00 | 100.00 | 100.00 | 100.00 | 100.00 | 100.00 |
| XGB | 5 | 87.66 | 91.56 | 94.14 | 94.27 | 95.33 | 96.28 | 97.12 | 97.34 | 98.02 | 98.19 |
| | 10 | 90.45 | 95.79 | 96.01 | 97.95 | 97.86 | 98.57 | 98.39 | 98.73 | 98.64 | 99.38 |
| | 15 | 93.13 | 96.53 | 96.93 | 97.94 | 98.05 | 98.40 | 98.69 | 98.82 | 98.92 | 99.32 |
| | 20 | 94.71 | 97.87 | 96.89 | 98.26 | 98.34 | 98.49 | 99.13 | 99.00 | 98.94 | 99.42 |
| | 25 | 95.23 | 97.80 | 97.55 | 98.71 | 98.98 | 98.94 | 99.01 | 99.07 | 98.99 | 99.58 |
| | 30 | 96.14 | 97.76 | 98.65 | 98.84 | 98.62 | 99.00 | 99.07 | 98.98 | 99.38 | 99.38 |
| | 35 | 95.96 | 98.17 | 98.92 | 99.01 | 99.02 | 99.15 | 99.07 | 98.83 | 99.39 | 99.53 |
| | 40 | 96.51 | 98.63 | 99.25 | 99.34 | 99.40 | 99.20 | 99.58 | 99.17 | 99.57 | 99.72 |
| | 45 | 97.43 | 98.97 | 99.27 | 99.37 | 99.39 | 99.13 | 99.54 | 99.54 | 99.58 | 99.85 |
| | 50 | 97.95 | 98.89 | 99.30 | 99.24 | 99.49 | 99.26 | 99.66 | 99.43 | 99.62 | 99.91 |

Table 24: HELOC Average Cost Mean

| MODEL | INITIAL CLUSTERS DIVERSE ACTIONS | 10.0 | 20.0 | 30.0 | 40.0 | 50.0 | 60.0 | 70.0 | 80.0 | 90.0 | 100.0 |
|---|---|---|---|---|---|---|---|---|---|---|---|
| DNN | 5 | 9.03 | 7.44 | 10.79 | 9.14 | 11.58 | 10.49 | 10.20 | 11.21 | 9.67 | 10.55 |
| | 10 | 10.97 | 10.12 | 10.58 | 9.85 | 9.97 | 10.77 | 10.95 | 11.35 | 10.37 | 11.25 |
| | 15 | 10.54 | 11.09 | 10.96 | 10.35 | 10.36 | 10.23 | 10.57 | 11.17 | 9.83 | 9.43 |
| | 20 | 10.85 | 11.27 | 11.35 | 10.38 | 11.87 | 12.11 | 10.98 | 11.87 | 10.19 | 10.50 |
| | 25 | 10.80 | 11.30 | 11.86 | 10.13 | 10.76 | 11.54 | 11.51 | 11.23 | 10.09 | 11.75 |
| | 30 | 11.70 | 11.08 | 10.87 | 12.77 | 10.92 | 11.51 | 10.96 | 10.80 | 10.34 | 9.54 |
| | 35 | 11.70 | 11.06 | 10.86 | 11.70 | 11.97 | 12.22 | 10.54 | 11.08 | 10.11 | 10.82 |
| | 40 | 12.03 | 11.06 | 11.01 | 11.29 | 11.56 | 12.22 | 10.87 | 11.48 | 10.12 | 10.81 |
| | 45 | 12.03 | 11.25 | 11.63 | 12.47 | 11.59 | 12.65 | 11.90 | 9.85 | 10.21 | 8.84 |
| | 50 | 12.03 | 11.20 | 11.62 | 12.17 | 11.59 | 11.64 | 11.67 | 10.70 | 11.39 | 11.52 |
| LR | 5 | 2.67 | 2.40 | 1.92 | 1.77 | 1.78 | 1.92 | 2.00 | 1.72 | 1.66 | 1.68 |
| | 10 | 1.91 | 2.04 | 1.88 | 1.63 | 1.79 | 1.66 | 1.66 | 1.50 | 1.57 | 1.52 |
| | 15 | 2.09 | 1.73 | 1.65 | 1.77 | 1.48 | 1.41 | 1.56 | 1.49 | 1.46 | 1.35 |
| | 20 | 1.80 | 1.56 | 1.59 | 1.60 | 1.54 | 1.52 | 1.59 | 1.53 | 1.47 | 1.25 |
| | 25 | 1.80 | 1.65 | 1.62 | 1.51 | 1.51 | 1.52 | 1.50 | 1.48 | 1.43 | 1.46 |
| | 30 | 1.69 | 1.62 | 1.54 | 1.51 | 1.57 | 1.38 | 1.52 | 1.52 | 1.43 | 1.32 |
| | 35 | 1.69 | 1.62 | 1.57 | 1.41 | 1.49 | 1.24 | 1.43 | 1.40 | 1.49 | 1.34 |
| | 40 | 1.66 | 1.53 | 1.41 | 1.38 | 1.52 | 1.28 | 1.34 | 1.39 | 1.40 | 1.35 |
| | 45 | 1.64 | 1.53 | 1.50 | 1.40 | 1.46 | 1.30 | 1.43 | 1.39 | 1.37 | 1.31 |
| | 50 | 1.64 | 1.47 | 1.44 | 1.31 | 1.46 | 1.27 | 1.37 | 1.43 | 1.23 | 1.35 |
| XGB | 5 | 12.22 | 14.05 | 15.44 | 14.77 | 15.91 | 16.25 | 17.14 | 17.13 | 17.87 | 18.28 |
| | 10 | 13.83 | 16.14 | 16.51 | 18.75 | 17.54 | 17.57 | 20.25 | 20.62 | 21.32 | 25.72 |
| | 15 | 15.33 | 18.93 | 17.05 | 17.40 | 18.07 | 21.27 | 19.51 | 21.48 | 27.93 | 18.61 |
| | 20 | 14.92 | 18.59 | 16.79 | 20.42 | 21.98 | 23.13 | 22.44 | 20.12 | 22.97 | 23.07 |
| | 25 | 16.03 | 20.48 | 17.85 | 19.41 | 22.98 | 22.43 | 21.69 | 25.36 | 27.18 | 22.68 |
| | 30 | 15.74 | 20.30 | 18.35 | 21.33 | 23.88 | 24.85 | 23.86 | 23.11 | 24.39 | 24.77 |
| | 35 | 17.16 | 22.75 | 20.80 | 21.62 | 23.70 | 27.08 | 24.47 | 20.27 | 25.74 | 24.13 |
| | 40 | 16.58 | 25.14 | 21.67 | 23.77 | 24.82 | 29.50 | 26.66 | 23.40 | 26.58 | 29.42 |
| | 45 | 18.55 | 25.83 | 22.86 | 25.32 | 25.87 | 26.71 | 25.22 | 25.62 | 22.86 | 27.09 |
| | 50 | 18.75 | 23.21 | 26.56 | 24.85 | 28.62 | 27.21 | 24.82 | 28.62 | 24.27 | 26.10 |

