# OpenReview forum: "GLANCE: Global Actions in a Nutshell for Counterfactual Explainability"
_ICLR.cc/2025/Conference — Submitted to ICLR 2025_

### Official Review · Reviewer_pNKM · 2024-10-27

**Soundness:** 3
**Presentation:** 3
**Contribution:** 3
**Rating:** 5
**Confidence:** 4

**Summary:**

This paper proposes GLANCE,  a technique used to provide actionable feedback to individuals impacted by machine learning decisions. The framework focuses on creating global counterfactual explanations (GCE) that can be applied across larger groups instead of individual cases, making it more useful for real-world applications like loan approvals or hiring. The GLANCE framework optimizes three factors for effective recourse: minimizing the number of actions to keep it interpretable, ensuring the actions are cost-effective, and maximizing coverage so that most individuals benefit from these actions.

GLANCE consists of two algorithms: C-GLANCE, which clusters individuals based on features and counterfactual actions to create generalizable solutions, and T-GLANCE, which uses a tree structure for more flexible and interpretable group-specific actions.

**Strengths:**

The paper addresses a problem relevant to many real-world users of explanation systems.

The work is easy to read and seems mathematically sound. The experiment section is sane and all the claims in the intially sections seem to be justified through the paper.

I like the premise of this paper. I'm willing to increase my score if the authors do the suggested work.

**Weaknesses:**

It seems that Equi-explanation maps[1] also divides the feature subspace into regions that have similar recourse options, but in a top down manner. They also seem to compute sub-space centroids, nearest decision boundary points etc and even merge the clusters under certain constraints. Given that these works are very similar, I think it is important that the authors place their work against Equi-explanation maps, at least in the related work section, and preferably treat it as a competing system.

Additionally, I think it would be good to evaluate the effectiveness of such algorithms via a user study. While recourse cost is a good human-engineered metric, it would be interesting to see if the recourse computed by different algorithms were equally helpful to end users.

[1] Chowdhury, Tanya, Razieh Rahimi, and James Allan. "Equi-explanation maps: concise and informative global summary explanations." Proceedings of the 2022 ACM Conference on Fairness, Accountability, and Transparency. 2022.

**Questions:**

Why did you prefer a clustering approach over a top-down division approach in order to create subspaces? Can you discuss the pros and cons of each approach for solving the recourse problem in explanations?

What would a user study design, to test the real world effectiveness of this algorithm as compared to competing algorithms look like?

If deployed in the real world today, what would bad edge cases for the proposed algorithm look like? Are there scenarios where the recourse algorithm can malfunction? Can it suggest incorrect recourse in any case?

---

> ### Author Response · Authors · 2024-11-23
>
> ### W1. It seems that Equi-explanation maps[1] also divides the feature subspace into regions that have similar recourse options, but in a top down manner. They also seem to compute sub-space centroids, nearest decision boundary points etc and even merge the clusters under certain constraints. Given that these works are very similar, I think it is important that the authors place their work against Equi-explanation maps, at least in the related work section, and preferably treat it as a competing system.
>
> Thank you for bringing this paper to our attention. This approach is indeed intriguing for creating subspaces that are explained cohesively and consistently. While Equi-explanation maps also consider aspects like subspace centroids and nearest decision boundary points, their focus is on Global Explanations rather than Global Counterfactual Actions. Additionally, while they address a counterfactual-like problem through inverse classification, their approach differs fundamentally from ours, making a direct comparison with our work challenging.
>
> That said, we believe it could be even more interesting to explore whether equi-explanation maps could be integrated into GLANCE, potentially replacing other clustering techniques, to assess whether they can enhance our results. Unfortunately, we were unable to locate the code for this work, which prevented us from conducting additional experiments.
>
> We have added a dedicated paragraph on global explainability in Section 1.2, shown in slate blue.
>
> ----
>
> ### W2. Additionally, I think it would be good to evaluate the effectiveness of such algorithms via a user study. While recourse cost is a good human-engineered metric, it would be interesting to see if the recourse computed by different algorithms were equally helpful to end users.
>
> Thank you for highlighting this opportunity to improve our work. After reviewing the design of the user studies in GLOBE-CE and Equi-explanation maps, we have designed a user study, with the following goals.
> - Assess (a) how different concepts for global counterfactual explainability are understood and (b) which concepts are preferred by **individuals who seek recourse**.
> - Assess (a) how different concepts for global counterfactual explainability are understood and (b) which concepts are preferred by the **system/model owner who seeks to provide recourse** to the affected population.
> - Assess how people evaluate the **trade-off** between **effectiveness** and **average recourse cost**.
>
> We have included more information about the user study in the revised paper, in Appendix I.

---

> ### Author Response · Authors · 2024-11-23
>
> ### Q1. Why did you prefer a clustering approach over a top-down division approach in order to create subspaces? Can you discuss the pros and cons of each approach for solving the recourse problem in explanations?
>
> We chose to use both clustering (C-GLANCE) and top-down division (T-GLANCE) approaches for subspace creation because they address complementary needs and provide a robust, versatile framework. Clustering offers data-driven subspace formation, aligning naturally with inherent groupings in the data while optimizing feature and action spaces simultaneously to balance interpretability and effectiveness. It is particularly well-suited for complex decision boundaries and diverse datasets but lacks the hierarchical structure and user control that some applications require. In contrast, the top-down division provides hierarchical, multi-granularity solutions that are intuitive and interpretable, allowing user-defined features to guide the partitioning process in alignment with specific policy or audit goals. However, it may not always align with natural data groupings, potentially reducing actionability, and relies on predefined split conditions, limiting flexibility.
>
> Each method is tailored to different use cases and we can leverage the strengths of each algorithm—C-GLANCE for compact, data-driven solutions tailored to actionability and T-GLANCE for structured, user-guided methodologies—ensuring effectiveness across diverse scenarios. We can add such a discussion in the supplementary of the paper for a more detailed explanation of the motivation and strengths behind each approach.
>
>
> ----
>
> ### Q2. What would a user study design, to test the real world effectiveness of this algorithm as compared to competing algorithms look like?
>
> See response to W2.
>
> ----
>
> ### Q3. If deployed in the real world today, what would bad edge cases for the proposed algorithm look like? Are there scenarios where the recourse algorithm can malfunction? Can it suggest incorrect recourse in any case?
>
> While the proposed algorithms are designed to provide robust and actionable recourse, certain edge cases may pose challenges:
> 1. **Highly Nonlinear Decision Boundaries**
> The model’s decision boundary might vary significantly across small regions, leading to oversimplified recourse suggestions. We account for this by using proximity-aware clustering and local counterfactual action generation to better approximate the decision boundary.
> 2. **Data Sparsity and Outliers**
> Sparse data regions or outlier points might result in unrealistic, infeasible, or overly tailored recourse actions. We mitigate these issues by generating many clusters and diverse counterfactuals, ensuring that feasible solutions are found without being overly influenced by outliers.
> 3. **Ambiguity in Selecting the “Optimal” Action**
> Multiple actions with similar costs and effectiveness might lead to overly complex recourse suggestions. We resolve this through predefined criteria that prioritize actionable and interpretable solutions while balancing cost and effectiveness.
>
> We have included this discussion in a dedicated section (Appendix J) in the revised paper.

---

> > ### Comment · Reviewer_pNKM · 2024-11-25
> > **Post Rebuttal Update**
> >
> > While I greatly appreciate the effort and the revisions made, I have reservations about accepting the work without the user study being completed. I value the authors' dedication to addressing the questions posed, especially their design of the user study. However, without the study's results, I remain hesitant to accept this work. I recommend the authors carry out a user study and resubmit their draft.

---

> ### Author Response · Authors · 2024-11-26
>
> ### W2. Additionally, I think it would be good to evaluate the effectiveness of such algorithms via a user study. While recourse cost is a good human-engineered metric, it would be interesting to see if the recourse computed by different algorithms were equally helpful to end users.
>
> *Addendum to Response*
>
> Thank you for the positive feedback on the design of our user study. We have now concluded the user study involving 40 participants and analysed the findings (see Appendix I.2 for details).
>
> We find that:
> - Participants **overwhelmingly prefer information conveyed through action-based GCEs** over direction-based GCEs, whether they are in the role of an individual seeking recourse or a system owner providing recourse to affected individuals. For instance, 71% of participants preferred information associated with concrete actions over information tied to average-cost actions.
> - Participants **unanimously identify GLANCE methods as the superior approach** for both implicit and explicit global counterfactual explainability, when asked to evaluate the average cost and effectiveness of GCEs produced by different methods. Specifically, participants found that C-GLANCE outperforms GLOBE-CE, dGLOBE-CE, and GroupCF in approximately 11 out of 15 cases.

---

### Official Review · Reviewer_uVZG · 2024-10-31

**Soundness:** 3
**Presentation:** 3
**Contribution:** 2
**Rating:** 6
**Confidence:** 4

**Summary:**

This paper proposes a new framework for the global counterfactual explanation problem, which is a task of assigning actions to a given set of individuals by considering their effectiveness, cost, and size. The authors propose two algorithms: C-GLANCE and T-GRANCE. The former first partitions individuals into some clusters and then merges them by taking into account not only the dissimilarity of their centroids but also that of actions generated for the centroids. The latter algorithm recursively partitions individuals like CART, where the best split condition is determined by the improvement in the effectiveness score after splitting. By extensive experiments, the authors demonstrated the effectiveness of the proposed methods compared to the existing baselines, including AReS and CET.

**Strengths:**

Overall, I think this is a good paper, and the authors make solid contributions to the research area of counterfactual explanation.
- S1. This paper is well-written, well-organized, and easy to read.
- S2. I think the proposed framework is simple but effective. The proposed algorithms have only intuitive hyper-parameters, which may be a  desirable advantage compared to AReS and CET whose hyper-parameters are hard to tune.
- S3. The effectiveness of the proposed method was demonstrated through extensive experiments with major baselines for the global counterfactual explanation.

**Weaknesses:**

While the efficacy of the proposed methods is well demonstrated empirically, I believe the following concerns should be addressed to improve the quality of this paper.
- W1. One of my concerns is the computation cost of the proposed algorithms. For both algorithms, we need to generate diverse actions for each centroid many times. In particular, Algorithm 2 needs to compute the total effectiveness gain for all the candidates of split conditions in Line 9, which may incur a significantly high computational cost. To compute the total effectiveness gain for one split condition, we have to generate actions for two centroids after splitting and then compute each effectiveness value. In addition, we need to repeat such a heavy procedure for all the candidate split conditions. Although I recognize the average running times of the proposed algorithms in the experiments were less than roughly a few minutes in Appendix F, I am worried about their scalability when we have more candidates of split conditions.
- W2. Another concern is that this paper includes no theoretical result on the proposed method. I am aware that Theorem 3 in Appendix A shows the hardness of the global counterfactual explanation problem. However, the proposed algorithms are heuristic without theoretical perspectives, and there is no guarantee for the quality of their obtained solution. In addition, there is no analysis of the time complexity of the proposed algorithms as well.

**Questions:**

- Q1. How were the candidates of split conditions of Algorithm 2 determined in the experiments?
- Q2. Could you show analyses of the time complexity of the proposed algorithms?
- Q3 (optional). As far as I remember, AReS and CET have several hyper-parameters. How were these values determined in the experiments?
- Q4 (optional). As the authors mention in Line 285, we can use any dissimilarity score as two distances $d_1$ and $d_2$ in Algorithm 1. What dissimilarity score could be considered appropriate, especially for $d_2$? Also, how do they affect the solution, i.e., partition and actions?
- Q5 (optional). Is there any way to compute the total effectiveness gain more efficiently? For example, in the case of the popular CART-like algorithms, we can compute the information gain for each candidate split condition in an amortized constant time if we have sorted training samples by the value of each feature in advance. In a similar way, can we efficiently calculate the total effectiveness gain?

---

> ### Author Response · Authors · 2024-11-23
>
> ### W1. One of my concerns is the computation cost of the proposed algorithms. [...]
>
> Thank you for raising this concern. While generating diverse actions for each centroid and evaluating split conditions in Algorithm 2 can be computationally intensive, our algorithms are structured to ensure reasonable runtimes (as demonstrated in Appendix H now), where typical runs complete within a few minutes. Notably, runtime is not the primary metric in this context, as this process is executed only once to produce an interpretable solution.
>
> Nonetheless, compared to other methods, our approach achieves a strong balance between computational efficiency and solution quality, as experiments demonstrate. While GLOBE-CE is faster, it performs significantly worse across key metrics. Other methods may require longer runtimes and often fail to provide solutions. In contrast, our algorithms consistently deliver high-quality solutions with near-optimal effectiveness, effectively addressing the inherent complexity of global counterfactual explanations
>
> ----
>
> ### W2. Another concern is that this paper includes no theoretical result on the proposed method. I am aware that Theorem 3 in Appendix A shows the hardness of the global counterfactual explanation problem. However, the proposed algorithms are heuristic without theoretical perspectives, and there is no guarantee for the quality of their obtained solution. In addition, there is no analysis of the time complexity of the proposed algorithms as well.
>
> Thank you for raising these concerns. Our problem does not assume the presence of predefined actions; instead, actions must be generated by the algorithm itself, making the optimal solution inherently not well-defined. The quality of these solutions depends on the distribution of points in the feature space, the nature of the decision boundary (which varies across model-dataset combinations and is difficult to approximate), and our method’s design choices, such as the local counterfactual method.
>
> Even if we assume the optimal actions are available, our problem is a multi-objective optimization problem, not a classical single-objective one. For example, assuming all optimal local counterfactuals are known, a naive Pareto-optimal solution could be constructed by selecting the cheapest local counterfactuals, providing a solution with optimal cost, albeit potentially suboptimal effectiveness. Theoretical analysis would be more feasible in scenarios where the actions were already given, and the objective was to optimize one metric while constraining the other two. For example, as shown in Appendix A, when actions are given and have no associated cost, the problem reduces to the set cover problem. Ultimately, our goal was to prioritize practical, scalable solutions, and our heuristics are designed to perform reliably and efficiently without the need for restrictive assumptions
>
> We agree that adding a time complexity analysis can provide useful insights, and we've included it in Appendix C.

---

> > ### Author Response · Authors · 2024-11-26
> >
> > ### W2. Another concern is that this paper includes no theoretical result on the proposed method. I am aware that Theorem 3 in Appendix A shows the hardness of the global counterfactual explanation problem. However, the proposed algorithms are heuristic without theoretical perspectives, and there is no guarantee for the quality of their obtained solution. [...]
> >
> > *Addendum to Response*
> >
> > Theoretical analysis of global counterfactual explanations is particularly challenging due to the problem’s inherent complexity and variability, as demonstrated by a new experiment. Specifically, we computed all local counterfactuals for instances in the Compas and German Credit datasets and averaged their costs (see Table 20 of Appendix K, row "LOCAL""). Despite utilizing the same local counterfactual method, our approach—which generates only four global counterfactual actions—achieved significantly lower costs than these local counterfactuals.
> >
> > This counterintuitive result highlights the variability in the quality of solutions produced by local counterfactual methods and underscores the difficulty of defining an optimal solution even at the local level. At the global level, the problem becomes even more complex, requiring trade-offs between cost, effectiveness, and interpretability.
> >
> > Additionally, factors such as variability in decision boundaries across model-dataset combinations and the need to account for the distribution of points in the feature space further complicate the analysis. Combined with the multi-objective nature of global counterfactual explanations, these challenges emphasize the practical importance of heuristic approaches like ours.

---

> ### Author Response · Authors · 2024-11-23
>
> ### Q1. How were the candidates of split conditions of Algorithm 2 determined in the experiments?
>
> In our experiments, the split conditions were selected to maximize effectiveness gains, ensuring that each split meaningfully improved subgroup outcomes by optimizing beneficial outcomes across the resulting subgroups.
>
> However, our algorithm is flexible and can be customized to prioritize different objectives. For example, it can be adapted to minimize cost while maintaining effectiveness above specific thresholds or to balance multiple criteria depending on the desired trade-offs for the application.
>
> ----
>
> ### Q2. Could you show analyses of the time complexity of the proposed algorithms?
>
> We have included a time complexity analysis in Appendix C.
>
> ----
>
> ### Q3 (optional). As far as I remember, AReS and CET have several hyper-parameters. How were these values determined in the experiments?
>
> AReS: We used the Fast AReS version, an enhanced implementation of the original AReS method introduced by Rawal & Lakkaraju (2020), as developed by Ley et al. (2023). The code for this version was sourced from the GitHub repository at https://github.com/danwley/GLOBE-CE, and we used the default hyper-parameters provided in this implementation.
>
> CET: For CET, introduced by Kanamori et al. (2022), we employed the original implementation available on GitHub at https://github.com/kelicht/cet. However, we substituted the solver used in their code with the IBM CPLEX solver to solve the Mixed Integer Programming (MIP) problem at each node in the tree. Default hyper-parameters were also used for this method.
>
> We have included this information about hyperparameters in Appendix F.
>
> ----
>
> ### Q4 (optional). As the authors mention in Line 285, we can use any dissimilarity score as two distances and in Algorithm 1. What dissimilarity score could be considered appropriate, especially for ? Also, how do they affect the solution, i.e., partition and actions?
>
> Regarding dissimilarity scores, the Euclidean distance offers simplicity and computational efficiency for well-aligned spaces, while the Manhattan distance is better suited for high-dimensional data with more independent features. The Wasserstein distance, or Earth Mover’s Distance, is particularly valuable for complex or imbalanced data distributions, as it captures structural relationships and aligns distributions in feature and action spaces. However, it is computationally intensive, which may limit its practicality for large-scale datasets or real-time applications.
>
> Regarding the impact of dissimilarity scores, we note the following.
> - Partitions: Different measures influence how clusters are formed. Euclidean and Manhattan distances emphasize geometric proximity, while Wasserstein distance accounts for distributions, creating more coherent clusters in imbalanced or nonlinear datasets.
> - Actions: The clustering approach impacts derived actions, with Wasserstein distance potentially yielding more balanced and lower-cost solutions. However, this comes at the cost of significantly increased computational requirements.
>
>
> ---
>
> ### Q5 (optional). Is there any way to compute the total effectiveness gain more efficiently? [...]
>
> In T-GLANCE, calculating total effectiveness gain involves generating actions for centroids and evaluating their effectiveness, which incorporates both feature and action spaces. This process is inherently more computationally intensive than CART-like algorithms, which rely on predefined metrics like information gain or the Gini index.
>
> While techniques such as reusing intermediate results, approximating gains using data subsets, or parallelizing computations could improve runtime efficiency, they inherently sacrifice solution quality by reducing precision or overlooking important interactions between features and actions. These trade-offs make them less suitable for research-focused or high-quality solutions but could be valuable in practical applications where the algorithm needs to be run frequently, and slightly lower solution quality is acceptable in exchange for faster runtimes.

---

### Official Review · Reviewer_kTWC · 2024-11-03

**Soundness:** 3
**Presentation:** 2
**Contribution:** 2
**Rating:** 6
**Confidence:** 3

**Summary:**

This paper introduces new algorithms for finding global counterfactual explanations (GCE) as a small set of actions that can offer recourse to most of the affected population. Any solution to the GCE problem must navigate a tradeoff among the number of actions, the cost of implementing these actions, and the proportion of the population that can effectively use them for recourse. This work specifically examines the case where the number of actions is fixed and explores the tradeoff between cost and effectiveness.

The paper differentiates between explicit and implicit GCE and proposes two algorithmic frameworks for each. Given that solving GCE with a fixed size is NP-hard in general, the proposed solutions focus on finding efficient GCEs. Empirical evaluations on standard CE datasets show that the proposed approach outperforms existing GCE methods in terms of Pareto optimality and is also computationally efficient.

**Strengths:**

The proposed framework is intuitive, well-designed, and novel. Although it lacks theoretical guarantees, several design choices are particularly effective. For instance, in C-GLANCE, it is insightful to consider the similarity of actions rather than focusing solely on features.

I appreciate how the paper differentiates between explicit and implicit GCE.

The empirical results are strong.

**Weaknesses:**

I think the formulation of GCE as a set of actions rather than directions could benefit from further justification. My default understanding of GCE aligns with GLOBE-CE, which focuses on directions for recourse rather than the actions themselves.

The overall presentation of the paper could be significantly improved. The authors are encouraged to elaborate on the rationale behind certain design choices. I have asked clarification questions in the question section regarding this. In particular, I’m not sure I fully understand Section 4 (T-GLANCE).

I also found it challenging to understand average cost (avc) and the tradeoff with size. If an action is added to the action set C, the recourse cost (rc) for x in the previous C decreases. However, by definition, |X_C| increases, meaning new instances in X_C contribute positively to avc. As a result, it’s unclear how avc behaves as size(C) increases. I expected a definition that clearly illustrates this tradeoff. Given that the results hinge on this concept, further discussion on this choice would be beneficial.

Minor issues: page 3 (computationally efficiently -> efficient), page 5 (min in Problem 1 can be misread as a min of the set), page 8 (can you avoid using red and purple in Table 1? It's hard for me to distinguish them visually)

**Questions:**

When calculating the distance between two clusters, why do you focus only on the distance between the average actions in each cluster? I’m curious if the average is the right summary statistic in this context.

Additionally, when merging clusters, do you retain all of their respective actions? Wouldn’t it be beneficial to prune the action set as part of the merging process?

I’m also unclear on why the evaluations of implicit and explicit GCE are not separated. Further elaboration here could help clarify the distinction between explicit and implicit GCE.

Finally, you might consider using additional metrics beyond Pareto dominance. For instance, Tables 2 and 3 suggest that Fast AReS and GLANCE perform similarly, but Table 1 gives the impression that GLANCE has notably better effectiveness.

---

> ### Author Response · Authors · 2024-11-23
>
> ### W1. I think the formulation of GCE as a set of actions rather than directions could benefit from further justification. My default understanding of GCE aligns with GLOBE-CE, which focuses on directions for recourse rather than the actions themselves.
>
> We prefer an action-based formulation over directions for several key reasons:
> 1. **Clear Endpoints**: Directions fail to specify the magnitude of change required, leading to uncertainty for individuals seeking recourse. Actions, on the other hand, provide specific, interpretable steps, such as "increase income by $500," ensuring actionable and reliable guidance.
> 2. **Interpretability**: Directions often result in numerous individualized "micro-actions," which dilute interpretability. By contrast, actions can be seen as structured, endpoint-specific cases of directions, offering a balance between simplicity and granularity.
> 3. **Real-World Applicability**: Concrete actions ensure clarity and feasibility in practical scenarios, eliminating the ambiguity inherent in abstract directional guidance (e.g., "increase income by $500" vs. "increase income"). This makes actions more aligned with real-world needs and implementation.
>
> Section 1.2 now better reflects this discussion, with changes shown in slate blue.
>
> ----
>
> ### W2. The overall presentation of the paper could be significantly improved. The authors are encouraged to elaborate on the rationale behind certain design choices. I have asked clarification questions in the question section regarding this. In particular, I’m not sure I fully understand Section 4 (T-GLANCE).
>
> We have revised Section 4 to enhance readability, with changes highlighted in slate blue. We address the more specific questions in the following responses.
>
> ----
>
> ### W3. I also found it challenging to understand average cost (avc) and the tradeoff with size. If an action is added to the action set C, the recourse cost (rc) for x in the previous C decreases. However, by definition, |X_C| increases, meaning new instances in X_C contribute positively to avc. As a result, it’s unclear how avc behaves as size(C) increases. I expected a definition that clearly illustrates this tradeoff. Given that the results hinge on this concept, further discussion on this choice would be beneficial.
>
> We thank the reviewer for this observation. For clarity regarding the definition and computation of average cost, please refer to our response to Q1 of Reviewer jVqt, where we have provided a detailed explanation of how cost, recourse cost, and average cost are defined. In what follows, we discuss how the average cost (avc) of C behaves if we add one new action to it (i.e., size(C) increases by 1).
>
> Case 1: The new action does not flip any new instances.
> - If the new action is cheaper than the current actions:
> Some instances already flipped by existing actions may now be flipped by the new action, thereby decreasing the average cost.
> - If the new action is costlier than the current actions:
> The already flipped instances will continue to receive recourse from the existing actions, so the average cost remains unchanged.
>
> Case 2: The new action flips new instances.
> - If the cost of the new action is lower than the current average cost:
> The new action will decrease the average cost, as it introduces recourse for additional instances at a lower cost.
> - If the cost of the new action is higher than the current average cost:
> The average cost will increase, as the new instances contribute positively to the total cost, pulling the average upward.
>
> These cases demonstrate how the addition of new actions interacts with the size-cost tradeoff. Notably, achieving near-optimal effectiveness (close to 100%) often requires adding actions that address outliers, which may have higher costs. However, one of the strengths of our method is its ability to achieve near-optimal effectiveness with relatively low average costs across most datasets.
>
> This discussion can be found in Appendix B in slate blue, with additional comments on the trade-off between size and effectiveness.

---

> ### Author Response · Authors · 2024-11-23
>
> ### Q1. When calculating the distance between two clusters, why do you focus only on the distance between the average actions in each cluster? I’m curious if the average is the right summary statistic in this context.
>
> Thank you for this question. We chose the average action as a summary statistic to efficiently capture the overall characteristics of each cluster, balancing both interpretability and computational efficiency. Focusing on average actions, which captures the central trend, enables a straightforward measure of similarity, which is especially useful given that our method prioritizes interpretability in global explanations.
>
> ----
> ### Q2. Additionally, when merging clusters, do you retain all of their respective actions? Wouldn’t it be beneficial to prune the action set as part of the merging process?
>
> We retain all actions to preserve diversity and ensure maximum solution quality. While pruning can simplify the action set and slightly improve runtime, it comes at the cost of potentially reduced effectiveness, particularly for edge cases. For users prioritizing runtime over quality, optional pruning mechanisms can be enabled to balance these trade-offs according to specific needs
>
> ----
>
> ### Q3. I’m also unclear on why the evaluations of implicit and explicit GCE are not separated. Further elaboration here could help clarify the distinction between explicit and implicit GCE.
>
> We opted for a unified evaluation to compare our methods against all relevant baselines while maintaining conciseness. However, in the revised paper, we improve clarity by distinguishing between implicit and explicit GCE within the evaluation section (Section 5), across all tables.
>
> ----
>
> ### Q4. Finally, you might consider using additional metrics beyond Pareto dominance. For instance, Tables 2 and 3 suggest that Fast AReS and GLANCE perform similarly, but Table 1 gives the impression that GLANCE has notably better effectiveness.
>
> As shown in Table 1, our methods significantly outperform CET and Fast AReS in terms of effectiveness, with Fast AReS failing to exceed 80% effectiveness in any case and CET achieving near-optimal effectiveness in less than half the cases while sometimes failing to find a solution. These low effectiveness levels make a fair comparison challenging, as solving for a small subset of the population is inherently easier.
>
> While Table 1 highlights our superior effectiveness, the Pareto comparisons in Table 2 do not fully capture this advantage, as they require superior performance in both metrics. Low-effectiveness solutions from other methods skew these comparisons (note that solutions with 0% effectiveness have zero cost and are never dominated). Table 3 shows an increased domination rate when low-effectiveness solutions are excluded, though the total comparisons also decrease.
>
> To further investigate this, in Appendix H2 (Table 9), we demonstrate that by intentionally lowering our solutions' effectiveness and selecting cost-efficient actions, our approach can still dominate other methods by achieving both greater effectiveness and lower cost. This highlights the adaptability of our methodology, capable of maintaining superior performance even when optimizing for different trade-offs.

---

> > ### Comment · Reviewer_kTWC · 2024-11-26
> >
> > Thanks for your response.
> >
> > Section 4 seems to be more clear tho it's been a while since I read your paper.
> >
> > The answers to my questions improved my understanding.
> >
> > I also like the updated results that distinguish explicit and implicit approaches.
> >
> > Due to these improvements, I increased my overall rating from 5 to 6.

---

> > > ### Author Response · Authors · 2024-11-26
> > >
> > > ### Q4. Finally, you might consider using additional metrics beyond Pareto dominance. For instance, Tables 2 and 3 suggest that Fast AReS and GLANCE perform similarly, but Table 1 gives the impression that GLANCE has notably better effectiveness.
> > >
> > >
> > > *Addendum to Response*
> > >
> > > We have now completed a user study involving 40 participants, where we explicitly asked them to evaluate the solutions generated by all GCE methods in terms of average cost and effectiveness, beyond Pareto dominance. We find that participants **unanimously identify GLANCE methods as the superior approach** for both implicit and explicit global counterfactual explainability. Specifically, for explicit GCE, participants found that T-GLANCE outperformed the other methods (ARES, CET) approximately 13.41 out of 15 times. For implicit GCE, participants found that C-GLANCE outperformed the other methods (GLOBE-CE, dGLOBE-CE, GroupCF) approximately 10.81 out of 15 times.

---

### Official Review · Reviewer_jVqt · 2024-11-04

**Soundness:** 3
**Presentation:** 2
**Contribution:** 2
**Rating:** 5
**Confidence:** 3

**Summary:**

The paper introduces two heuristics for global counterfactual explanations, namely C-Glance and T-Glance. The idea of C-Glance is to cluster instances based on their similarity in the feature space or the possible recourse actions. T-Glance is a decision tree based method. For both the introduced techniques, the goal is to maximise effectiveness and minimise the cost, with a small number of actions.

**Strengths:**

1) The idea of clusters both based on the similarity in the feature space and the possible recourse actions is simple and very relevant.

2) Overall, the paper is clear and easy to follow.

**Weaknesses:**

1) The novelty of the proposed work is very limited, particularly the methodological novelty.

2) It is unclear what are the main innovations and differences with the existing literature (e.g., how is T-Glance different from Kanamori et al., 2022.  Is the idea of clustering according to both features and actions novel?)

3) The discussion of the proposed methods lacks completeness. For example, C-Glance depends on the number of candidate actions, the initial number of clusters, and the number of actions, but there is no adequate discussion on their impact on the results and on how to select them.

**Questions:**

Please, comment on weaknesses 2) and 3).

Furthermore, what is the definition of cost in the paper? It seems defined only for the toy example, is t the same employed in the whole paper?

---

> ### Author Response · Authors · 2024-11-23
>
> ### W1.The novelty of the proposed work is very limited, particularly the methodological novelty.
>
> We appreciate the reviewer’s feedback regarding the novelty of our work. Our methods, C-GLANCE and T-GLANCE, introduce several key contributions:
> 1. Joint Clustering in Feature and Action Spaces: We are the first to propose joint clustering. Unlike previous claims that clustering is ineffective for global counterfactuals (see e.g., Kanamori et al.), we demonstrate its power when performed jointly in feature and action spaces, producing interpretable and actionable global counterfactuals across large populations.
> 2. Integration of Local Methods: We bridge local and global explanations by combining clustering with local counterfactual generation, offering a scalable and effective methodology for global counterfactuals.
> 3. Actionability-Interpretability Trade-off: Our methods balance actionable, minimal changes with interpretability through a tunable size objective, ensuring practical and interpretable solutions.
> 4. Policy-Relevant Applications: T-GLANCE uniquely enables user-defined partitions and tailored subgroup actions, directly supporting policymaking and audit scenarios—an application not addressed by existing methods.
>
> These contributions are now better reflected in Section 1.1 and appear in slate blue.
>
> ----
>
> ### W2. It is unclear what are the main innovations and differences with the existing literature (e.g., how is T-Glance different from Kanamori et al., 2022. Is the idea of clustering according to both features and actions novel?)
>
> The novelties of GLANCE were discussed in detail in our response to W1.
>
> T-GLANCE introduces several methodological distinctions and innovations compared to Kanamori et al. (2022) and other approaches:
> 1. User-Defined Feature Selection and Structured Tree Construction: T-GLANCE allows users to define branching features, providing transparency and flexibility for aligning the tree structure with specific priorities, unlike Kanamori et al.’s stochastic local search approach.
> 2. Multi-Level Solutions: T-GLANCE offers counterfactual actions at every level of the tree, enabling both high-level and granular solutions. Kanamori et al. optimize actions only at the leaf nodes, limiting their adaptability.
> 3. Multi-Criteria Optimization: T-GLANCE balances interpretability, cost, and effectiveness, providing tailored solutions for practical applications. This contrasts with Kanamori et al.’s focus on rule selection without explicit multi-criteria management.
> 4. Deterministic Solution Guarantees: Unlike the stochastic local search used by Kanamori et al., which may not converge (in about 20% of our experiments), T-GLANCE ensures feasible and interpretable solutions at every node.
>
> This discussion is now reflected in Section 4, at the approach strengths paragraph in slate blue.
>
> ----
>
>
> ### Q1. Furthermore, what is the definition of cost in the paper? It seems defined only for the toy example, is the same employed in the whole paper?
>
> We acknowledge that the terms "cost," "average cost," and "recourse cost" may have been used interchangeably in the paper, which might have caused confusion. These terms have distinct meanings in our work and are consistent with literature:
> - Action Cost: Defined as the L1 norm of the action (consistent with GLOBE-CE).
> - Recourse Cost:  The minimum cost of the action that flips the individual’s class.
> - Average Cost: The average recourse cost across the population that achieves recourse; this quantifies the cost of a GCE solution.
>
> To address this concern, we have clarified the definitions in Section 2 of the revised paper. Additionally, we ensured that all tables consistently refer to average cost. All changes are shown in slate blue.

---

> > ### Comment · Reviewer_jVqt · 2024-11-26
> >
> > Thanks for your rebuttal, in particular wrt W2. However, the methodological novelty of the paper remains limited and W3 has not been discussed. As such, I will keep the evaluation unchanged.

---

> > > ### Author Response · Authors · 2024-11-26
> > >
> > > ### W3. The discussion of the proposed methods lacks completeness. For example, C-Glance depends on the number of candidate actions, the initial number of clusters, and the number of actions, but there is no adequate discussion on their impact on the results and on how to select them.
> > >
> > > We recognize the importance of explaining the parameters in C-GLANCE and their impact on solution quality, interpretability, and runtime. We have performed an additional experiment to study this.
> > >
> > > **Number of Initial Clusters and Diverse Actions**: The product of these parameters determines the total number of generated actions. Increasing the number of actions generally improves effectiveness but often results in higher cost. However, when actions with optimal effectiveness are increased, the overall cost tends to decrease. This pattern is illustrated in a new experiment reported in Appendix L, Tables 21 to 24. Specifically, increasing the number of initial clusters is especially beneficial for larger datasets with widely distributed points, as it allows for better grouping in feature space. On the other hand, increasing the diversity of candidate actions is more advantageous for complex models with intricate decision boundaries, as it enables the generation of counterfactuals that better align with the model's structure.
> > >
> > > **Number of Final Actions**: This user-defined parameter balances interpretability and effectiveness. A lower threshold results in simpler, more interpretable solutions with fewer actions, while a higher threshold offers more tailored and effective recourse at the cost of increased complexity. While the main paper presents results for four final actions, Appendix H.3 explores results for eight final actions. These findings show improved metrics across all methods, demonstrating that increasing the number of final actions can enhance both effectiveness and cost efficiency.

---

### Author Response · Authors · 2024-11-26
**Revision Summary**

We sincerely thank all reviewers for their thoughtful evaluation of our work and the valuable feedback they provided. We have now completed all planned revisions, including conducting additional experiments and a user study, and have updated our submission accordingly.

The highlights of our revision are as follows:

1. We **extensively rewrote** several parts of the paper, improving the readability and addressing all concerns raised by the reviewers. All changes, both in the main text and in the appendix, are highlighed with slate blue.


2. We **designed and performed a user study** involving 40 participants (see Appendix I), with two general objectives:
	- Examine how people perceive two approaches to global counterfactual explanations (GCEs): GCEs framed as actions versus GCEs framed as directions. Participants were asked to adopt two perspectives: that of an individual seeking recourse and that of a system owner providing recourse to adversely affected individuals. The key finding is that **participants overwhelmingly prefer information conveyed through action-based GCEs over direction-based GCEs**.
	- Evaluate how participants determine the superiority of GCE methods based on the average cost and effectiveness of the explanations generated. The primary finding is that **participants unanimously identify GLANCE methods as superior**, consistently reporting that these methods outperform others in over 72% of the evaluated cases.


3. We conducted an **additional experiment to examine the impact of key parameters in GLANCE**, such as the number of initial clusters and the number of actions generated for the cluster centroids (see Appendix L). The results generally indicate that increasing both parameters: (a) consistently improves effectiveness, (b) enhances average cost when effectiveness is nearly perfect, and (c) improves effectiveness at a slightly higher average cost when effectiveness is high but not perfect. We provide an explanation for these patterns and offer **guidelines for optimal parameter selection in GLANCE**.


4. We conducted an **additional experiment highlighting the challenges of providing theoretical guarantees** for solutions to the s-GCE problem (see Appendix K). Notably, we demonstrate that GLANCE outputs with four actions achieve the same effectiveness but significantly lower average cost compared to the set of all local counterfactual explanations, effectively *dominating* them. This counterintuitive result arises because GLANCE generates counterfactual actions *after jointly clustering* in the feature and action spaces. Therefore, providing theoretical guarantees in the local setting is challenging, and this difficulty is further compounded in the more complex global setting, which involves multiple objectives.

---

### Meta-Review · Area_Chair_E3Ps · 2024-12-23

**Metareview:**

The paper proposes two new counterfactual explanation methods, C-Glance and T-Glance where the former uses clustering over features and recommended recourses to enable counterfactual explanations (minimize cost, maximize individuals getting valid recourse) and T-glance uses decision trees to accomplish the same. Empirical demonstrates show improvement over existing baselines.

Overall the contribution is easy to follow and the paper is overall well written. However, reviewers have raised several concerns including some lack of clarity in the writing with very limited methodological novelty, especially when compared against existing prior works reviewers have highlighted. There are also concerns regarding computational cost, and lack of theoretical grounding.

During the rebuttal authors have responded to many of the reviewer concerns, including adding a user study demonstrating the benefit of their method compared to existing approaches, clarifying novelty, and distinguishing from prior work. While some reviewers have raised their score based on the rebuttal, during the discussion, reviewers feel that the contribution's novelty and overall improvements still are insufficient to make for a high-quality ICLR paper. As such in agreement with reviewers' comments, I recommend a rejection.

**Additional Comments On Reviewer Discussion:**

Reviewers discussed that details of the user study are insufficient to enable rigorous evaluation and the existing comparison has limitations. Therefore I agree with the reviewers that the paper cannot be accepted in its current state.

---

### Decision · Program_Chairs · 2025-01-22

Reject